**EMBO** *reports*

# DNMT3B PWWP mutations cause hypermethylation of heterochromatin

Francesca Taglini[1,2], Ioannis Kafetzopoulos [1,2,5,11], Willow Rolls [1,3,11], Kamila Irena Musialik [1,2,6], Heng Yang Lee [1,2,7], Yujie Zhang [3], Mattia Marenda [4], Lyndsay Kerr [1,8], Hannah Finan [1,2,9], Cristina Rubio-Ramon[1,2,10], Philippe Gautier [1], Hannah Wapenaar [3], Dhananjay Kumar [3], Hazel Davidson-Smith[1], Jimi Wills [2], Laura C Murphy [1], Ann Wheeler [1], Marcus D Wilson [3✉] & Duncan Sproul [1,2✉]

## Abstract

The correct establishment of DNA methylation patterns is vital for mammalian development and is achieved by the de novo DNA methyltransferases DNMT3A and DNMT3B. DNMT3B localises to H3K36me3 at actively transcribing gene bodies via its PWWP domain. It also functions at heterochromatin through an unknown recruitment mechanism. Here, we find that knockout of DNMT3B causes loss of methylation predominantly at H3K9me3-marked heterochromatin and that DNMT3B PWWP domain mutations or deletion result in striking increases of methylation in H3K9me3-marked heterochromatin. Removal of the N-terminal region of DNMT3B affects its ability to methylate H3K9me3-marked regions. This region of DNMT3B directly interacts with HP1α and facilitates the bridging of DNMT3B with H3K9me3-marked nucleosomes in vitro. Our results suggest that DNMT3B is recruited to H3K9me3-marked heterochromatin in a PWWP-independent manner that is facilitated by the protein's N-terminal region through an interaction with a key heterochromatin protein. More generally, we suggest that DNMT3B plays a role in DNA methylation homeostasis at heterochromatin, a process which is disrupted in cancer, aging and Immunodeficiency, Centromeric Instability and Facial Anomalies (ICF) syndrome.

**Keywords** DNA Methylation; Epigenetics; Heterochromatin
**Subject Category** Chromatin, Transcription & Genomics

## Introduction

DNA methylation is a repressive epigenetic mark that occurs predominantly on the cytosines of CpG dinucleotides in mammals. The mammalian genome is heavily methylated with the exception of some regulatory elements, particularly CpG islands (Suzuki and Bird, 2008). Studies in mice showed that the DNA methylation landscape is established during early development by the de novo DNA methyltransferases (DNMTs), DNMT3A and DNMT3B (Okano et al, 1999). This developmentally established pattern is then largely maintained by the maintenance DNA methyltransferase DNMT1 (Goll and Bestor, 2005). However, some methylation maintenance is performed by the de novo DNMTs (Elliott et al, 2016; Liang et al, 2002; Liao et al, 2015).

In addition to their catalytic domains, DNMT3A and DNMT3B each possess chromatin reading ADD (ATRX-Dnmt3-Dnmt3L) and PWWP (Pro-Trp-Trp-Pro) domains, which mediate their recruitment to the genome through interaction with histone modifications (Jeltsch and Jurkowska, 2016). The ADD domain of DNMT3A interacts with unmodified H3K4 (Otani et al, 2009; Zhang et al, 2010) to allosterically activate the protein (Guo et al, 2015; Li et al, 2011). This antagonizes DNMT3A's activity at H3K4me3-marked promoters (Hu et al, 2009). The PWWP domains of both proteins bind to methylated H3K36 (Dhayalan et al, 2010; Morselli et al, 2015; Rondelet et al, 2016). DNMT3A's PWWP domain preferentially binds H3K36me2, found at intergenic regions (Weinberg et al, 2019), whereas DNMT3B's PWWP domain shows preference for H3K36me3 (Rondelet et al, 2016; Weinberg et al, 2019), recruiting the protein to the body of actively transcribed genes (Baubec et al, 2015; Morselli et al, 2015; Neri et al, 2017).

*Dnmt3b* is essential for development and knockout mice die at E11.5 (Okano et al, 1999). In humans, *DNMT3B* is mutated in the recessive Mendelian disorder immunodeficiency centromeric

[1]MRC Human Genetics Unit, Institute of Genetics and Cancer, University of Edinburgh, Edinburgh, UK. [2]CRUK Edinburgh Centre, Institute of Genetics and Cancer, University of Edinburgh, Edinburgh, UK. [3]Wellcome Centre for Cell Biology, University of Edinburgh, Edinburgh, UK. [4]IEO, European Institute of Oncology IRCCS, Department of Experimental Oncology, Milan, Italy. [5]Present address: Altos Labs, Cambridge Institute, Cambridge, UK. [6]Present address: MRC London Institute of Medical Sciences and Institute of Clinical Sciences, Imperial College London, London, UK. [7]Present address: Endocrine Oncology Research Group, Department of Surgery, The Royal College of Surgeons RCSI, University of Medicine and Health Sciences, Dublin, Ireland. [8]Present address: Department of Mathematics and Statistics, University of Strathclyde, Glasgow, UK. [9]Present address: Swiss Federal Institute of Technology, ETH Zürich, Institute of Molecular Health Sciences, Zürich, Switzerland. [10]Present address: Université Paris Cité, CNRS, Institut Jacques Monod, Paris, France. [11]These authors contributed equally: Ioannis Kafetzopoulos, Willow Rolls. ✉E-mail: Marcus.Wilson@ed.ac.uk; d.sproul@ed.ac.uk

instability and facial abnormalities type 1 syndrome (ICF1) (Okano et al, 1999; Xu et al, 1999). The majority of missense mutations causing ICF1 syndrome occur in the catalytic domain of DNMT3B (Weemaes et al, 2013). However, a missense mutation of the PWWP domain, S270P, has also been identified in ICF1 (Shirohzu et al, 2002). ICF1 is characterised by widespread hypomethylation of satellite repeats and other heterochromatic loci (Heyn et al, 2012; Jeanpierre et al, 1993). DNMT3B also localises to heterochromatic chromocenters by microscopy in mouse cells (Bachman et al, 2001). These constitutive heterochromatic loci are associated with the histone modification H3K9me3 (Allshire and Madhani, 2018; Brandle et al, 2022). Losses of DNA methylation from constitutive heterochromatin are also frequently observed in cancer and lead to the formation of large partially methylated domains (PMDs) (Berman et al, 2011; Hansen et al, 2011). Similar losses of DNA methylation from heterochromatin and PMD formation are observed in aging (Zhou et al, 2018).

Using ChIP-seq, the majority of DNMT3B localises to H3K36me3 (Baubec et al, 2015) and it is therefore unclear how the protein is recruited to H3K9me3-marked constitutive heterochromatin. However, removal of the PWWP domain is reported to abolish DNMT3B localisation to chromocenters in mouse cells (Chen et al, 2004; Ge et al, 2004). The ICF1 syndrome S270P mutation has also been reported to impair localisation to chromocenters (Chen et al, 2004; Ge et al, 2004), suggesting that the PWWP domain may play a role in DNMT3B's recruitment to heterochromatin in addition to its better described role at H3K36me3.

To understand how DNMT3B is recruited to the genome, we have analysed *DNMT3B* knockout cells and the effect of DNMT3B mutations on methylation patterns. We show that loss of *DNMT3B* disproportionality results in hypomethylation of heterochromatic H3K9me3 domains, and that mutations or removal of DNMT3B's PWWP cause increased DNA methylation at H3K9me3-marked heterochromatin. Our results suggest that DNMT3B is recruited to constitutive H3K9me3-marked heterochromatin in a PWWP-independent manner. They also suggest that this is facilitated by DNMT3B's N-terminal region through an interaction with HP1α.

## Results

### DNMT3B methylates heterochromatin

To understand the role of DNMT3B in methylating different parts of the genome, we examined DNMT3B knockout HCT116 cells (DNMT3B KO) (Rhee et al, 2002) using whole genome bisulfite sequencing (WGBS, mean autosomal CpG coverage 2.6, see Appendix Table S2). Overall methylation levels were decreased to 96% and 86% of the levels observed in HCT116 wild-type cells by WGBS and mass spectrometry, respectively (Appendix Fig. S1a).

To determine which type of genomic region was most affected by the loss of DNMT3B, we performed ChIP-seq for the histone modifications: H3K4me3 (promoter-associated), H3K36me3 (gene body-associated), H3K9me3 (constitutive heterochromatin-associated) and H3K27me3 (facultative heterochromatin-associated) in HCT116 cells. We then correlated the levels of these histone modifications to changes in DNA methylation between HCT116 and DNMT3B KO cells in 2.5-kb windows across the genome

(Fig. 1A; Appendix Fig. S1b). Losses of DNA methylation in DNMT3B KO compared to HCT116 cells were significantly correlated with each of the modifications (Appendix Fig. S1b, Pearson correlations, all $P < 2.2 \times 10^{-16}$). However, the correlations with the constitutive H3K9me3 and facultative H3K27me3 heterochromatin modifications (Pearson correlations, $R = -0.124$ and $-0.175$, respectively, Appendix Fig. S1b) were greater in magnitude than those observed with H3K36me3 and H3K4me3 (Pearson correlations, $R = -0.029$ and $-0.012$ respectively, Appendix Fig. S1b).

Given previous observations that DNMT3B is recruited to H3K36me3 (Baubec et al, 2015), we next examined gene bodies which are the primary locations marked by H3K36me3 (Bannister et al, 2005). Despite the role of DNMT3B at H3K36me3, the correlation between DNMT3B KO DNA methylation loss and H3K36me3 levels in gene bodies was very low and non-significant (Pearson correlation, $R = -0.003$, $P$ value $= 0.671$, Appendix Fig. S1c).

We then further examined the relationship between DNMT3B loss and the heterochromatin-associated H3K9me3 and H3K27me3 histone modifications. H3K9me3-marked broad genomic domains of reduced DNA methylation in HCT116 cells (Fig. 1A). In support of this, we found that hidden Markov model-defined H3K9me3 domains significantly overlapped with PMDs defined in our HCT116 WGBS (Jaccard=0.575, $P = 1.07 \times 10^{-6}$, Fisher's test). They also significantly overlapped heterochromatic regions resistant to nuclease digestion identified using Protect-seq in the same cells (Jaccard=0.623, $P = 2.11 \times 10^{-30}$, Fisher's test) (Spracklin and Pradhan, 2020). HCT116 H3K9me3 domains lost significantly more methylation in DNMT3B KO cells than non-heterochromatic domains which were marked by neither H3K9me3 nor H3K27me3 (other domains, $P < 2.2 \times 10^{-16}$ Wilcoxon test, Fig. 1A–C). Although HCT116 H3K9me3 domains showed some enrichment for H3K27me3 (Appendix Fig. S1d), facultative heterochromatic domains marked by H3K27me3 alone lost DNA methylation to a significantly lesser degree in DNMT3B KO cells than H3K9me3 domains ($P < 1.8 \times 10^{-9}$ Wilcoxon test, Fig. 1C). This suggests that DNMT3B is more active in regions of constitutive heterochromatin marked by H3K9me3 than in facultative heterochromatin marked by H3K27me3.

The major catalytically active DNMT3B isoform expressed in somatic cells is DNMT3B2 (Weisenberger et al, 2004). To confirm that DNMT3B methylates DNA at H3K9me3-marked heterochromatin, we expressed DNMT3B2 (henceforth referred to as DNMT3B) in DNMT3B KO cells to generate DNMT3B$^{WT}$ cells (Appendix Fig. S1e, top). We then compared gains of DNA methylation to histone modification levels assayed by ChIP-seq from DNMT3B KO cells. H3K9me3 domains were also observed in DNMT3B KO cells, and we defined them using a hidden Markov model (Fig. 1D). DNMT3B KO H3K9me3 domains significantly overlapped HCT116 H3K9me3 domains (Jaccard=0.582, $P = 9.82 \times 10^{-162}$, Fisher's exact test) but showed a greater enrichment of H3K27me3 than HCT116 H3K9me3 domains (Appendix Fig. S1d,f). Changes in DNA methylation in DNMT3B$^{WT}$ cells significantly correlated with H3K27me3 and H3K9me3 (Pearson correlations, $R = 0.306$ and $0.230$, respectively, both $P < 2.2 \times 10^{-16}$) and to a lesser extent with H3K36me3 and H3K4me3 (Pearson correlations, $R = 0.027$ and $0.143$, respectively, both $P < 2.2 \times 10^{-16}$). In DNMT3B$^{WT}$ cells, gains of DNA methylation at DNMT3B KO

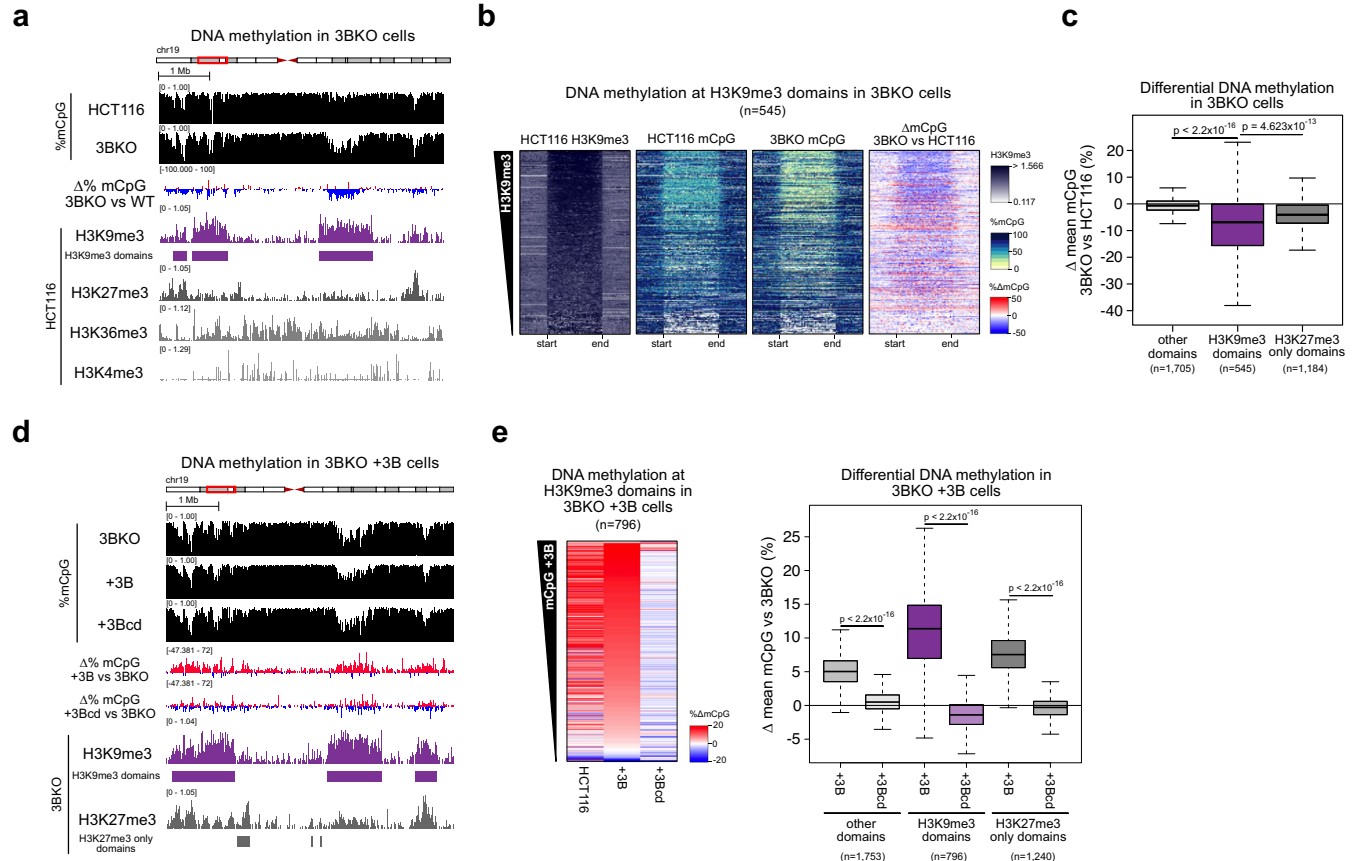

**Figure 1. DNMT3B methylates heterochromatin.**

(A–C) Loss of DNA methylation in DNMT3B KO cells occurs predominantly at heterochromatic domains. (A) Representative genomic location showing loss of DNA methylation at H3K9me3-marked domains in DNMT3B KO cells (3BKO). Genome browser plots showing absolute (black) and differential (gain=red/loss=blue) DNA methylation levels alongside ChIP-seq and for histone modifications from HCT116 cells. ChIP-seq are normalised reads per $10^6$. H3K9me3 domains are defined in HCT116 cells. (B) Heatmaps showing levels of H3K9me3 and of absolute or differential DNA methylation at HCT116 H3K9me3 domains ($n = 545$ domains), in HCT116 and DNMT3B KO cells. The domains are ranked by their mean H3K9me3 levels. (C) Boxplot showing differential DNA methylation between DNMT3B KO and HCT116 at H3K9me3 domains ($n = 545$ domains), H3K27me3-only domains ($n = 1184$) and the domains marked by neither modification (other domains, $n = 1705$) defined in HCT116 cells. (D, E) H3K9me3 domains gain DNA methylation when DNMT3B is re-expressed in DNMT3B KO cells. (D) Representative genomic location showing gain of DNA methylation at H3K9me3 domains in DNMT3B KO cells expressing DNMT3B (+3B) or DNMT3B catalytically dead (+3Bcd). Genome browser plots show DNA methylation levels (absolute in black and gain=red/loss=blue), DNMT3B KO ChIP-seq signals and domains defined in DNMT3B KO cells. ChIP-seq are normalised reads per $10^6$. (E) Left, heatmaps of relative DNA methylation levels at DNMT3B KO cell H3K9me3 domains ($n = 796$ domains). Values denote the change in DNA methylation relative to DNMT3B KO cells; H3K9me3 domains are defined in DNMT3B KO cells and ranked by the mean gain of methylation in DNMT3B KO cells expressing DNMT3B. Right, boxplots of DNA methylation difference to DNMT3B KO cells at H3K9me3 ($n = 796$ domains), H3K27me3-only marked domains ($n = 1240$) and domains not marked by either modification (other domains, $n = 1753$). Data information: For boxplots in (C, E): Lines = median; box = 25th–75th percentile; whiskers = 1.5 × interquartile range from box. *P* values are from two-sided Wilcoxon rank-sum tests. All histone ChIP-seq data shown are derived from the mean of two biological replicates. *P* values for all tests are shown in the figure panels.

H3K9me3 domains were significantly greater than those in domains defined by H3K27me3 alone or lacking either mark (Fig. 1D,E), consistent with greater DNMT3B activity at H3K9me3 than H3K27me3-marked heterochromatin. Gains of DNA methylation in DNMT3B^WT cells were also significantly greater than those observed upon expression of a catalytically inactive mutant DNMT3B (Hsieh, 1999) (DNMT3Bcd, Fig. 1D,E).

To determine whether activity at H3K9me3 was specific to DNMT3B, we also expressed DNMT3A1, the DNMT3A isoform expressed in somatic cells (Chen et al, 2002), and catalytically inactive DNMT3A1 (Hsieh, 1999), in DNMT3B KO cells (Appendix Fig. S1e, bottom). The DNA methylation gains observed upon expression of DNMT3B were significantly greater at

H3K9me3 domains compared to the DNA methylation gains induced by DNMT3A and catalytically inactive DNMT3A1 (Appendix Fig. S1g,h). In contrast, the weaker gains of DNA methylation observed at H3K27me3 domains were more similar between DNMT3B and DNMT3A (Appendix Fig. S1g,h).

To understand the impacts of DNMT3B removal on gene expression in heterochromatin, we performed RNA-seq on HCT116 and DNMT3B KO cells. Analysis of these data revealed that 5188 genes were significantly differentially expressed (2292 upregulated and 2896 downregulated at FDR < 0.05). These differentially expressed genes were significantly enriched in genes located in HCT116 H3K9me3 domains, suggesting that transcriptional disruption was observed in this chromatin compartment in

DNMT3B KO cells (Appendix Fig. S1i, $P = 0.0042$, Fisher's exact test). However, the overlap was modest, suggesting that additional mechanisms were responsible for regulating gene expression of H3K9me3 domains.

Overall, these data from the knockout and re-expression of DNMT3B suggest that DNMT3B specifically methylates constitutive heterochromatic domains marked by H3K9me3.

## Mutation of DNMT3B H3K36me3-binding residues causes gains of DNA methylation at heterochromatin

Having observed a role for DNMT3B in methylating constitutive H3K9me3-marked heterochromatin throughout the genome, we sought to understand the mechanism(s) responsible for its recruitment to this genomic compartment. Previous work has suggested a role for DNMT3B's PWWP in localising it to constitutive heterochromatic chromocenters by microscopy (Chen et al, 2004; Ge et al, 2004). To impair the H3K36me3-binding function of DNMT3B's PWWP, we therefore mutated a key residue involved in methyl-lysine binding, tryptophan 263 to alanine (W263A) (Fig. 2A; Appendix Fig. S2a) and expressed this protein in DNMT3B KO cells (DNMT3B$^{W263A}$ cells). Mutation of the paralogous residue in DNMT3A disrupts H3K36me2/3 binding and causes Heyn–Sproul–Jackson Syndrome (HESJAS) (Heyn et al, 2019). DNMT3B tryptophan 263 forms part of the aromatic cage-binding pocket for recognition of methylated H3 lysine 36 (Appendix Fig. S2a) (Rondelet et al, 2016).

DNMT3B$^{W263A}$ was expressed to a lower level than DNMT3B$^{WT}$ in DNMT3B KO cells (Fig. S2b). We therefore checked for protein stability using a biscistronic reporter expressing GFP-DNMT3B$^{WT}$ or GFP-DNMT3B$^{W263A}$ and dsRed (Huang et al, 2022) in DNMT3B KO cells. In this system, GFP intensity reports DNMT3B levels, and transfection efficiency is controlled for by dsRed intensity. The mean ratio of GFP to dsRed intensity in DNMT3B$^{W263A}$ cells was 91.6% of that observed in DNMT3B$^{WT}$ cells (Appendix Fig. S2c), indicating that the proteins are similarly stable. In addition, the melting temperature of DNMT3B$^{W263A}$ was similar to that of DNMT3B$^{WT}$ in vitro when constructs consisting of the PWWP and ADD domain were assayed by thermal denaturation assays (46.7 and 49.9 °C, respectively, Appendix Fig. S2d).

We then assessed the effect of DNMT3B$^{W263A}$ on the methylome using WGBS. Despite the lower protein level observed, global levels of methylation were higher in DNMT3B$^{W236A}$ cells than in DNMT3B$^{WT}$ cells (Appendix Fig. S2e). Surprisingly given the PWWP domains previous association with heterochromatic localisation (Chen et al, 2004; Ge et al, 2004), gains of DNA methylation in DNMT3B$^{W263A}$ cells versus DNMT3B KO cells were highly significantly correlated with H3K9me3 levels in DNMT3B KO cells (Pearson correlation, $R = 0.516$, $P < 2.2 \times 10^{-16}$, Appendix Fig. S2f, right). This correlation and the absolute gains of DNA methylation were greater than the differences between DNMT3B$^{WT}$ and DNMT3B KO cells (Pearson correlation, $R = 0.299$, $P < 2.2 \times 10^{-16}$, Appendix Fig. S2f, left). In support of this, significantly more methylation was present at heterochromatic H3K9me3 domains in DNMT3B$^{W263A}$ cells than DNMT3B$^{WT}$ cells (Fig. 2B,C). This suggests that the W263A mutation causes the relocalisation of DNMT3B to H3K9me3-marked heterochromatin.

Having seen gains of methylation at interspersed heterochromatin in DNMT3B$^{W326A}$ cells, we then asked if similar gains were

seen at constitutively heterochromatic satellite II repeats which are hypomethylated in ICF1 syndrome (Hassan et al, 2001; Xu et al, 1999). Using a methylation-sensitive Southern blot, we observed greater digestion of satellite II repeats in DNMT3B KO cells as compared to HCT116 cells, consistent with hypomethylation of satellite II repeats upon loss of DNMT3B (Appendix Fig. S2g). Protection from digestion was restored by expression of DNMT3B$^{WT}$ (Appendix Fig. S2g). However, DNMT3B$^{W263A}$ cells displayed an even greater protection from digestion indicating that the mutation results in hypermethylation of satellite II compared to DNMT3B$^{WT}$ (Appendix Fig. S2g).

DNMT3B can interact with DNMT3A (Li et al, 2007). To test whether the increased in DNA methylation observed upon DNMT3B$^{W263A}$ expression was due to DNMT3B activity rather than via recruitment of other DNMT enzymes, we expressed catalytically inactive DNMT3B$^{W263A}$ (DNMT3B$^{W263Acd}$, Appendix Fig. S2h) in DNMT3B KO cells. Methylation-sensitive Southern blot showed that DNMT3B$^{W263Acd}$ failed to methylate satellite II repeats (Appendix Fig. S2i). We also assessed DNA methylation levels of two representative non-repetitive loci within heterochromatic H3K9me3 domains (Appendix Fig. S2j) by bisulfite PCR in these cells. Methylation levels at these loci in DNMT3B$^{W263Acd}$ cells were significantly lower than in DNMT3B$^{W263A}$ cells (Appendix Fig. S2k), suggesting that the methylation heterochromatin observed with the W263A mutation was a direct consequence of DNMT3's catalytic activity.

To validate that the hypermethylation of heterochromatin in DNMT3B$^{W263A}$ cells was caused by the disruption H3K36me3 binding, we then mutated another residue of the DNMT3B PWWP's methyl-lysine binding pocket, aspartic acid 266, to alanine (Fig. 2A; Appendix Fig. S2a) (Dhayalan et al, 2010), and expressed this in DNMT3B KO cells (DNMT3B$^{D266A}$ cells, Appendix Fig. S2b). By methylation-sensitive Southern blot, satellite II repeats were similarly protected from digestion in DNMT3B$^{D266A}$ cells as in DNMT3B$^{W263A}$ cells (Appendix Fig. S2g). We also assessed the methylation level at two representative non-repetitive H3K9me3 loci by bisulfite PCR, finding them to be significantly more methylated in DNMT3B$^{W263A}$ and DNMT3B$^{D266A}$ cells than in DNMT3B$^{WT}$ cells or DNMT3B KO cells expressing GFP (Appendix Fig. S2l).

To confirm that the hypermethylation of H3K9me3-marked heterochromatin observed in DNMT3B$^{W263A}$ cells was not solely a result of ectopically elevated protein levels, we expressed DNMT3B$^{W263A}$ in DNMT3B KO cells using the EF1α promoter which we have previously has lower levels of expression in HCT116 cells than the CAG promoter used in our other experiments (Masalmeh et al, 2021). Consistent with our previous observations, levels of DNA methylation at two representative H3K9me3 loci were significantly higher in DNMT3B$^{W263A}$ cells than DNMT3B$^{WT}$ cells using the EF1α promoter (Appendix Fig. S2m).

To further confirm that hypermethylation of H3K9me3-marked heterochromatin was not due to overexpression of DNMT3B$^{W263A}$, we generated three homozygous *DNMT3B$^{W263A}$* knock-in clonal lines from HCT116 cells using CRISPR-Cas9 (Appendix Fig. S3a). We then compared their DNA methylation pattern to two similarly treated control cell line clones which lacked the mutation using WGBS. DNMT3B$^{W263A}$ knock-in clones showed gains of DNA methylation at H3K9me3 domains compared to parental HCT116 cells, and these gains were significantly greater than the small

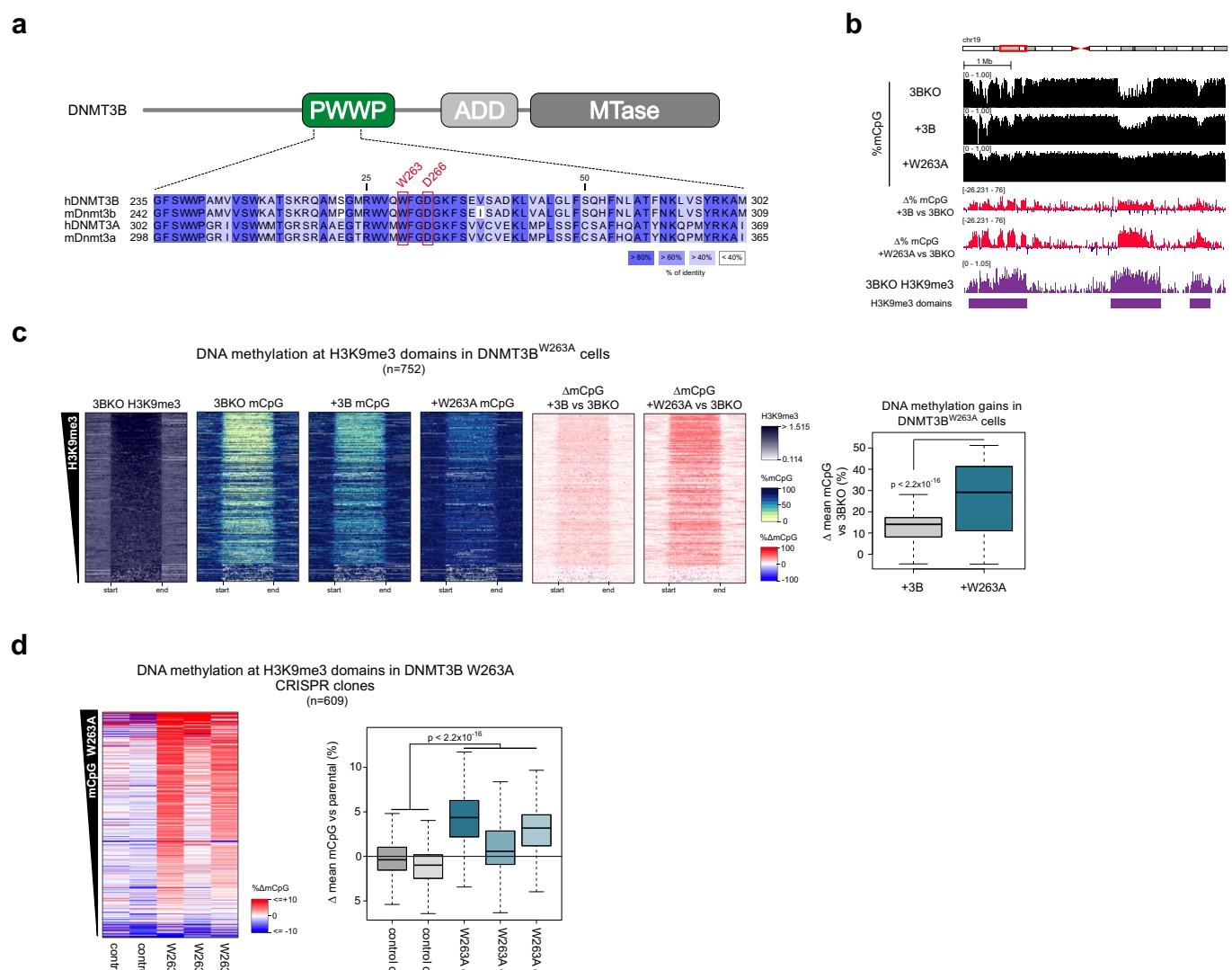

**Figure 2. Mutation of DNMT3B H3K36me3-binding residues causes gains of DNA methylation at heterochromatin.**

(A) Schematic of DNMT3B protein and domains (top) and part of a multiple sequence alignment of DNMT3B's PWWP domain with the DNMT3A and mouse orthologues (bottom). (B, C) DNMT3B^W263A remethylates heterochromatin to a significantly higher level than DNMT3B^WT. (B) Genome browser plots showing absolute (black) and differential (gain=red/loss=blue) DNA methylation levels, DNMT3B KO ChIP-seq signals at H3K9me3 domains in DNMT3B KO, DNMT3B^WT and DNMT3B^W263A cells. ChIP-seq are normalised reads per 10⁶. (C) Left, heatmaps showing levels of H3K9me3 and DNA methylation at H3K9me3 domains (n = 752 domains) in DNMT3B KO, DNMT3B^WT and DNMT3B^W263A cells. The domains are ranked by their mean H3K9me3 levels. Right, boxplot showing gains of DNA methylation at H3K9me3 domains in DNMT3B^WT and DNMT3B^W263A cells. (D) Endogenous expression of DNMT3B^W263A in HCT116 cells leads to hypermethylation of heterochromatin. Left, heatmaps of DNA methylation levels at H3K9me3 domains (n = 609 domains) in CRISPR/Cas9-edited DNMT3B^W263A and control clones relative to parental HCT116 cells. H3K9me3 domains are defined in HCT116 cells and ranked by the mean gain of DNA methylation in DNMT3B^W263A clones. Right, boxplot showing differential DNA methylation at H3K9me3 domains in CRISPR clones compared to the parental cell line. Data information: For boxplots in (C, D): lines = median; box = 25th–75th percentile; whiskers = 1.5 × interquartile range from box. P values are from two-sided Wilcoxon rank-sum tests. All histone ChIP-seq data shown are derived from the mean of two biological replicates. P values for all tests are shown in the figure panels.

differences in DNA methylation seen in control clones (Fig. 2D). We also asked whether the mutation of the PWWP domain affected methylation at H3K36me3 sites by examining the change in methylation in DNMT3B^W263A knock-in clones at gene bodies. Although the correlation between mean change in DNA methylation in DNMT3B^W263A knock-in clones and H3K36me3 levels at gene bodies was significant, it was very weak (Pearson Correlation, $R = -0.037$, $P$ value $= 6.56 \times 10^{-7}$, Appendix Fig. S3b). We also observed no consistent relationship between change in DNA

methylation and HCT116 H3K36me3 levels in the individual DNMT3B^W263A knock-in clones (Appendix Fig. S3b). This observation together with the retention of DNA methylation levels at H3K36me3-marked genes bodies in DNMT3B KO cells (Appendix Fig. S1b,c) suggests that other DNMTs are capable of compensating for DNMT3B in maintaining DNA methylation levels at H3K36me3-marked loci.

We then confirmed that the effect of DNMT3B^W263A at H3K9me3-marked heterochromatin was not specific to HCT116

by expressing the protein in another cell line, RKO (Appendix Fig. S3c). By Southern blot, we observed that RKO cells showed hypermethylation of satellite II upon DNMT3B$^{W263A}$ expression even in the presence of endogenous DNMT3B (Appendix Fig. S3d).

These results therefore suggest that disruption of DNMT3B's H3K36me3 binding through the PWWP domain results in hypermethylation of H3K9me3-marked heterochromatin.

## PWWP mutation leads to increased DNMT3B localisation at heterochromatin

To understand the localisation of DNMT3B$^{W263A}$ in more detail, we performed ChIP-seq analysis of DNMT3B$^{W263A}$ cells and compared this to DNMT3B$^{WT}$ cells.

We first checked that the ectopically expressed T7-tagged DNMT3B localised similarly to endogenous DNMT3B by comparing ectopic DNMT3B localisation in HCT116 cells to our previously published endogenously tagged ChIP-seq of DNMT3B in these cells (Masalmeh et al, 2021). ChIP-seq signal for ectopically expressed DNMT3B was significantly correlated with that of endogenously tagged DNMT3B (Pearson correlation, $R = 0.630$, $P < 2.2 \times 10^{-16}$, Appendix Fig. S4a,b). ChIP-seq signal for both also correlated significantly with H3K36me3 signal across the genome (Pearson correlations $R = 0.395$ and $0.550$, respectively, both $P < 2.2 \times 10^{-16}$, Appendix Fig. S4a), suggesting that ectopically expressed DNMT3B recapitulates the localisation of the endogenous protein.

We then examined the effect of the W263A mutation on DNMT3B localisation in DNMT3B KO cells. The ChIP-seq signal from DNMT3B$^{WT}$ cells correlated with levels of DNMT3B KO H3K36me3 across the genome (Pearson correlation, $R = 0.386$, $P < 2.2 \times 10^{-16}$, Fig. 3A; Appendix Fig. S4c) and in gene bodies (Fig. S4d). In contrast, DNMT3B levels at gene bodies in DNMT3B$^{W263A}$ cells were far lower than DNMT3B$^{WT}$ cells (Fig. 3A; Appendix Fig. S4d). However, DNMT3B$^{W263A}$ was seen at some H3K36me3-marked loci (Fig. 3A, orange arrows), and low enrichment at gene bodies marked by the highest levels of H3K36me3 could still be detected (Appendix Fig. S4d). This suggests DNMT3B$^{W263A}$ impairs but does not completely abolish localisation to H3K36me3 in cells.

In contrast to its correlation with H3K36me3, DNMT3B$^{WT}$ levels negatively correlated with H3K9me3 levels genome-wide in DNMT3B KO cells (Pearson's correlation, $R = -0.083$, $P < 2.2 \times 10^{-16}$, Fig. 3B,C). We did not detect obvious peaks of DNMT3B$^{W263A}$ enrichment within H3K9me3 domains (Fig. 3B). However, in contrast to DNMT3B$^{WT}$, DNMT3B$^{W263A}$ signal was positively correlated with H3K9me3 signal (Pearson's correlation, $R = 0.250$, $P < 2.2 \times 10^{-16}$, Fig. 3C). In agreement with this, DNMT3B$^{W263A}$ enrichment at H3K9me3 domains was significantly higher than at other domains, whereas DNMT3B$^{WT}$ enrichment was significantly lower at H3K9me3 domains than at other domains (Fig. 3D). This is consistent with increased DNMT3B$^{W263A}$ localisation to H3K9me3 and our observations of increased DNA methylation at H3K9me3 in these same cells.

To further understand the localisation of DNMT3B$^{W263A}$ on chromatin using an independent approach, we compared the localisation of SNAP-tagged DNMT3B$^{WT}$ and DNMT3B$^{W263A}$ in live DNMT3B KO cells using confocal microscopy. Both proteins were broadly distributed in the nucleus and showed variable patterns (Fig. 3E). To assay their degree of localisation at heterochromatin, we

then compared their distribution to a marker of H3K9me3-marked constitutive heterochromatin, HP1α-GFP (Fig. 3E). The ratio of DNMT3B signal at HP1α-marked heterochromatin versus the rest of nucleus was significantly higher for DNMT3B$^{W263A}$ than for DNMT3B$^{WT}$, consistent with a greater localisation of DNMT3B$^{W263A}$ to H3K9me3-marked heterochromatin (Fig. 3E; Appendix Fig. S4e). We also assessed the degree of localisation at another constitutive heterochromatic compartment, the nuclear periphery (van Steensel and Belmont, 2017). Consistent with our ChIP-seq results, a significantly greater proportion of DNMT3B$^{W263A}$ signal was found at the nuclear periphery than DNMT3B$^{WT}$ signal (Appendix Fig. S4f).

Taken together, these results suggest that interfering with the interaction of DNMT3B's PWWP with H3K36me3 causes relocalisation of DNMT3B to H3K9me3-marked constitutive heterochromatin.

## The ICF1-associated PWWP S270P mutation destabilises DNMT3B

Having observed that mutations in the PWWP interfering with DNMT3B's interaction with H3K36me3 resulted in increased localisation and gains of methylation at heterochromatin, we next sought to understand the effect of the ICF1-associated PWWP mutation, S270P (Shirohzu et al, 2002) (Appendix Fig. S5a).

Similar to the aromatic cage mutations we have examined, S270P has been reported to reduce the interaction of DNMT3B with H3K36me3-marked nucleosomes (Baubec et al, 2015). Serine 270 is located distal to the methyl-lysine recognising binding pocket (Appendix Fig. S5b) and has been hypothesised to make a hydrogen bond with the backbone of bound H3 peptide (Rondelet et al, 2016). However, in contrast to our observations with aromatic cage PWWP mutations, S270P has been reported to reduce pericentromeric localisation in mouse cells (Chen et al, 2004; Ge et al, 2004).

We therefore expressed DNMT3B$^{S270P}$ in DNMT3B KO cells (DNMT3B$^{S270P}$ cells) and asked whether it increased methylation at satellite II repeats by methylation-sensitive Southern blot. In contrast to the increased protection from digestion observed in DNMT3B$^{WT}$ and DNMT3B$^{W263A}$ cells, the level of digestion in DNMT3B$^{S270P}$ cells was similar to that of DNMT3B KO cells (Appendix Fig. S5c), suggesting that expression of DNMT3B$^{S270P}$ did not lead to an increase of DNA methylation at satellite II. The levels of DNA methylation at two non-repetitive H3K9me3 loci in DNMT3B$^{S270P}$ cells were also similar to those seen in DNMT3B KO cells expressing GFP when measured by bisulfite PCR (Appendix Fig. S5d).

While these data were consistent with previous findings that S270P impairs DNMT3B recruitment to heterochromatin (Chen et al, 2004; Ge et al, 2004), they were inconsistent with its proposed effect on H3K36me3 binding and our observations in DNMT3B$^{W263A}$ and DNMT3B$^{D266A}$ cells. We noticed that DNMT3B$^{S270P}$ cells expressed much lower levels of protein than DNMT3B$^{WT}$ cells by western blot suggesting that DNMT3B$^{S270P}$ might not be stable (Appendix Fig. S5e). Mutations to prolines can often prematurely terminate secondary structures and destabilise proteins. Indeed serine 270 lies within a β-sheet integral to the fold of the PWWP domain. We therefore tested the stability of the DNMT3B$^{S270P}$ mutant. Using our fluorescent reporter protein stability assay, we observed that the ratio of GFP to dsRed intensity

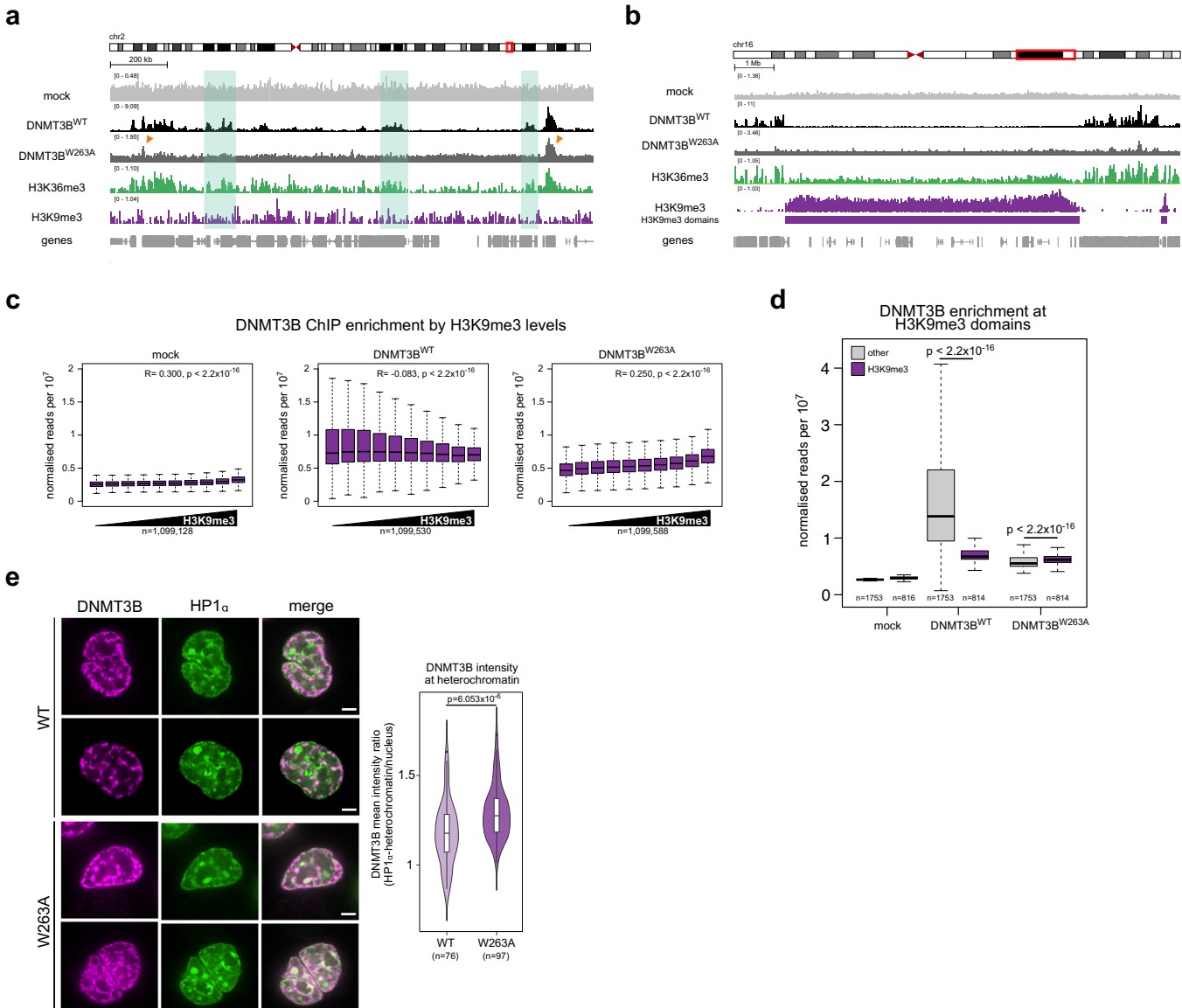

**Figure 3. PWWP mutation leads to increased DNMT3B localisation at heterochromatin.**

(A) DNMT3B^WT and DNMT3B^W263A binding profiles at a representative euchromatic genomic region. (B) DNMT3B^WT and DNMT3B^W263A binding profiles at a representative heterochromatic genomic region. Both (A, B) are genome browser plots showing T7-DNMT3B ChIP signal along with H3K36me3 and H3K9me3 ChIP-seq in DNMT3B KO cells, normalised reads per 10^6. (A) green rectangles highlight DNMT3B^W263A loss from several H3K36me3-marked regions; orange arrows indicate remaining DNMT3B^W263A peaks at some H3K36me3 loci. (C, D) DNMT3B^W263A has increased localisation at H3K9me3 domains. (C) T7 ChIP-seq counts at 2.5 kb genomic windows were ranked by increased H3K9me3 levels measured in DNMT3B KO cells before being divided into deciles of equal size based on rank. Pearson's correlations (R) and associated p values are shown. n is the number of genomic windows analysed in each case (1,099,128, 1,099,530 and 1,099,588 for mock, DNMT3B^WT and DNMT3B^W263A, respectively). (D) Boxplot showing T7-DNMT3B enrichment at H3K9me3 domains (n = 816, 814 and 814 domains for mock, DNMT3BWT and DNMT3B^W263A respectively) and genomic regions marked by neither H3K9me3 nor H3K27me3 (other, n = 1753 domains for all). (E) DNMT3B^W263A localises to HP1α. Right, representative confocal images of SNAP-DNMT3B and HP1α-GFP localisation in live DNMT3B KO cells. Bars = 3 μm. Left, violin plot showing the distribution of DNMT3B mean intensity ratio between HP1α-marked heterochromatin and the rest of the nucleus from one biological replicate (n = 76 and 97 cells for DNMT3BWT and DNMT3B^W263A, respectively). Data from a second biological replicate are shown in Appendix Fig. S4e. Data information: For boxplots in (C, D) and violin plot in (E): lines = median; box = 25th–75th percentile; whiskers = 1.5 × interquartile range from box. Boxplot and violin plot P values in (D, E) are from two-sided Wilcoxon rank-sum tests. All histone and T7-DNMT3B ChIP-seq data shown are derived from the mean of two biological replicates. P values for all tests are shown in the figure panels. Source data are available online for this figure.

in DNMT3B^S270P cells was significantly lower than DNMT3B^WT cells consistent with destabilisation of DNMT3B by S270P (Appendix Fig. S5f). In addition, in vitro a purified S270P mutant construct containing the PWWP and ADD domains was far less

stable, as evidenced by the lower melting temperature of DNMT3B^S270P compared to DNMT3B^WT when assessed by thermal denaturation assays (34.7 and 49.9 °C, respectively, Appendix Fig. S5g, S5h).

These results suggest that the failure of DNMT3B[S270P] to methylate heterochromatin is caused by the destabilising effect of the mutation rather than its effect on H3K36me3 binding.

## DNMT3B's PWWP domain is dispensable for DNMT3B localisation to heterochromatin

Having observed that DNMT3B PWWP mutations affecting H3K36me3 binding result in increased localisation and methylation at H3K9me3 domains, we then sought to understand which parts of the protein might be responsible for recruiting DNMT3B to heterochromatin.

As well as methylated H3K36, PWWP domains are reported to bind DNA through positively charged surface residues (Eidahl et al, 2013; van Nuland et al, 2013). DNMT3B's PWWP domain has also been reported to directly bind DNA (Qiu et al, 2002). We therefore sought to understand whether DNA binding mediated through DNMT3B's PWWP might be responsible for its recruitment to heterochromatin.

Analysis of the structure of DNMT3's PWWP domain reveals two patches of positively charged residues adjacent to the H3K36me3-binding pocket that might be involved in DNA binding (Fig. 4A, see 'Methods' for details). These contain two highly conserved lysines, K276 and K294, and two less conserved basic charged residues, K251 and R252 (Fig. 4A). We tested the role of these residues in DNA binding using purified components by EMSA and found that mutating all four amino acids drastically reduced DNA binding (Fig. 4B; Appendix Fig. S6a). However, charge swapping mutations of Lys-276 or Lys-294 alone are sufficient to highly decrease binding to DNA (Fig. 4B; Appendix Fig. S6a).

We therefore expressed these single mutations in DNMT3B KO cells (DNMT3B[K276E] and DNMT3B[K294E] cells, respectively, Appendix Fig. S6b) and compared DNA methylation at satellite II sequences to DNMT3B[WT] and DNMT3B[W263A] cells by methylation-sensitive Southern blot. Both DNMT3B[K276E] and DNMT3B[K294E] cells exhibited protection from digestion at a similar level to that observed for DNMT3B[W263A] cells (Fig. 4C). Using bisulfite PCR we found that methylation levels at two non-repetitive H3K9me3 loci were significantly higher in DNMT3B[K276E] or DNMT3B[K294E] cells compared to DNMT3B[WT] cells (Fig. 4D). These results suggest that these two mutations, that affect DNA binding, result in increased localisation of DNMT3B to heterochromatin in a manner similar to mutations affecting the interaction with H3K36me3. Expression of DNMT3B containing these lysine mutants alongside the W263A mutation (DNMT3B[W263A K276E] and DNMT3B[W263A K276E] cells) resulted in higher methylation levels than the single lysine mutations at both satellite II and H3K9me3 loci (Fig. 4C,D).

To further understand the requirement for the PWWP domain in localising DNMT3B to heterochromatin, we expressed DNMT3B lacking the entire PWWP domain (DNMT3B[ΔPWWP]). DNMT3B[ΔPWWP] levels were lower than DNMT3B by western blot (Appendix Fig. S7a). Despite this lower level of protein, methylation-sensitive Southern blot showed that DNMT3B[ΔPWWP] satellite II was protected from digestion to a greater degree than in DNMT3B[WT] cells (Fig. 4C). We also observed significantly greater levels of DNA methylation at two non-repetitive H3K9me3 loci in DNMT3B[ΔPWWP] cells than in DNMT3B[WT] cells (Fig. 4D).

Overall, these results suggest that interference with the DNA or H3K36me3-binding capacity of DNMT3B's PWWP domain results in hypermethylation of heterochromatin and that the PWWP domain itself is dispensable for recruitment of DNMT3B to heterochromatin.

## The N-terminal region facilitates methylation of heterochromatin by DNMT3B

Recently, the N-terminal region of DNMT3A has been shown to interact with H2AK119ub (Gu et al, 2022; Weinberg et al, 2021) and recruit the protein to H3K27me3-marked DNA methylation valleys, particularly in the context of PWWP domain mutations (Heyn et al, 2019; Manzo et al, 2017; Weinberg et al, 2021). We therefore wondered whether the N-terminal region of DNMT3B might be involved in its recruitment to H3K9me3-marked heterochromatin, particularly given that this is the most divergent region between DNMT3A and DNMT3B (Manzo et al, 2017).

We expressed DNMT3B lacking the N-terminal region in DNMT3B KO cells (lacking residues 1–199, DNMT3B[ΔN] cells, Appendix Fig. S7a,b) and assayed their methylation profile by WGBS (Fig. 5A). Methylation levels at H3K9me3 domains in DNMT3B[ΔN] cells were significantly greater than in DNMT3B KO cells, but significantly lower than DNMT3B[WT] cells in two independently generated cell lines (mean of two replicates in Fig. 5A,B; Appendix Fig. S7c). Levels of DNA methylation at other domains were not significantly different between DNMT3B[ΔN] and DNMT3B[WT] cells (Fig. 5B). Methylation-sensitive southern blot revealed that methylation levels at satellite II repeats in DNMT3B[ΔN] cells were lower than in DNMT3B[WT] cells, as seen by increased digestion due to higher levels of hypomethylated DNA (Appendix Fig. S7d). Significantly lower levels of DNA methylation were also observed at two non-repetitive H3K9me3 loci in DNMT3B[ΔN] cells compared to DNMT3B[WT] cells (Appendix Fig. S7e). Similarly, cells expressing a doubly mutated DNMT3B[W263A] lacking the N-terminal region (DNMT3B[ΔN W263A] cells) showed less methylation at satellite II (Appendix Fig. S7e) and at the two selected non-repetitive H3K9me3 loci (Appendix Fig. S7e) compared to DNMT3B[W263A] cells. These data suggest that the N-terminal region of DNMT3B facilitates its activity at heterochromatin.

We then assessed DNMT3B[ΔN] localisation on chromatin using ChIP-seq. DNMT3B[ΔN] signal was significantly correlated with H3K36me3 levels across the genome, similarly to DNMT3B[WT] (Appendix Fig. S7f,g). However, DNMT3B[ΔN] signal had a stronger negative correlation with H3K9me3 than DNMT3B[WT] (Pearson's correlations, $R = -0.148$ and $-0.083$, respectively, both $P < 2.2 \times 10^{-16}$, Fig. 5C), consistent with deletion of the N-terminus disproportionately affecting DNMT3B localisation to H3K9me3-marked heterochromatin.

Using the biscistronic fluorescent protein stability assay, we observed that the ratio of GFP-DNMT3B[ΔN] to dsRed was similar to that of GFP-DNMT3B[WT] protein (Appendix Fig. S7h), suggesting that removal of the N-terminus did not affect the stability of the protein. However, different expression levels of DNMT3B[ΔN] and DNMT3B[WT] in cell populations could account for the lower DNA methylation levels at heterochromatin in DNMT3B[ΔN] cells. To independently test the role of DNMT3B's N-terminal region in recruiting it to heterochromatin, we analysed the localisation of SNAP-tagged DNMT3B[ΔN] in live cells using confocal microscopy. The ratio of DNMT3B signal at HP1α -marked heterochromatin versus the rest of the nucleus was significantly lower for

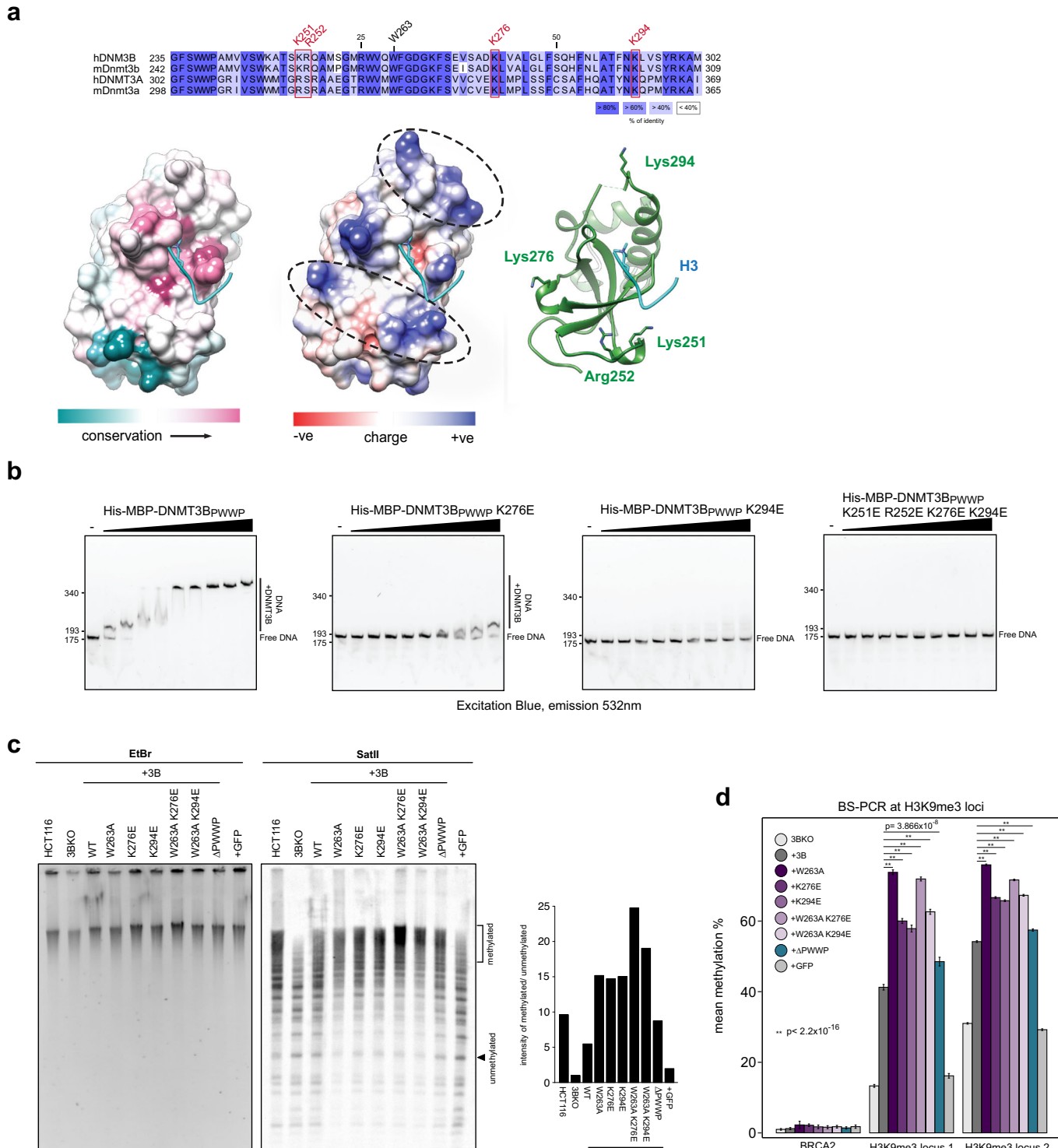

DNMT3B$^{\Delta N}$ than for DNMT3B$^{WT}$ (Fig. 5D; Appendix Fig. S7i), supporting the hypothesis that removal of the N-terminal region affects the recruitment of DNMT3B to heterochromatin.

To confirm the N-terminal region contributes to DNMT3B interaction with chromatin we performed fluorescence recovery after photobleaching (FRAP) in cells expressing SNAP-DNMT3B$^{WT}$ or DNMT3B$^{\Delta N}$ (Fig. 5E, top). Over the time course, DNMT3B$^{\Delta N}$

showed higher recovery over the time course (Fig. 5E, bottom left) and a significantly smaller immobile fraction (Fig. 5E, bottom right) than DNMT3B$^{WT}$. This is consistent with the proportion of DNMT3B$^{\Delta N}$ with slow dynamics in cells being lower than for DNMT3B$^{WT}$. It suggests that less of DNMT3B$^{\Delta N}$ is bound to chromatin than DNMT3B$^{WT}$ and supports a role for the N-terminus in stabilising DNMT3B's association with chromatin.

**Figure 4. DNMT3B PWWP binding to DNA is dispensable for localisation to heterochromatin.**

(A, B) Positively charged residues on the surface of DNMT3B's PWWP bind DNA. (A) Sequence of the DNMT3B PWWP domain alongside its structure (PDB: 5CIU) (Rondelet et al, 2016). Surface representation coloured according to amino-acid sequence conservation calculated by ConSurf (left) (Ashkenazy et al, 2016), or columbic surface charge (middle). Dashed ovals indicate clusters of positive charge. Ribbon model of domain with putative DNA-binding basic residues mutated in this study highlighted (right). (B) Representative gel images from electrophoretic mobility shift assays using increasing concentrations of tagged DNMT3B PWWP domains containing mutations with fluorescently labelled DNA. (C, D) DNMT3B PWWP mutants affecting DNA-binding remethylate heterochromatin to a greater extent than WT protein. (C) Methylation-sensitive Southern blot showing digestion of satellite II sequences in DNMT3B KO cells expressing DNMT3B mutants (centre). Ethidium bromide-stained gel (EtBr) is shown as a loading control (left). Barplot shows signal quantification of satellite II Southern blot using the ratio of the methylated over unmethylated regions indicated (right). (D) Mean methylation by BS-PCR at H3K9me3 loci alongside the H3K4me3-marked BRCA2 promoter in DNMT3B mutant cells. $P$ values are from two-sided Wilcoxon rank-sum tests. The number of reads analysed per each sample are shown in Appendix Table S5 ($n$ = number of reads). The error bars show the standard error in the mean. $P$ values for all tests are shown in the figure panel. Source data are available online for this figure.

Taken together, these data suggest that the N-terminal region of DNMT3B facilitates the PWWP-independent recruitment of DNMT3B to constitutive H3K9me3-marked heterochromatin.

## DNMT3B's N-terminal region interacts with HP1α

Having observed that DNMT3B methylates and localises to H3K9me3-marked heterochromatin, we sought to understand the molecular nature of this recruitment.

We first asked whether DNMT3B directly interacts with H3K9me3. The ATRX ADD domain can bind H3K9me3 (Eustermann et al, 2011; Iwase et al, 2011). However, the residues responsible are not conserved in DNMT3B's ADD domain (Iwase et al, 2011). Using a fluorescence-based peptide assay, we found that a purified fragment containing the full chromatin interaction modules of the DNMT3B (DNMT3B N-PWWP-ADD, Appendix Fig. S8a) does not show preferential binding to H3K9me3 over unmodified H3 N-terminal peptides (Appendix Fig. S8b). Furthermore, no preference was observed using in vitro pulldowns of recombinant unmodified and H3Kc9me3-modified nucleosomes (Appendix Fig. S8c) with immobilised DNMT3B constructs (Appendix Fig. S8d). Purified N-terminal region-PWWP-ADD DNMT3B pulled down modified and H3Kc9me3-modified nucleosomes to an equivalent extent (Appendix Fig. S8d). However, removal of the ADD did not alter this interaction and the ADD domain alone displayed an overall weaker interaction and a similar lack of preference for H3Kc9me3-modified nucleosomes (Appendix Fig. S8d). This suggests that while DNMT3B can interact directly with nucleosomes and tolerate H3K9me3, there does not appear to be strong preference for H3K9me3 that might explain our observations in cells.

We next investigated if DNMT3B's interaction with H3K9me3-marked heterochromatin could be mediated indirectly. Previous studies have reported that DNMT3B can interact with HP1α (Lehnertz et al, 2003), a marker of constitutive heterochromatin that binds H3K9me3 (Jacobs and Khorasanizadeh, 2002; Nielsen et al, 2002). We have also observed colocalization of DNMT3B and HP1α in an N-terminal region-dependent manner (Fig. 5D). Given our observation that the N-terminal region of DNMT3B facilitates the methylation of H3K9me3-marked heterochromatin, we hypothesised that the N-terminal region mediates the interaction with HP1α. We therefore performed DNMT3B immunoprecipitations (IPs) from DNMT3B KO cells transiently expressing DNMT3B^WT and DNMT3B^ΔN using their T7 tag. IPs of DNMT3B^WT contained both HP1α and H3K9me3 (Fig. 6A). In contrast, we did not detect HP1α and H3K9me3 in IPs of DNMT3B^ΔN (Fig. 6A).

This suggests that the interaction between DNMT3B, HP1α and H3K9me3-modified nucleosomes is dependent on DNMT3B's N-terminal region.

To understand whether the interaction between DNMT3B and HP1α was direct, we then conducted in vitro pulldowns of HP1α using His-MBP-tagged DNMT3B constructs (Appendix Fig. S8a). This revealed that a construct consisting of the N-terminus, PWWP and ADD domain of DNMT3B was able to enrich purified HP1α (Fig. 6B). However, removal of the N-terminal region abrogated this interaction (Fig. 6B). In addition, a construct consisting of only the N-terminal region was able to pull-down HP1α, but this interaction was absent when either the PWWP or ADD domain alone were used (Fig. 6B). This suggests that the N-terminal region of DNMT3B is both necessary and sufficient for direct interaction with HP1α.

We next wanted to understand whether HP1α could bridge the interaction of DNMT3B with H3K9me3-marked nucleosomes, stimulating the overall affinity of DNMT3B for H3K9me3-marked heterochromatin that we observe in cells. Using an EMSA assay with a purified N-PWWP DNMT3B fragment, we found that DNMT3B interacted with fluorescently labelled H3Kc9me3 nucleosomes in a concentration-dependent manner (Fig. 6C). The addition of purified HP1α, greatly increased the affinity of this interaction (Fig. 6C), demonstrating that HP1α stimulates the interaction of DNMT3B with H3K9me3-modified nucleosomes in vitro.

Taken together, these experiments suggest that HP1α interacts directly with the N-terminal region of DNMT3B and facilitates the interaction of DNMT3B with H3K9me3-modified nucleosomes.

## Discussion

Here, we have analysed *DNMT3B* mutations and knockout human cells to show that mutations in DNMT3B's PWWP domain result in hypermethylation of H3K9me3-marked heterochromatin loci. Our results suggest that DNMT3B is recruited to and methylates heterochromatin in a PWWP-independent manner that is facilitated by its N-terminal region through interaction with HP1α. We propose that the hypermethylation of heterochromatin observed with PWWP mutations results from the redistribution of DNMT3B away from H3K36me3-marked loci (Fig. 7).

Previous work has shown that the majority of DNMT3B localises to actively transcribed gene bodies due to the interaction of the PWWP domain with H3K36me3 (Baubec et al, 2015) and that losses of methylation in *Dnmt3b* knockout mouse embryonic stem cells correlate with H3K36me3 (Neri et al, 2017). However,

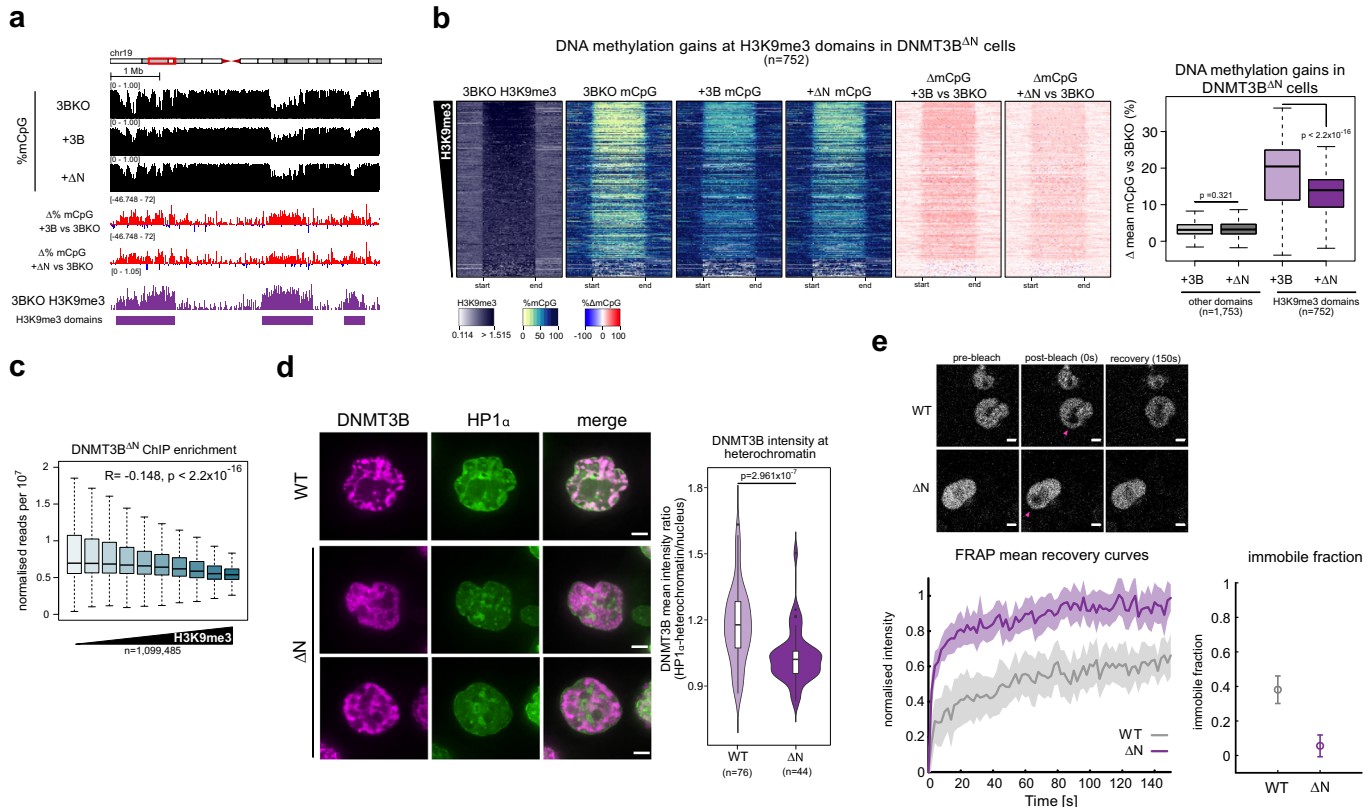

**Figure 5. The N-terminus facilitates methylation of heterochromatin by DNMT3B.**

(A, B) Lower DNA methylation recovery at heterochromatin upon expression of DNMT3B lacking the N-terminal region compared to full-length DNMT3B. (A) Representative genomic location showing gains of DNA methylation at H3K9me3 domains in DNMT3B KO cells expressing DNMT3B or DNMT3B$^{\Delta N}$. Genome browser plots show absolute (black) and differential (gain=red, loss=blue) DNA methylation levels, DNMT3B KO ChIP-seq signals and H3K9me3 domains defined in DNMT3B KO cells. ChIP-seq are normalised reads per $10^6$. (B) Left, heatmaps showing levels of H3K9me3, of absolute and differential DNA methylation at DNMT3B KO H3K9me3 domains ($n = 752$ domains) in DNMT3B$^{WT}$ or DNMT3B$^{\Delta N}$ cells. The domains are ranked by their mean H3K9me3 levels. Right, boxplot showing gains of DNA methylation at H3K9me3 domains ($n = 752$ domains) or domains not marked by neither H3K9me3 nor H3K27me3 ($n = 1753$) for DNMT3B$^{WT}$ and DNMT3B$^{\Delta N}$ cells. All measurements for DNMT3B$^{\Delta N}$ are mean of two independently generated cell lines. H3K9me3 ChIP-seq data are repeated from Fig. 2C as they are the same experiment. (C, D) Decreased DNMT3B localisation at heterochromatin in the absence of the N-terminal region. (C) Boxplot showing T7-DNMT3B$^{\Delta N}$ enrichment normalised over input at 2.5 kb genomic windows ($n = 1,099,485$ windows) of increased H3K9me3 enrichment in DNMT3B KO cells grouped into deciles of equal size based on rank. Pearson's correlation coefficient ($R$) is shown alongside its associated $P$ value. (D) Left, representative confocal images of SNAP-DNMT3B and GFP-HP1α localisation in live DNMT3B KO cells. Bars= 3 µm. Right, violin plot showing the distribution of DNMT3B mean intensity ratio between HP1α-marked heterochromatin and the rest of the nucleus from one biological replicate experiment. The second biological replicate is shown in Appendix Fig. S7i. $P$ value is from two-sided Wilcoxon rank-sum test. DNMT3B$^{WT}$ data are repeated from Fig. 3E as the data were part of the same experiment. (E) Top, representative images from a FRAP experiment (pink arrow indicates the side of the bleached area). Bottom left, mean fluorescence recovery curves for DNMT3B$^{WT}$ and DNMT3B$^{\Delta N}$ from three biological replicates of the FRAP experiment. The propagated error is shown by the shaded area. Bottom right, mean immobile fraction of DNMT3B$^{WT}$ and DNMT3B$^{\Delta N}$ calculated from 100 to 150 s for three biological replicates. Bars represent the propagated error from the standard deviations. Data information: For boxplot in (B, C) and violon plot in (D): lines = median; box = 25th–75th percentile; whiskers = 1.5 × interquartile range from box and $P$ values are from two-sided Wilcoxon rank-sum tests. All histone and T7-DNMT3B ChIP-seq data shown are derived from the mean of two biological replicates. WGBS for DNM3B$^{WT}$ and DNMT3B$^{\Delta N}$ cells in (A, B) are the mean of two biological replicates. These are shown separately in Appendix Fig. S7c. $P$ values for all tests are shown in the figure panels. Source data are available online for this figure.

loss of DNMT3B has also been shown to result in hypomethylation of heterochromatic satellite loci in mouse (Okano et al, 1999) and human cells (Jeanpierre et al, 1993; Liao et al, 2015) and DNMT3B localises to heterochromatin blocks in interphase cells in mouse embryonic stem cells and other cell types (Chen et al, 2004; Ge et al, 2004). In these studies, mutations of DNMT3B's PWWP domain were reported to abrogate the localisation of DNMT3B to these heterochromatic domains, but the effect of the mutations on protein stability was not tested. We instead show that multiple, stable mutations that are predicted to affect the interaction of the PWWP domain with H3K36me3 or DNA result in increased activity of DNMT3B at heterochromatin.

Our results parallel gains of methylation at DNA methylation valleys reported for PWWP mutations affecting the interaction of DNMT3A with methylated H3K36 in HESJAS patients (Heyn et al, 2019) and a mouse model (Sendžikaitė et al, 2019). Recent papers have proposed that this relocalisation is due to the recognition of the Polycomb Repressive Complex 1-associated histone mark H2AK119ub by DNMT3A's N-terminus (Gu et al, 2022; Weinberg et al, 2021). Here we show that the N-terminal region of DNMT3B directly interacts with HP1α and that HP1α increases the affinity of DNMT3B's interaction with H3K9me3-modified nucleosomes in vitro. DNMT3B has previously been suggested to interact with HP1α and is recruited to pericentromeric repeats in mouse

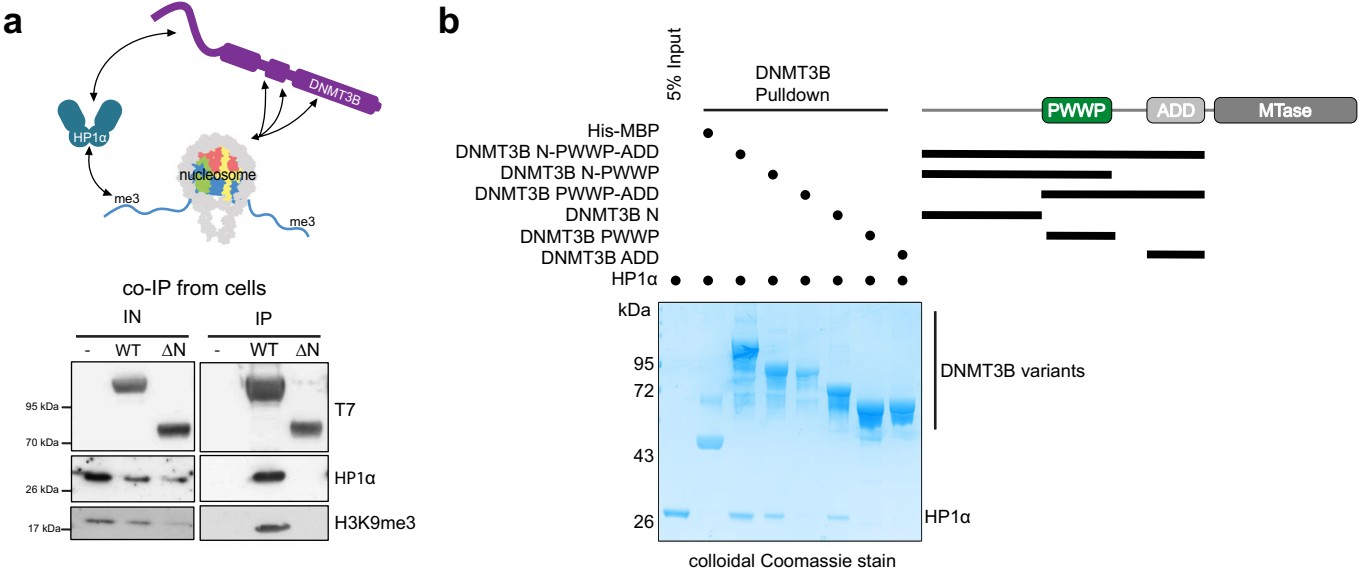

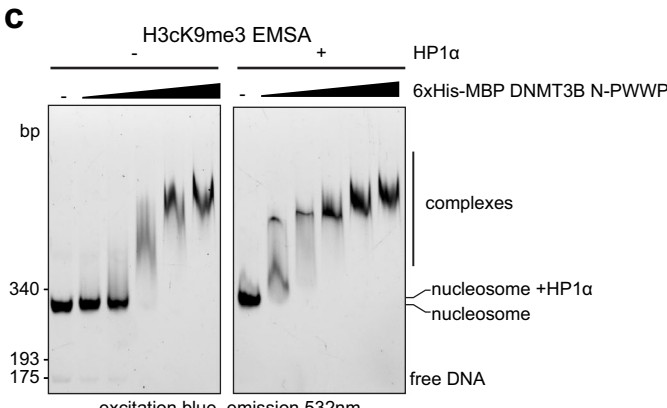

**Figure 6.  HP1α interacts with DNMT3B's N-terminal region.**

(A) DNMT3B binds to HP1α in cells. Western blots following immunoprecipitation of DNMT3B^WT or DNMT3B^ΔN from DNMT3B KO cells. HP1α and H3K9me3-modified nucleosomes immunoprecipitate with DNMT3B^WT but not DNMT3B^ΔN. (B) Binding of DNMT3B to HP1α is direct and mediated by the N-terminal region of DNMT3B. The coomassie-stained gel from pull-down assay using His-MBP-tagged DNMT3B constructs immobilised on Ni-NTA agarose beads and incubated with HP1α. All constructs 6xHis-MBP tagged, residues: N-PWWP-ADD = 1–554, N-PWWP = 1–351, PWWP-ADD = 213–554, ADD = 412–554 PWWP = 213–351, *N* = 1–205. His-MBP is used as a negative control. (C) HP1α increases DNMT3B binding to H3Kc9me3 nucleosomes in vitro. Native-PAGE gel imaged using fluorescein filters of electrophoretic mobility shift assay (EMSA) of fluorescent H3Kc9me3 nucleosomes incubated with increasing concentrations of 6xHis-MBP-DNMT3B 1^-351 N-PWWP with the left-hand panel in the absence of HP1α and the right-hand side pre-bound with saturating presence of 1 μM HP1α. Source data are available online for this figure.

embryonic stem cells in a Suv39h dependent manner (Lehnertz et al, 2003). However, this study did not define the nature of the interaction with HP1α or its role in recruitment to pericentromeric repeats. A previous report has suggested that the N-terminal region of DNMT3B is involved in nucleosomal recruitment (Jeong et al, 2009) but did not ask how the requirement for this part of the protein varies in different genomic compartments. Further work will be required to dissect the precise role that the N-terminal region and interaction with HP1α play in DNMT3B recruitment and whether DNMT3B also interacts with the other 2 HP1 paralogues found in humans (Jones et al, 2000).

We note that here we have not directly demonstrated that the mutations studied affect the interaction between DNMT3B's

PWWP and H3K36me3. It is also possible that the mutations may have other effects. For example, a recent study suggests that the HESJAS-causing mutation W330R (Heyn et al, 2019) may also mis-localise DNMT3A by promoting DNA binding (Lue et al, 2023). This is unlikely to explain our results with DNMT3B, as while we have analysed a mutation of a paralogous residue (W263A), the change to alanine conserves overall charge in the aromatic cage. Furthermore, we observe similar results with mutation of another key residues within of the PWWP domain's aromatic cage, D266 (Qin and Min, 2014; Rondelet et al, 2016). We also see similar effects upon the removal of the PWWP domain. While some of the mutations we studied do not directly affect DNMT3B's interaction with H3K36me3, it has previously been

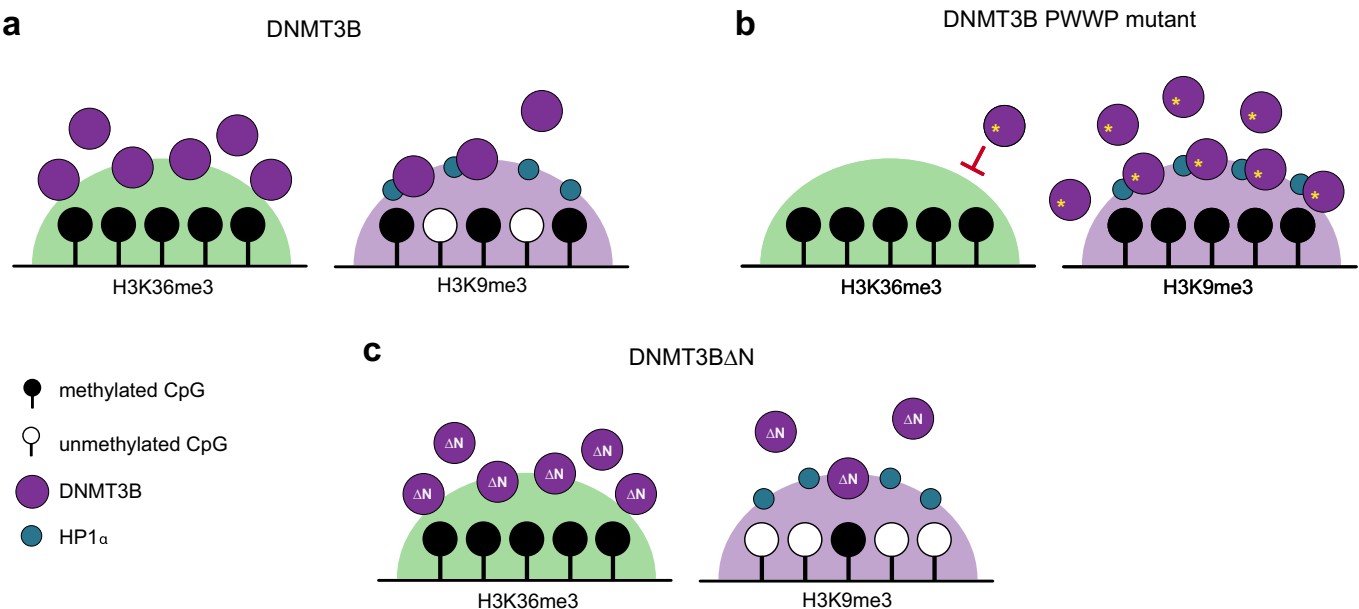

**Figure 7. DNMT3B PWWP mutations cause hypermethylation of heterochromatin.**

Normally, DNMT3B predominantly localises to H3K36me3-marked regions through the action of its PWWP domain (Baubec et al, 2015). A small proportion of DNMT3B is also localised to H3K9me3-marked heterochromatin. When DNMT3B's PWWP is mutated, we propose binding to H3K36me3-marked regions is inhibited and an increased pool of DNMT3B localises to H3K9me3-marked heterochromatin in a manner that is facilitated by its N-terminal region which interacts with HP1α. The relocalisation of DNMT3B results in gains of DNA methylation in this genomic compartment. (A–C) Schematics depicting distribution of DNMT3B, PWWP-mutated DNMT3B and N-terminal truncated DNMT3B respectively.

suggested that the high-affinity association of PWWP domains with chromatin requires cooperative DNA and histone-mark binding (Eidahl et al, 2013; van Nuland et al, 2013; Wang et al, 2020). It is also possible that other parts of DNMT3B play a role in its action at heterochromatin. For example, a recent study suggests that the ADD domain of DNMT3B behaves differently to that of DNMT3A (Boyko et al, 2022). Future work will be required to dissect the combined effect of the multiple domains in the recruitment of DNMT3B to chromatin.

Here we observe that the enrichment of DNMT3B at H3K9me3-marked heterochromatin by ChIP-seq is less distinct than at H3K36me3 but that heterochromatic localisation is more apparent by microscopy. Previous work has suggested that formaldehyde cross-linking fails to capture transient interactions in cells (Schmiedeberg et al, 2009). Such an effect could affect our ability to detect DNMT3B at H3K9me3 by ChIP-seq. H3K9me3-marked heterochromatin is also resistant to sonication (Becker et al, 2017) and this could decrease DNMT3B ChIP-seq signal at H3K9me3. Furthermore, our data also suggests that DNMT3B's interaction with H3K9me3-modified nucleosomes is indirect and mediated by HP1α. This may also lead to weaker ChIP-seq enrichment at H3K9me3 compared to the direct interaction with H3K36me3 through DNMT3B's PWWP domain.

Loss of DNA methylation from heterochromatic loci is a hallmark of cancer that results in the formation of PMDs (Berman et al, 2011; Zhou et al, 2018) and is thought to lead to activation of retrotransposons and genome instability (Du et al, 2019; Zhou et al, 2018). Our observation that DNMT3B plays a role in DNA methylation homeostasis in PMDs suggest that modulation of its activity could play a role in this

hypomethylation. Our results here and those of our previous study of CpG islands (Masalmeh et al, 2021) suggest that, similarly to normal cells, the majority of DNMT3B in cancer cells localises and is active at H3K36me3-marked regions. However, here we show that this balance of DNMT3B activity in cells can be shifted by mutation or removal of the PWWP domain to cause increased methylation at H3K9me3-marked regions. Many different DNMT3B splice isoforms have been reported to be aberrantly expressed in cancer (Gopalakrishnan et al, 2009; Ostler et al, 2007). These include ΔDNMT3B4 which lacks the PWWP domain and a portion of the N-terminal region (Wang et al, 2006b) and is reported to be expressed in non-small cell lung cancer (Wang et al, 2006a). It is also possible that this balance between recruitment of DNMT3B to H3K9me3 and H3K36me3 is regulated by cellular signalling pathways which could be altered in carcinogenesis.

Recessive mutations of DNMT3B cause ICF1 syndrome (Weemaes et al, 2013), a Mendelian condition characterised by hypomethylated heterochromatin (Heyn et al, 2012; Jeanpierre et al, 1993). The majority of mutations truncate DNMT3B or reduce its catalytic activity (Weemaes et al, 2013). A recent study also suggested that some ICF1 mutations can affect the formation of DNMT3B homo-oligomers (Gao et al, 2022). However, two patients have been described as carrying the S270P mutation which lies within the PWWP domain (Shirohzu et al, 2002). This mutation has been shown to reduce the binding of the domain to H3K36me3 (Baubec et al, 2015). Differences in methylation in gene bodies marked by H3K36me3 have also been proposed to lead to altered splicing in ICF1 patient cells (Gatto et al, 2017). Our results show that S270P destabilises DNMT3B in vitro and in vivo

explaining why it does not cause hypermethylation of heterochromatin like the other PWWP mutations we have analysed. This suggests that the pathogenic nature of S270P in causing ICF1 syndrome is due to the reduction in DNMT3B levels in cells rather than its effect on association with H3K36me3. However, we note that HCT116 are a colorectal cancer cell line and thus it is possible that the mutation behaves differently in the lymphocytes which are the primary cell type affected by ICF1 (Weemaes et al, 2013).

In conclusion, we report of an unexpected consequence of DNMT3B PWWP mutations which reveal that DNMT3B plays a role in DNA methylation homoeostasis at heterochromatin, a genomic compartment whose DNA methylation levels are frequently altered in cancer and other human diseases.

# Methods

## Cell culture

HCT116 and DNMT3B KO cells were gifts from B. Vogelstein (Rhee et al, 2002). Cells were cultured in McCoy's 5A medium (Gibco). RKO cells were cultured in Dulbecco's Modified Eagles Medium (Sigma-Aldrich). Both were supplemented with 10% foetal calf serum (Life Technologies) and penicillin–streptomycin antibiotics at 140 and 400 µg/ml, respectively. Cell lines were routinely tested for mycoplasma contamination. Knockout of DNMT3B was confirmed by sequencing and western blot.

## Generation of CRISPR/Cas9-edited HCT116 cell lines

To introduce the W263A mutation in endogenous *DNMT3B* in HCT116 cell, the CRISPR Targets track from the USCS genome browser (https://genome.ucsc.edu) was used for guide RNA design and the corresponding oligonucleotides cloned into pSpCas9(BB)-2A-GFP (pX458, Addgene Plasmid 48138, a gift of F. Zhang). The donor template for homology-directed repair was an 80 nt ss-oligo Alt-R HDR Donor Oligo (Integrated DNA Technologies) designed on exon 7 and carrying the W263A mutation. HCT116 cells were transfected with the vector and ss-oligo, and GFP+ve cells selected by FACS 48 h after transfection and plated at clonal density. Individual colonies were screened by PCR followed by Sanger sequencing. Primers and oligonucleotides used are listed in Appendix Table S1.

## Generation of plasmid constructs for expression in cells

To create piggyBac DNMT3 expression vectors, DNMT3 sequences from pcDNA3-Myc-DNMT3B2 and pcDNA3-Myc-DNMT3A1 plasmids (Addgene plasmids 36942 and 35521, a gift from A. Riggs) (Chen et al, 2005) were initially subcloned into the pCG plasmid (a gift from N. Gilbert) downstream of the T7 tag. The catalytically inactive point mutations C631S (DNMT3B2) and C710S (DNMT3A1) and the point mutations in the PWWP domain of DNMT3B (W263A, D266A, S270P, K276E and K294E) were introduced using the QuikChange II site-directed mutagenesis kit (Agilent). DNMT3B$^{\Delta N}$ starts from M200 and was generated by PCR. DNMT3B$^{\Delta PWWP}$ was synthesised using gBlocks (Integrated DNA Technologies) and generates a DNMT3B truncated from

E206 to Y375. T7-tagged DNMT3s were then cloned into PB-CGIP (a gift from M. McCrew) (Macdonald et al, 2012) by swapping eGFP or into pB503A-puroVal2 (to generate the EF1α expression vector). To create vectors for live-cell imaging a SNAP-tag was initially subcloned from pCS2-SNAP (a gift from D. Papadopolous) to pB530A-puroVal2, previously created by substituting the copepod GFP from the pB530A plasmid (System Biosciences) with the Puromycin resistance gene. DNMT3B and mutants sequences were cloned downstream the SNAP-tag. To generate vectors for the protein stability assay DNMT3B and mutant sequences were cloned into pLenti-DsRed-IRES-EGFP (Addgene plasmid 92194, a gift from Huda Zoghbi).

## DNMT3 expression in cells

HCT116 and DNMT3B KO cells were transfected with FuGENE HD transfection reagent (Promega). For stable integrants, DNMT3 expression constructs were co-transfected with a plasmid expressing piggyBac transposase. After 48 h, cells stably expressing DNMT3s were selected with 1 µg/ml puromycin and expanded in the presence of puromycin for 3–4 weeks before being harvested for analysis.

## DNA extraction

To extract genomic DNA, cells were resuspended in genomic lysis buffer (300 mM NaCl, 1% SDS, 20 mM EDTA) and incubated with Proteinase K (Roche) at 55 °C overnight. RNA was removed by incubation with RNase A/T1 Cocktail (Ambion) at 37 °C for 1 h, in between two phenol–chloroform extraction steps. DNA was quantified by Nanodrop 8000 spectrophotometer and purity was assessed by electrophoresis.

## Global measurement of DNA methylation by mass spectrometry

DNA for mass spectrometry and following analysis was performed as previously described (Masalmeh et al, 2021). In all, 1 µg genomic DNA was denatured at 95 °C for 10 min in 17.5 µl water. DNA was then digested to nucleotides overnight at 37 °C with T7 DNA polymerase (Thermo Scientific). The reaction was inactivated by incubating at 75 °C for 10 min. Samples were then centrifuged for 45 min at >12,000 × $g$ and the supernatant was transferred into new tubes for analysis. The enzyme was removed by solvent precipitation. The samples were adjusted back to initial aqueous condition and volume and LC-MS was performed on a Thermo Ultimate 3000/ Thermo Q Exactive system, using a Hypercarb 3 µm × 1 mm × 30 mm Column (Thermo 35003-031030) and gradient from 20 mM ammonium carbonate to 2 mM ammonium carbonate 90% acetonitrile in 5 min. Data were acquired in negative mode, scanning at 70 k resolution from 300 to 350 $m/z$. Extracted ion chromatograms were analysed using Xcalibur (Thermo Scientific, v2.5-204201/2.5.0.2042) to extract peak intensities at the $m/z$ values expected for each nucleotide (based on annotation from METLIN) (Smith et al, 2005) following manual inspection to ensure that they were resolved as clear single peaks. The percentage of 5-methylcytosine present in the sample was calculated as the ratio of the area under the 5-methylcytosine peak to the area under the guanine peak.

## WGBS data generation

Overall, 100 or 200 ng of purified DNA samples were sheared using the Covaris E220 Evolution Focused Ultrasonicator to create fragments of ~350 bp. In total, 0.5 ng of unmethylated phage-λ DNA (NEB) was spiked into each DNA sample prior to shearing to allow assessment of the efficiency of the bisulfite-conversion reaction. In total, 100 ng of each sheared DNA sample was then processed using the EZ DNA Methylation-Gold Kit or EZ DNA Methylation-Lightning Kit (Zymo Research) according to the manufacturer's protocol to create bisulfite-converted single-stranded (BC-ss) DNA. For WGBS experiments in Fig. 1 and S1 comparing DNMT3B KO cells and reintroduction of DNMT3B, the TruSeq DNA Methylation Kit (Illumina Inc.) was used. Bisulfite-converted single-stranded DNA was randomly primed with a polymerase able to read uracil nucleotides to synthesise DNA strands containing a sequence-specific tag. 3' ends of the newly synthesised DNA strands were then selectively tagged with a second specific sequence tag to produce di-tagged DNA molecules with known sequence tags at 5' and 3' ends. Di-tagged molecules were then enriched by PCR with unique indexed PCR primers to provide dsDNA libraries with Illumina sequencing adapters that could be multiplexed and sequenced on a single flow cell. For all other WGBS experiments, the Accel-NGS Methyl-Seq DNA Library Kit (Swift BioSciences) was used. For the Swift kit, bisulfite-converted single-stranded DNA was first denatured before simultaneous tailing and ligation of truncated adapters to 3' ends with Adaptase. An extension step then incorporates the truncated adapter 1 by a primer extension reaction, before a ligation step adds truncated adapter 2 to the bottom strand only. Nine cycles of PCR were used to increase yield and incorporate full-length adapters for unique dual-indexed sequencing. Finally, Agencourt AMPure XP beads (Beckman Coulter) were used to remove oligonucleotides and small fragments and change enzymatic buffer composition. Libraries were quantified using the Qubit dsDNA HS assay kit, assessed for size and quality using the Agilent Bioanalyser with the DNA HS Kit and combined in a single equimolar pool. Sequencing was performed on a P2 flow cell on the Illumina NextSeq 2000 platform using NextSeq 1000/2000 P2 Reagents v3 kit (200 Cycles), or on the NextSeq 550 platform using NextSeq 500/550 High-Output v2 kit (150 cycles). PhiX Control v3 Library was spiked in at a concentration of ~5% to increase library diversity for sequencing. Library preparation and sequencing was performed by the Edinburgh Clinical Research Facility.

## WGBS data processing

Sequencing quality was assessed with FASTQC (*v0.11.4*). Low-quality reads and remaining adaptors were removed using TrimGalore (*v0.4.1*, Settings: *--adapter AGATCGGAAGAGC --adapter2 AAATCAAAAAAAC*). The paired-end reads were then aligned to the hg38 genome using Bismark (*v 0.18.1* with Bowtie2 *v2.3.1* and settings: *-N 0 -L 20*, trimming settings *--clip_r1 10 --clip_r2 13 --three_prime_clip_r1 1 --three_prime_clip_r2 1*) (Krueger and Andrews, 2011; Langmead and Salzberg, 2012) before PCR duplicates were identified and removed using Bismark's *deduplicate_bismark* command. Aligned BAM files were processed to report coverage and the number of methylated reads for each CpG observed. Forward and reverse strands were combined using

Bismark's *methylation extractor* and *bismark2bedgraph* modules with custom Python and AWK scripts. Processed WGBS files were assessed for conversion efficiency based on the proportion of methylated reads mapping to the phage-λ genome spike-in (>99.5% in all cases). For summary of WGBS alignment statistics, see Appendix Table S2. BigWigs for visualisation of WGBS data were generated using the coverage in 2.5 kb-sized windows across the genome. These were defined using BEDtools (*v2.27.1*) (Quinlan and Hall, 2010) and coverage and converted to bigWigs using UCSC tools *bedGraphToBigWig* (*v326*). Windows with a total coverage <5 were excluded before conversion to bigWigs.

## WGBS data analysis

WGBS data were analysed by defining genomic windows for analysis. For genome-wide analyses, non-overlapping 2.5 kb-sized genomic windows were defined using BEDtools *makewindows* (*v2.27.1*). The percentage methylation within these windows was calculated as the weighted mean methylation using the observed coverage of CpGs within each window (unconverted coverage/total coverage) using BEDtools. Windows with a total coverage <10 were excluded from the analysis.

To analyse methylation within domains, BEDtools was used to calculate the weighted mean coverage from CpGs located within the domain as defined above. To analyse the profile around domains, heatmaps were generated by defining 40 scaled windows across each domain and 20 windows of 25 kb upstream and downstream of each domain relative to the forward strand using custom R scripts. The weighted mean coverage in each window was then defined as above using BEDtools. Domains were excluded from analysis if ≥80% of the 40 scaled windows across the domain had a coverage of 0 in all samples in the analysis.

For the methylation profile across genes, a similar analysis was performed. Transcript locations were downloaded from ENSEMBL (*v106*). Only coding transcripts were considered (those defined as 'protein_coding' by ENSEMBL) and each gene's weighted mean methylation was defined from the mean of all coding transcripts annotated to it. To analyse the profile around genes, heatmaps were generated by defining 40 scaled windows across each coding transcript and 20 windows of 250 bp upstream and downstream of each coding transcript relative to its direction of transcription using custom R scripts. Before calculating the mean profile for each gene, transcripts were excluded if ≥80% of the 40 scaled windows across the domain had a coverage of 0 in all samples in the analysis.

To statistically test differences in DNA methylation levels, differential mean methylation percentages across domains were compared using a Wilcoxon rank-sum test.

## ChIP-seq data generation

ChIP-seq was performed as previously described (Masalmeh et al, 2021). For T7-DNMT3B ChIP-Rx-seq experiments, $1 \times 10^7$ cells were harvested, washed and cross-linked with 1% methanol-free formaldehyde in PBS for 8 min at room temperature. Cross-linked cells were lysed for 10 min on ice in 50 μl of lysis buffer (50 mM Tris-HCl pH 8, 150 mM NaCl, 1 mM EDTA, 1% SDS) freshly supplemented with proteinase inhibitor (Sigma-Aldrich). IP dilution buffer (20 mM Tris-HCl pH 8, 150 mM NaCl, 1 mM EDTA, 0.1% Triton X-100) freshly supplemented with proteinase inhibitor,

DTT and PMSF was added to the samples to reach a final volume of 500 μl. As prolonged sonication caused T7-DNMT3B degradation, chromatin was fragmented using Benzonase (Pchelintsev et al, 2016): samples were sonicated on ice with Soniprep 150 twice for 30 s to break up nuclei; then 200 U of Benzonase Nuclease (Sigma) and MgCl$_2$ (final concentration 2.5 mM) were added and samples were incubated on ice for 15 min. The reaction was blocked by adding 10 μl of 0.5 M EDTA pH 8. Following centrifugation for 30 min at $18,407 \times g$ at 4 °C, supernatants were collected and supplemented with Triton X-100 (final concentration 1%) and 5% input aliquots were retained for later use. Protein A Dynabeads (Invitrogen) previously coupled with 10 μl of T7-Tag antibody (D9E1X, Cell Signaling Technology, RRID: AB_2798161) per $1 \times 10^7$ cells in blocking solution (1× PBS, 0.5% BSA) were added and the samples incubated overnight under rotation at 4 °C. Beads were then washed for 10 min at 4 °C with the following buffers: IP dilution buffer 1% Triton X-100 (20 mM Tris-HCl pH 8, 150 mM NaCl, 2 mM EDTA, 1% Triton X-100), buffer A (50 mM HEPES pH 7.9, 500 mM NaCl, 1 mM EDTA, 1% Triton X-100, 0.1% Na-deoxycholate, 0.1% SDS), buffer B (20 mM Tris pH 8, 1 mM EDTA, 250 mM LiCl, 0.5% NP-40, 0.5% Na-deoxycholate), TE buffer (1 mM EDTA pH 8, 10 mM Tris pH 8). Chromatin was eluted by incubating the beads in extraction buffer (0.1 M NaHCO$_3$, 1% SDS) for 15 min at 37 °C. To reverse the cross-linking Tris-HCl pH 6.8 and NaCl were added to final concentrations of 130 mM and 300 mM, respectively, and immunoprecipitations were incubated at 65 °C overnight. Samples were then incubated at 37 °C for 1 h after addition of 2 μl of RNase Cocktail Enzyme Mix (Ambion). Then 40 μg of Proteinase K (Roche) were added, followed by 2-h incubation at 55 °C. Input material was similarly de-cross-linked. Samples were purified with the MinElute PCR purification kit (QIAGEN). For ChIP-Rx-seq of endogenous T7-DNMT3B and ectopic T7-DNMT3B expressed in DNMT3B KO cells, 20 μg of Spike-in chromatin (ActiveMotif 53083) was added to each sample after sonication. In total, 2 μl of spike-in antibody per sample (ActiveMotif 61686) was also added in a ratio 1:5 versus the T7 antibody. A similar protocol was used for H3K4me3, H3K9me3, H3K27me3 and H3K36me3 ChIP-seq experiments, except: $0.5 \times 10^7$ cells were harvested and cross-linked with 1% methanol-free formaldehyde in PBS for 5 min at room temperature. For H3K36me3 ChIP-Rx-seq cross-linked Drosophila S2 cells were spiked into samples before sonication at a ratio of 20:1 human to Drosophila cells. Following nuclei rupture by sonication on ice with Soniprep 150, chromatin was fragmented using Bioruptor Plus sonicator (Diagenode) for 40 cycles (30 s on/30 s off on high setting at 4 °C). In total, 2 μl/$1 \times 10^6$ cells of the following antibodies were used for immunoprecipitations: H3K4me3 (EpiCypher 13-00041), H3K9me3 (ActiveMotif 39161, RRID: AB_2532132), H3K27me3 (Cell Signaling Technology C36B11, RRID: AB_2616029) and H3K36me3 (Abcam ab9050, RRID: AB_306966). Libraries were prepared using the NEBNext Ultra II DNA Library Prep Kit for Illumina (E7645) according to the manufacturer's instructions. NEBNext Multiplex Oligos for Illumina (NEB) barcode adapters were used. Specifically, Illumina Index Primers Set 1 (E7335) for endogenous T7-DNMT3B and H3K36me3, and Unique Dual Index UMI Adaptors DNA Set 1 (E7395) for the other libraries. For histone modifications ChIP-seq, adapter-ligated DNA was size selected for an insert size of 150 bp using Agencourt AMPure XP beads. Libraries were quantified using the Qubit dsDNA HS or BR

assay kit and assessed for size and quality using the Agilent Bioanalyser. Endogenous T7-DNMT3B and H3K36me3 ChIP-Rx-seq libraries were sequenced using the NextSeq 500/550 high-output version 2.5 kit (75 bp paired-end reads, DNMT3B or 75-bp single-end reads, H3K36me3). The other ChIP-seq libraries were sequenced using the NextSeq 2000 P3 (50-bp paired-end reads). Libraries were combined into equimolar pools to run within individual flow cells. Sequencing was performed by the Edinburgh Clinical Research Facility.

## ChIP-seq data processing

All ChIP-seq experiments were processed as previously described (Masalmeh et al, 2021). Read quality was checked using FASTQC (*v0.11.4*, https://www.bioinformatics.babraham.ac.uk/projects/fastqc), with low-quality reads and adaptors removed using TrimGalore with default settings (*v0.4.1*). Reads were aligned to hg38 using bowtie 2 (*v2.3.1*, with settings: *-N 1 -L 20 --no-unal*) (Langmead and Salzberg, 2012). For paired-end data, additional settings were used during alignment to remove discordant reads: *--no-mixed –nodiscordant -X 1000*. ChIP-Rx-seq reads were aligned to a combination of hg38 and dm6 genomes. Multimapping reads excluded using SAMtools (*v1.6*, with settings: *-bq 10*) (Li et al, 2009) and PCR duplicates excluded using SAMBAMBA (*v0.5.9*) (Tarasov et al, 2015). For T7 ChIP-Rx-seq UMIs were used for deduplication instead of SAMBAMBA. We first extracted UMIs from the appropriate read and placed them in the read FASTQ headers using UMI tools *extract* (*v1.0.0* and setting: *--bc-pattern = NNNNNNNNNNNN*). Then deduplication was performed using the UMI tools *dedup* function (*v1.0.0* and setting: *--paired*). For summary of ChIP-seq alignment statistics, see Appendix Table S3.

Tracks for the visualisation of histone ChIP-seq were generated using Deeptools (*v3.2.0*) (Ramirez et al, 2016). Counts per million normalised tracks were generated using the *bamCoverage* function (settings: *--normalizeUsing CPM*) with the default bin size of 50 bp. The mean of replicate tracks and normalisation over the input was calculated using the *bigwigCompare* function (settings: *--operation mean*). For the single-ended H3K36me3 ChIP-seq, the estimated fragment length of 150 bp was used. For the paired-end knock-in T7-DNMT3B ChIP-Rx-seq and the ChIP-seq of other histone marks, the actual fragment size was used. Tracks for visualisation of T7 ChIP-Rx-seq were generated as follows: fragment counts in 2.5-kb-sized windows were generated using BEDtools, normalised over input and scaled according to the spike-in (see 'ChIP-seq data analysis' for details). The bedgraphs were then converted into bigwigs using UCSC tools *bedGraphToBigWig* (*v326*).

## ChIP-seq data analysis

For ChIP-seq data analyses, data were analysed by defining genomic windows for analysis as previously described (Masalmeh et al, 2021). Non-overlapping 2.5 kb-sized genomic windows were defined using BEDtools *makewindows* (*v2.27.1*). Histone modifications and T7-DNMT3B normalised coverage in these genomic windows or domains were derived from ChIP-seq by first counting the number of reads or fragments overlapping the windows/domains using BEDtools *coverage* function. For paired-end data, the BAM file was first converted to a BED file of fragment locations using BEDtools *bamtobed* function. Coverage counts were scaled to

counts per 10 million based on total number of mapped reads per sample and divided by the input read count to obtain a normalised read count. An offset of 0.5 was added to all windows prior to scaling and input normalisation to prevent windows with zero reads in the input sample generating a normalised count of infinity. Regions where coverage was 0 in all samples were removed from the analysis. We also excluded windows overlapping poorly mapped regions of the human genome (https://github.com/Boyle-Lab/Blacklist) (Amemiya et al, 2019) using BEDtools *intersect*. Gaps and centromeres identified using annotations from the UCSC browser (hg38 gap and centromere tracks) were similarly removed. First, annotations were downloaded from the UCSC table browser and regions annotated as heterochromatin, short arm and telomeres from the gaps track were merged with the centromeres track using BEDtools *merge* with -d set to 10 Mb. Windows overlapping this merged file were then excluded from the analysis using BEDtools *intersect*.

For ChIP-Rx-seq, before proceeding with the analysis, normalised coverage values were scaled using a scaling factor generated from the number of reads mapping to the *D.melanogaster* genome as previously described (Masalmeh et al, 2021). Reads mapping to the *D. melanogaster* genome in each ChIP and input sample were first scaled to reads per $1 \times 10^7$. The scaling factor was then calculated as the ratio of the scaled *D. melanogaster* reads in two ChIP samples over their respective ratio from the input samples (modified from the published method to take account of the presence of an input sample (Orlando et al, 2014)). This analysis was applied to each biological replicate and where multiple replicates were available, the mean was calculated for each window.

The profile around defined heterochromatic domains was similarly calculated but using 40 scaled windows across each domain and 20 windows of 25 kb upstream and downstream of each domain relative to the forward strand using custom R scripts. Scaled normalised coverage values were calculated as above for genomic windows except that domains were excluded from analysis if ≥80% of the 40 scaled windows across the domain had a coverage of 0 in all samples in the analysis. The mean profile was calculated from these values using R. For the ChIP-seq profile across genes, a similar analysis was performed. Transcript locations were downloaded from ENSEMBL (*v106*). Only coding transcripts were considered (those defined as 'protein_coding' by ENSEMBL) and each gene's scaled normalised coverage was calculated from the mean of all coding transcripts annotated to it. To analyse the profile around genes, heatmaps were generated by defining 40 scaled windows across each coding transcript and 20 windows of 250 bp upstream and downstream of each coding transcript relative to its direction of transcription using custom R scripts. Scaled normalised coverage values were calculated as above for genomic windows, except that transcripts were excluded from analysis if ≥80% of the 40 scaled windows across the domain had a coverage of 0 in all samples in the analysis. Colour scales for ChIP-seq heatmaps range from the minimum to the 90% quantile of the normalised read count.

## Definition and comparison of heterochromatic domains

H3K9me3 and H3K27me3 domains were called using a hidden Markov model as previously described (Spracklin and Pradhan, 2020). Normalised mean H3K9me3 and H3K27me3 ChIP-seq coverage in 25 kb genomic windows was analysed using the *bigwig_hmm.py* script

(https://github.com/gspracklin/hmm_bigwigs) to define two states (-n 2). We excluded poorly mapped regions of the genome from these domains using annotations of gaps and centromeres from the UCSC browser (hg38 gap and centromere tracks). Annotations were downloaded from the UCSC table browser. Regions annotated as heterochromatin, short arm and telomeres from the gaps track were merged with the centromeres track using BEDtools *merge* with -d set to 10 Mb. This merged file was then excluded from the domains BED files using BEDtools *subtract*. H3K27me3-only domains were then defined as those which did not overlap a H3K9me3 domain using BEDtools intersect.

Comparisons between heterochromatic domains were conducted using BEDtools *jaccard* and *fishers* modules. PMDs were defined in HCT116 cells using methpipe (*v5.0.0*) (Decato et al, 2020). WGBS BAM files generated by Bismark before being processed and deduplicated using methpipe. PMDs were then called using the *pmd* module from methpipe and the recommended settings (bin size was determined as 8 kb by methpipe using the setting -i 1000). We excluded poorly mapped regions of the genome from PMDs as for the H3K9me3 and H3K27me3 domains. We also excluded PMDs shorter than 200 kb. We downloaded a BED file of regions resistant to nuclease digestion in HCT116 cells from the 2-state definitions in Gene Expression Omnibus accession GSE135580 (Spracklin and Pradhan, 2020).

## RNA-seq data generation

RNA extraction was performed using RNeasy Plus Mini Kit (QIAGEN) following the manufacturer's instructions. Quantity and quality of RNA samples were assessed using the Nanodrop spectrophotometer (Nanodrop ND-1000, Thermo Scientific) as well as the Qubit fluorometer (Invitrogen) with the Qubit High sensitivity RNA reagents. In addition, size distribution of RNA fragments and the RNA integrity number (RIN) value were determined by using Agilent 2100 BioAnalyzer with the RNA nano chip. RNA samples for whole genome sequencing were sent to Edinburgh Genomics for Library preparation and Sequence data generation. Library preparation was performed using the TruSeq stranded mRNA-seq library kit (Illumina). The libraries were then sequenced on NovaSeq with 50 paired-end (PE) reads and aiming for at least 375 M reads per sample.

## RNA-seq data processing

RNA-seq reads were aligned to the genome using the nf-core/rnaseq (*v3.12.0*, https://doi.org/10.5281/zenodo.1400710) workflow from the nf-core collection of workflows (Ewels et al, 2020). Briefly, the pipeline was executed with Nextflow *v23.04.2* (Di Tommaso et al, 2017) and aligned reads to the hg38 genome using the STAR aligner (Dobin et al, 2013). Reads aligning completely within ENSEMBL (version 103/GENCODE 37) transcripts were then counted using Salmon (Patro et al, 2017) and scaled for gene length. For a summary of RNA-seq alignment statistics, see Appendix Table S4.

## RNA-seq data analysis

Differential expression analysis was conducted using the DESeq2 package (*version 1.34.0*) in R (*version 4.1.2*) (Love et al, 2014). We

obtained gene counts from the *nf-core/rnaseq* pipeline output *salmon.merged.gene_counts_length_scaled.tsv* file and imported them into DESeq2 using the *DESeqDataSetFromMatrix* method. Lowly expressed transcripts were filtered out of this set using the default settings from DESeq2 leaving 30,017 transcripts. Counts were normalised using the variance stabilising transformation (VST). We then fitted negative binomial generalised linear models with genotype as the main factor and used the Wald test to identify differentially expressed genes ($P$ value < 0.05 and $s$-value < 0.01). We applied the apeglm method to shrink the log2 fold changes for estimation stability and improved ranking of genes (Zhu et al, 2019). Differentially expressed genes were annotated the genes using the biomaRt package (*version 2.50.3*) and the *org.Hs.eg.db* package (*version 3.14.0*).

To extract list of genes overlapping with HCT116 H3K9me3 domains, we used UCSC table browser to output a list of Ensembl Transcripts overlapping (GENCODE3 V43) with our bed file. These were then matched to Ensembl Gene IDs using Biomart in Ensembl Genes version 103 (*version 103/GENCODE 37*) on dataset *Human genes (GRCh38.p14)*. These genes were then filtered to retain only the 30,017 that were in the differential expression analysis by Deseq2.

## Bisulfite PCR data generation

In total, 500 ng of genomic DNA was bisulfite-converted with the EZ DNA Methylation-Gold kit (Zymo Research). Bisulfite PCR was performed using EpiTaq (Takara) using custom-made locus-specific primers containing 4 bp unique molecular identifiers (UMI) on each side of the amplicon and partial adapter sequences to enable library amplification. Primer sequences are listed in Appendix Table S1. Libraries for sequencing were amplified using unique dual index Illumina adapters and NEBNext Ultra II Q5 Master Mix (NEB). One purification step with Agencourt Ampure XP beads was performed before and after library amplification. Sequencing was performed on the Illumina iSeq 100 System using the iSeq 100 i1 Reagent v2 (300 cycle) Kit. PhiX Control v3 (Illumina) was spiked into the run at a concentration of 30% to improve cluster resolution and enable troubleshooting in the event of a run failure.

## Bisulfite PCR data analysis

To analyse bisulfite PCR data we calculated the mean percentage methylation for each sequenced fragment from its two reads. Following demultiplexing, we removed the Illumina indexes from the FASTQ headers to ensure compatibility with UMI tools (Smith et al, 2017). The extract function of UMI tools (*v1.0.0* with setting *--extract-method=string --bc-pattern = NNNN --bc-pattern2 = NNNN*) was then used to remove the UMIs from these FASTQ sequences and place them in the header. The 1-colour chemistry used in the iSeq results in the generation of artefactual high-confidence poly-G reads (https://sequencing.qcfail.com/articles/illumina-2-colour-chemistry-can-overcall-high-confidence-g-bases). A custom command line script was therefore used to identify and filter out paired reads where one, or both, of the reads had a guanine content of at least 90%. The remaining reads were trimmed using the *paired* setting of TrimGalore (*v0.6.6* with the setting: *--paired*) and aligned to the reference genome using Bismark (*v0.22.3* with settings: *--multicore 3 -N 0 -L 20*), indexed

using Samtools (*v1.13*) and deduplicated using the *dedup* functionality of UMI tools (*v1.0.0* with settings: *--paired --method=unique*). We then calculated the mean methylation of each fragment using a custom command line script. Briefly, Samtools was used to sort deduplicated reads by read name and the *bismark_methylation_extractor* function of Bismark (*v0.22.3* with settings: *-p --no_header --no_overlap*) was used to extract the methylation state of CpGs on individual reads. The command line was then used to identify the number of called CpGs per read ($n_r$) and the number of CpGs with a methylated call per read ($m_r$), and this information was used to calculate an overall mean methylation level ($m_r/n_r$) for each read. Read statistics are provided in Appendix Table S5.

## Methylation-sensitive southern blot

Overall, 1.5 μg of purified genomic DNA was digested with the methylation-sensitive BstBI enzyme (NEB) overnight and separated on 0.8% agarose electrophoresis gel. The gel was incubated in the following solutions: denaturation (0.5 M NaOH, 1.5 M NaCl), neutralisation (0.5 M Tris pH 7.5, 1.5 M NaCl) and 20x SSC, before the DNA was transferred to Hybond N+ membrane (Amersham). DNA was cross-linked to the membrane by UV irradiation (254 nm at 0.15 J), followed by hybridisation in DIG Easy Hyb buffer (Roche) with satellite II DIG-labelled probe at 42 °C for at least 4 h. The probe was generated by PCR amplification of satellite II sequence from plasmid p375M2.4 (Jackson et al, 1992) (a gift from N. Gilbert) using M13 universal primers. The blot was then washed in low (2× SSC, 0.1% SDS) and high (0.5× SSC, 0.1% SDS) stringency buffers, prior to incubation with anti-digoxigenin-AP antibody (Roche). Detection was performed using CSPD chemiluminescent substrate (Roche) on a ImageQuant LAS4000. Quantification was performed using Fiji measuring the intensity of the top 1/10th of each lane (undigested methylated DNA) normalised over the intensity of the bottom band indicated in the figure (digested hypomethylated DNA).

## Western blotting

Whole-cell extracts were obtained by sonication in urea buffer (8 M Urea, 50 mM Tris pH 7.5, 150 mM β-mercaptoethanol) and quantified using Pierce BCA Protein Assay Kit (Thermo Scientific). Extracts were analysed by SDS–polyacrylamide gel electrophoresis using 4–12% Bis-Tris NuPAGE protein gels (Life Technologies) and transferred onto a nitrocellulose membrane in 2.5 mM Tris-base, 19.2 mM glycine and 20% methanol. The primary antibodies used are: T7 Tag (D9E1X, Cell Signaling Technology, RRID: AB_2798161), DNMT3B (D7O7O, Cell Signaling Technology, RRID: AB_2799723), GAPDH (14C10, Cell Signaling Technology, RRID: AB_561053), alpha-Tubulin (T6199, Sigma, RRID: AB_477583). Images were acquired with ImageQuant LAS4000 or LAS800 following incubation with HRP-conjugated IgG (Invitrogen). For western blots of CRISPR clones, IRDye secondary antibodies (LI-COR) were used, and images acquired on The Odyssey CLx (LI-COR) machine.

## In cell fluorescent protein stability assay

DNMT3B KO cells were transfected with a polycistronic vector expressing dsRed and GFP-DNMT3Bs, generated by modifying

pLenti-DsRed-IRES-EGFP (Addgene plasmid 92194, a gift from Huda Zoghbi), or control plasmids expressing either GFP or dsRed, or no plasmids. After 48 h, cells were analysed by flow cytometry on BD LSR Fortessa Cell Analyser. Matrix compensation was set-up using negative (no plasmid), dsRed only and GFP only cells. FlowJo (BD Biosciences) was used to export the fluorescence intensities of individual dsRed+ cells. We then calculated the ratio for each cell and the mean ratio for each experiment using R. Each mutant was assessed in three independent transfection experiments.

## Protein purification

For recombinant expression DNMT3B fragments were cloned using human DNMT3B1 cDNA as template (Addgene #35552) performing ligation-independent cloning into 6xHis-Maltose binding protein (MBP) plasmid (Addgene plasmid # 29656). Mutants were incorporated using QuikChange II site-directed mutagenesis kit (Agilent) or direct cloning of synthesised double-strand gBlock fragments containing mutations (Integrated DNA technologies). A full list of DNMT3B fragments used in vitro can be found in Appendix Table S6.

DNMT3B constructs were expressed by induction at OD of 0.6–0.8 with 400 µM IPTG in BL-21 DE3 RIL $E.coli$ overnight at 18 °C in 2× YT broth. Cell pellets were resuspended in lysis buffer (30 mM Sodium Phosphate pH 7.6, 400 mM NaCl, 10% glycerol, 0.1% Triton X-100, 1 µM $ZnCl_2$, 1× Protease Inhibitor [284 ng/ml leupeptin, 1.37 µg/ml pepstatin A, 170 µg/ml phenylmethylsulfonyl fluoride and 330 µg/ml benzamindine], 1 mM AEBSF, 2 mM β-mercaptoethanol). For lysis, the solution was supplemented with lysozyme (500 µg/ml), 2 mM $MgCl_2$, and DNase (5 µg/ml) and sonicated. Purifications of proteins containing the ADD domain were supplemented with 10 µM $ZnCl2$ throughout the purification. Clarified lysate was applied to pre-equilibrated bed of Ni-NTA Agarose beads (Qiagen), washed extensively with wash buffer (15 mM Sodium Phosphate pH 7.6, 500 mM NaCl, 10% glycerol, 2 mM β-mercaptoethanol, 30 mM Imidazole) followed by elution buffer (20 mM Tris pH 7.5, 400 mM NaCl, 300 mM Imidazole, 10% glycerol, 2 mM β-mercaptoethanol). The N, N-PWWP and N-PWWP-ADD proteins were further purified by Ion exchange chromatography using 5 ml HiTrap Q column (Cytiva) pre-equilibrated with buffer A (20 mM Tris pH 7.5, 100 mM NaCl, 10% glycerol, 1 mM DTT). Proteins were dialysed into buffer A, loaded onto the column then eluted over a linear gradient of 20 column volumes (CVs) 0-50% buffer B (20 mM Tris pH 7.5, 1000 mM NaCl, 10% glycerol, 1 mM DTT). Protein-containing fractions were further purified by size exclusion chromatography using Superdex 200 10/300 (GE Healthcare) gel-filtration column in SEC buffer (20 mM HEPEs pH 7.5, 150 mM NaCl, 5% glycerol, 2 mM DTT) and the main mono-disperse protein containing peak was collected, concentrated, flash frozen in liquid nitrogen and stored at −80 °C. Protein concentrations were determined via absorbance on a NanoDrop One spectrophotometer and subsequent SDS-PAGE with comparison to known amounts of control proteins (Appendix Figs. S5h, S6a and S8a), and protein stability of mutants was compared by thermal denaturation assay (see below).

HP1α was purified using the pEC-Kan-His-3C-HP1α plasmid (a gift from Jeyaprakash Arulanadam). 6xHis-3C-HP1α constructs were expressed and cells lysed in the same manner and buffer as DNMT3B constructs. Clarified lysate was applied to 5 ml HiTrap

chelating column (Cytiva), preloaded with Nickel ions. The column was washed with 4 CV of wash buffer (20 mM Tris pH 7.5, 500 mM NaCl, 20 mM imidazole, 5% glycerol, 2 mM β-mercaptoethanol) then eluted over linear gradient of 20 CV of elution buffer (20 mM Tris pH 7.5, 400 mM NaCl, 20 mM imidazole, 5% glycerol, 2 mM β-mercaptoethanol). Protein-containing HP1α were pooled into 3.5 kDa MWCO snakeskin and cleaved with HRV3C protease (10 µg 3 C protease per 1 mg protein) overnight at 4 °C in 4 L of dialysis buffer (25 mM HEPES pH 8, 150 mM NaCl, 5% glycerol, 1 mM DTT) to remove the His-tag. 3 C protease, uncleaved HP1α and free 6xHis-tag were removed by nickel subtraction, passing over a 5 ml HiTrap chelating column (Cytiva) using the above wash buffer and collecting the flow through. After concentration, the protein was further purified by size exclusion chromatography using Superdex 200 10/300 (GE Healthcare) gel-filtration column in HP1α-SEC buffer (25 mM HEPEs pH 8, 150 mM NaCl, 5% glycerol, 2 mM DTT), flash frozen in liquid nitrogen and stored at −80 °C.

## Thermal denaturation assay

Previously purified His-MBP-TEV-DNMT3B$^{213-555}$ proteins were diluted to 1.5 µM in normalised TDA buffer (20 mM HEPES, 150 mM NaCl, 1 mM DTT, 0.5% glycerol, 1 µM $ZnCl_2$, 5× SYPRO orange from Life Technologies). In total, 50 µl reactions were imaged in triplicate in a 96-well plate. Thermal denaturation assays were performed in a Biometra TOptical RT-PCR machine, increasing temperature from 25 °C to 69.5 °C with 0.5 °C increments. Three measurements were taken for every step using an excitation wavelength of 490 nm and detection at 580 nm. Data were processed in instrument software, and melting point derived the first derivative of the signal curve. Results from two biological replicates, each with measurements from three technical replicates, were used to calculate mean $T_m$ values and standard deviations.

## Multiple sequence alignment

The multiple sequence alignments shown in Figs. 2A, 5A and Appendix Fig. S4a were performed with Clustal Omega using the following Uniprot sequences: Q9UBC3 (hDNM3B), O88509 (mDnmt3b), Q9Y6K1 (hDNMT3A) and O88508 (mDnmt3a). Colours represent the percentage of identity and were obtained using Jalview (v2.11.2) (Waterhouse et al, 2009).

## In silico analysis of PWWP domain and DNA-binding residues

Building on previous studies reporting DNMT3B binding to DNA (Zhou et al, 2018), the structures of DNMT3B's PWWP domains (mouse Protein Data Bank code 1KHC, (Zhou et al, 2018), human 5CIU (Rondelet et al, 2016)) were compared to homologous PWWP structures found to bind to DNA either in isolation (Protein Data Bank code 6IIT (Tian et al, 2019), 7LH9 (Zhang et al, 2021) https://www.rcsb.org/search?q=rcsb_pubmed_container_identifiers.pubmed_id:33556623) or in a modified nucleosome core particle (LEDGF PDB 6S01, (Wang et al, 2020)). Rigid body modelling revealed conserved lysine and arginine residues as potential DNA-binding residues. We identified two separable patches distal to the main H3 binding pocket, patch 1 comprising Lys-251 Arg-252 and patch 2 comprising Lys-276 Lys-232. We also observed

a further two positively charged residues, Lys-234 and Lys-268. However, these were deemed to be too close to the methyl-lysine binding pocket of the PWWP to mutate without affecting methylated lysine 36 binding. Comparison and PWWP domain figures were prepared in UCSF Chimera (Pettersen et al, 2004).

## Electrophoretic mobility shift assay for DNA binding

Overall, 175-bp Widom-601 DNA fragments were generated by PCR-based amplification as previously described (Wilson et al, 2019). A primer pair containing 5' 6-FAM fluorophore on one oligo was synthesised and HPLC purified by Integrated DNA technologies (sequences atggaaCacatTGCACAGGATGTAT & /56-FAM/ aATAgccacCtGCCCTGGAG). Large-scale PCR of 384 100 μl reactions were pooled, filtered and purified using a ResourceQ column and eluted over a salt gradient. Fractions with appropriate DNA and fluorophore were pooled and ethanol precipitated prior to resuspension in TLE buffer (10 mM Tris pH 8, 0.1 mM EDTA). In total, 3 nM of fluorescent 175 bp Widom-601 DNA was incubated with serial dilutions (from 4 μM to 62.5 nM) of recombinant 6xHis-MBP-DNMT3B$^{213-351}$ and variants and in 15 μl of EMSA buffer (15 mM HEPES pH 7.5, 75 mM NaCl, 0.05 mg/ml BSA, 10% glycerol, 0.05% Triton X-100, 1 mM DTT, 8% sucrose, 0.01% bromophenol blue). Samples were incubated at 4 °C for 1 h to ensure end point of binding was reached. Products were separated on a native 5% polyacrylamide gel with 1× Tris-glycine running buffer for 2 h at 4 °C. Gels were imaged on ChemiDoc MP (BioRAD) using Alexa 488 exposure setting (blue epi-illumination excitation and 532 nm/28 nm filter) to visualise fluorescent DNA. Gels were stained using Diamond DNA stain (Promega) and colloidal blue protein stain to confirm loading. EMSA assays were repeated, at least in triplicate.

## H3K9me3 peptide binding assay

Peptides of the first twenty amino acids of H3 with and without trimethylation at lysine position 9 and C-terminal fluorescein modification were synthesised and purified to greater than 95% purity (Francis Crick Institute, Chemical Biology Science Technology Platform). Initial binding assays using 6xHis-MBP-DNMT3B 1–555 protein showed a dose-response increase in fluorescence that was not observed in protein or peptide-alone samples. A full 1.5-fold titration series from 23 nM to 12 μM of 6xHis-MBP-DNMT3B 1–555 was incubated with 20 nM of fluorescent unmodified or methylated peptide in 20 mM HEPEs pH 7.5, 100 mM NaCl, 0.005% NP-40, 10 μM ZnCl$_2$, 1 mM DTT, 2.5% glycerol. Assays were performed in 25 μl reactions in a 384-well plate covered and incubated for 1 h at 18 °C. Fluorescence was measured with 480 nm excitation and 540 nm polarised filters (cut-off at 530 nm) in an M5 multimode plate reader (Molecular Devices). The resulting values were background subtracted (peptide only) and normalised by dividing relative fluorescent units by the highest concentration response to give fraction bound. Two technical repeats and two biological repeats were plotted in GraphPad Prism with a non-linear regression, and Bmax defined as 1.

## HP1-DNMT3B in vitro pull-down assay

In total, 25 μg of His-MBP or His-MBP tagged DNMT3B constructs were immobilised on 15 μl of Ni-NTA Agarose beads

(Qiagen) in 100 μl of nickel pull-down buffer (Ni-PDB) (20 mM Tris pH 7.5, 120 mM KCl, 0.01% (v/v) NP-40, 15 mM Imidazole, 10% glycerol, 50 μg/ml BSA, 1× AEBSF and 1 mM β-mercaptoethanol) by incubation at 4 °C for 2 h while rotating. Beads were washed by centrifugation at 500 × g for 1 min at 4 °C and resuspended in 100 μl of Ni-PDB and incubated with 15 μg of HP1α protein at 4 °C for 2 h while rotating. Beads were subsequently washed three times with 700 μl of PDB, resuspended in 30 μl of 2× SDS loading dye and boiled for 5 min to release bound proteins. Overall, 5% of input HP1α was loaded as a control alongside 10 μl of the pull-down reaction. Samples were analysed by SDS-PAGE and visualised by colloidal Coomassie protein stain.

## Co-immunoprecipitation

DNMT3B KO cells were transfected with DNMT3B$^{WT}$ or DNMT3B$^{ΔN}$ constructs as previously described, and cells were collected 48 h later. Around 5 million cells were resuspended in PBS 0.1% NP-40 and spun at maximum speed for 10 s. The supernatant was discarded and the pellet containing nuclei resuspended in IP Dilution Buffer 0.1% Triton (20 mM Tris-HCl pH 8, 150 mM NaCl, 1 mM EDTA, 0.1% Triton X-100). Samples were incubated on ice for 10 min before two rounds of 30 s sonication using Soniprep. Triton X-100 and MgCl$_2$ were added to a final concentration of 1% and 2 mM, respectively. In total, 2 μl of Benzonase Nuclease (Sigma) were added and samples incubated for 30 min on ice. Input aliquots were collected, and 40 μl of Protein A Dynabeads (Invitrogen) previously coupled with 8 μl T7-Tag antibody (D9E1X, Cell Signalling Technology, RRID: AB_2798161) were added and the samples incubated overnight under rotation at 4 °C. Beads were then washed three timed in IP Dilution Buffer 1% Triton at 4 °C. All buffers were supplemented with DTT, PMSF and Proteinase inhibitor Cocktails. Beads were resuspended in 40 μl of 2X Laemmli sample buffer (Sigma), boiled for 3 min, and samples loaded on SDS-PAGE for analysis.

## Nucleosome reconstitution

Human histones were expressed and purified as previously described (Dyer et al, 2004; Wilson et al, 2016; Wilson et al, 2019). Briefly, histones were expressed in BL-21 (DE3 RIL) cells and resolubilised from inclusion bodies. Histones were further purified by cation exchange chromatography prior to dialysis in 1 mM acetic acid and lyophilised and stored at −20 °C.

Histone octamers were reconstituted as previously described (Deak et al, 2023; Wilson et al, 2016). Histones were rehydrated to ~5 mg/ml in unfolding buffer (20 mM Tris pH 7.5, 7 M Guanidine-HCl, 10 mM DTT) for 30 min at room temperature. Unfolded histones were mixed together in 1.5:1.5:1:1 molar ratio of H2A:H2B:H3 (C96S C110A) or H3Kc9me3(C96S C110A):H4 and diluted in unfolding buffer to final concentration of 2 mg/ml. The histone mixture was then dialysed extensively in refolding buffer (15 mM Tris pH 7.5, 2 M NaCl, 1 mM EDTA, 5 mM β-mercaptoethanol) overnight at 4 °C. Refolded octamers were separated from soluble aggregates, tetramers, dimers and unpaired histones by size exclusion chromatography on a Superdex S200 16/60 (Cytiva) in refolding buffer. Peak fractions corresponding to the refolded octamers were pooled, concentrated and then used

immediately for nucleosome reconstitution, or stored at −20 °C after addition of 50% (v/v) glycerol.

Nucleosome-wrapping DNA fragments were generated by PCR-based amplification as previously described (Deak et al, 2023; Wilson et al, 2019). A primer pair containing 5′ 6-FAM fluorophore on one oligo were synthesised and HPLC purified by Integrated DNA technologies (sequences ATGGAACACCATTGCACAGGATGTAT & /56-FAM/AATAGCCACCtGCCCTGGAG) for 175 bp widom-601 DNA. Alternatively, a 193 bp fragment of Widom-603 DNA (Addgene #26658) was amplified (primers GAGCCGTAAAATCGTC-GACGCTCTCGGGTGCC & TGGCACCGAAACGGGTACC). This produced a fragment of 147 bp of Widom positioning sequence bounded by asymmetrical 30 bp and 16 bp linkers. Large-scale PCR of 384 100 µl reactions were pooled, filtered and purified using a ResourceQ column and eluted over a salt gradient. Fractions with appropriate DNA and fluorophore were pooled and ethanol precipitated prior to resuspension in TLE buffer (10 mM Tris pH 8, 0.1 mM EDTA).

Nucleosomes were reconstituted from octamers as previously described (Deak et al, 2023; Dyer et al, 2004; Wilson et al, 2016) with some minor modifications. Purified octamers were incubated with DNA, (FAM-175-bp Widom-601 or 193 bp Widom-603) wrapped using an 18 h exponential salt reduction gradient. The extent and purity of nucleosomes wrapping was checked by native-PAGE and SDS-PAGE analysis (Appendix Fig. 8c). Free DNA was removed by partial PEG precipitation, using 9% (w/v) PEG-6000 (FAM-175-bp Widom-601) or by 9% b(w/v) PEG-6000 with addition of 100 mM NaCl (193 bp Widom-603).

## Methyl-lysine analog installation

To generate H3Kc9me3 nucleosomes, a H3 mutant with cysteine at position 9 was purified as above (H3K9C). This protein was alkylated as previously described (Simon et al, 2007; Wilson et al, 2016). Briefly, H3K9C was resuspended in reduced denaturing buffer and alkylating (2-bromoethyl)-trimethylammonium bromide was added to a final concentration of 50 mM and incubated at 50 °C for 2 h. The reaction was quenched with β-mercaptoethanol and separated from unreacted products using PD-10 columns (GE Healthcare). Extent of formation of methyl-lysine analog was checked using 1D intact weight ESI mass spectrometry (SIRCAMs, School of Chemistry, University of Edinburgh).

## Electrophoretic mobility shift assay for DNA binding

In all, 3 nM of fluorescent 175 bp Widom-601 DNA was incubated with serial dilutions (from 4 µM to 62.5 nM) of recombinant 6xHis-MBP-DNMT3B[213-351] and variants and in 15 µl of EMSA buffer (15 mM HEPES pH 7.5, 75 mM NaCl, 0.05 mg/ml BSA, 10% glycerol, 0.05% Triton X-100, 1 mM DTT, 8% sucrose, 0.01% bromophenol blue). Samples were incubated at 4 °C for 1 h to ensure end point of binding was reached. Products were separated on a native 5% polyacrylamide gel with 1× Tris-glycine running buffer for 2 h at 4 °C. Gels were imaged on ChemiDoc MP (BioRAD) using Alexa 488 exposure setting to visualise FAM signal (blue epi-illumination excitation and 532 nm/28 nm filter). Gels were stained using Diamond DNA stain (Promega) and colloidal blue protein stain to confirm loading. EMSA assays were repeated, at least in triplicate.

## Electrophoretic mobility shift assay for nucleosome binding

6-Carboxyfluorescein (5′ 6-FAM) labelled H3Kc9me3 nucleosomes wrapped with 175 bp Widom-601 DNA fragments (0.325 ng/µl) were incubated with a twofold dilution series of 6xHis-MBP-DNMT3B 1–351 (N-PWWP) in the presence or absence of 1 µM HP1α in 12 µl of EMSA buffer (15 mM HEPES pH 7.5, 150 mM NaCl, 1 mM DTT, 5% glycerol, 0.05 mg/ml BSA, 0.1 mg/ml Salmon Sperm DNA, 0.005% NP-40). Samples were incubated on ice for 1 h before the addition of 3 µl of loading dye (40% (w/v) sucrose, bromophenol blue). Products were separated by native-PAGE using 1× Tris-Glycine running buffer for 90 min at 4 °C. Gels were imaged on ChemiDoc MP (BioRAD) using Alexa 488 exposure setting to visualise FAM signal (blue ep. Gels were stained with Diamond DNA stain (Promega) and Coomassie protein stain to confirm loading.

## Nucleosome pull-down assay

Pulldowns were performed essentially as described (Becker et al, 2021), with the amendment of using amylose beads to bind MBP tagged-proteins. 8 µg MBP, or 6xHis-MBP-DNMT3B 1–555 12 µg + 8 µg MBP, 10 µg 6xHis-MBP-DNMT3B 1–351 or 8 µg 6xHis-MBP-DNMT3B 412–555 were immobilised on 25 µl of prewashed Amylose beads (NEB) in pull-down buffer 2 (50 mM Tris pH 7.5, 150 mM NaCl, 0.02% NP-40, 0.1 mg/ml BSA, 10% glycerol, 2 mM β-mercaptoethanol). After washing 1.2 µg (based on DNA quantification) of 193 bp widom-603 unmodified or H3Kc9me3 nucleosome was added in 100 µl PDB and incubated with rotation for 2 h at 4 °C. After extensive washes in pull-down buffer 2, bound protein was eluted by boiling in 2× SDS loading buffer for 5 min. Proteins were separated on 17% SDS-PAGE and transferred to PVDF membranes for western blotting using H2B antibody (Abcam ab1790) and MBP antibody (NEB, E8032S, RRID: AB_302612) and relevant secondary antibodies. Membranes were incubated with ECL reagent (ECL supersignal, Thermo Scientific) and ChemiDoc MP (BioRAD) using chemiluminescent settings.

## DNMT3B localisation data acquisition and image analysis

For live-cell imaging, DNMT3B KO cells were seeded in poly-L-lysine-treated eight-well glass-bottom plates (ibidi GmBH) and transfected with SNAP-DNMT3B plasmid in addition to HP1α-GFP plasmid (a gift from Tom Misteli, Addgene plasmid 17652). After 48 h, cells were incubated for 30 min at 37 °C with SNAP-Cell 647-SiR substrate (NEB, final concentration 1.2 µM). Cells were then incubated for 30 min at 37 °C in fresh media to wash off unbound substrate. Imaging was performed using FluoroBrite DMEM (Gibco) supplemented with 10% foetal calf serum. Imaging was carried out with a SoRa (Nikon-Europe) using a 100 × 1.49NA oil immersion lens. Z-stack images were processed according to the manufacturer's recommended settings using NIS-Elements AR software (*v5.1*, Nikon-Europe).

To analyse the degree of DNMT3B localisation to heterochromatin, we used Imaris 9.8.0 (Bitplane, Oxford Instruments) to define heterochromatic regions of the nucleus using the 'Surfaces' function of Imaris to create a 3D isosurface of HP1α-GFP. The

'Image Calculator' Imaris XTension was used to create a channel that was the sum of both HP1α-GFP and the SNAP-tagged DNMT3B and the resultant channel was used to create a 3D isosurface representing the whole nucleus. This was done using the additive channel to ensure the whole nucleus was segmented. The Imaris module 'Cell' was then used to relate the segmented heterochromatin to the nucleus it was inside. This meant a ratio of the mean intensity of DNMT3B signal in the heterochromatic regions over the mean intensity in the rest of the nucleus could be obtained.

To analyse the degree of DNMT3B signal at the nuclear periphery from the same cells, we applied the 'Distance Transformation' XTension from Imaris to the nucleus isosurface to create a distance map channel where each voxel in the surface is valued by the distance in microns from the outside of the surface. By thresholding this channel from 0.1 to 0.5 a 3D isosurface representing the nuclear periphery was obtained. The mean intensity of this periphery has been represented as a ratio with the mean intensity of the rest of the nucleus. Z-planes used in the figures were assembled and annotated using Fiji (https://fiji.sc).

## FRAP data acquisition and analysis

For FRAP, DNMT3B KO cells were seeded in poly-L-lysine-treated eight-well glass-bottom plates (ibidi GmBH) and transfected with SNAP-DNMT3B plasmid. After 48 h, cells were incubated for 30 min at 37 °C with SNAP-Oregon green (NEB, final concentration 4 μM). Cells were then incubated for 30 min at 37 °C in fresh media to wash off unbound substrate. Imaging was performed using FluoroBrite DMEM (Gibco) supplemented with 10% foetal calf serum.

FRAP experiments were carried out on a Leica Stellaris 8 using a 20 mW 488 nm Diode laser and a 63 × 1.4NA oil immersion lens. Images were acquired every 2 s using the Resonant Scanner with line averaging of 2. Pixel size was set to nyquist (103 nm) sampling, and 256 × 256 array was acquired. Bleaching parameters: in all image series 5 'pre-bleach' images were acquired, a consistently sized area of 2 μm diameter was bleached. Bleach settings were 488 nm laser set to 100% 15 repetitions. Post-bleach images were acquired every 2 s for 150 s. FRAP analysis was performed using custom scripts, following a protocol similar to that described by Bancaud et al, (Bancaud et al, 2010). Cell movements were corrected by registering (through roto-translations) all frames with the starting one using the TurboReg plugin in Fiji (Thevenaz et al, 1998). The mean intensity of the bleaching region $I^B(t)$, as well as that of the entire nucleus $I^C(t)$ was measured. A first normalisation was performed to correct the photobleaching of the signal in time and to normalise the signal in the range [0,1], where 1 corresponds to the normalised pre-bleaching signal:

$$I^N(t) = \frac{I^B(t)/I^C(t)}{I^B_{pre}(t)/I^C_{pre}(t)}$$

where $I^B_{pre}(t)$ and $I^C_{pre}(t)$ are, respectively, the average pre-bleach signal in the bleached region and in the entire nucleus.

A second normalisation was performed to shift the minimum intensity of the signal to 0, while maintaining the signal in the range of 0 to 1. This is necessary for visually comparing FRAP curves for different conditions, as they typically have different minima in the intensity (bleaching is not 100% efficient). This second normalisation does not influence the fitting of the curves. The new normalised intensity is therefore:

$$I^{N2}(t) = \frac{I^N(t) - I^N_{min}}{1 - I^N_{min}}$$

Being $I^N_{min}$ the minimum intensity of the FRAP curve.

For every FRAP recovery curve, we calculated the fraction of the immobile (or very slow) population, defined as $IF = 1 - I^{N2}(t = +\infty)$. In our case, we approximated $I^{N2}(t = +\infty)$ as the average of the last 50 values of $I^{N2}$ (i.e., from $t = 100$ s to $t = 150$ s). The FRAP recovery curves and immobile fraction were calculated for every experiment, as well as their standard deviation. Data shown in Fig. 6E are obtained from the mean of the three biological repeats (experiment 1: WT $n = 3$, ΔN $n = 6$; experiment 2: WT $n = 5$, ΔN $n = 15$; experiment 3: WT $n = 10$, ΔN $n = 7$). The standard deviation has been calculated using the theory of error propagation for uncorrelated data.

## Statistical analysis

All tests used are two-sided unless otherwise stated. No blinding was conducted prior to analysis.

# Data availability

All sequencing data that were generated during this study are available from NCBI Gene Expression Omnibus in superseries GSE244520. The individual data types are available within the following subseries: Whole genome bisulfite sequencing, NCBI Gene Expression Omnibus: GSE244517. Histone ChIP-seq, NCBI Gene Expression Omnibus: GSE244519. T7-DNMT3B ChIP-seq, NCBI Gene Expression Omnibus: GSE244516. RNA-seq, NCBI Gene Expression Omnibus: GSE244518. All imaging data that were generated for this study are available from EBI's Bioimage archive with accession S-BIAD967.

# Peer review information

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

## Acknowledgements

We thank C Uggenti, P Heyn, S Pelliciari, AP Jackson, Y Crow, S Janssen, M Lorincz and members of the Sproul and Wilson labs for useful discussions. We thank Edinburgh Clinical Research Facility Genetics Core, MRC IGC FACs and imaging core facilities for technical support. This work has made use of the resources provided by the University of Edinburgh digital research services and the MRC IGC compute cluster. Human histones were supplied from Addgene from the Landry lab. We thank Jeyaprakash Arulanandam, University of Edinburgh, for gift of the HP1 plasmid. We are grateful to Dhira Joshi at peptide chemistry at The Francis Crick Institute for the synthesis of H3 peptides. We thank Logan Mackay in SIRCAMS school of chemistry, University of Edinburgh for mass spec analysis. DS is a Cancer Research UK Career Development fellow (reference C47648/A20837), and work in his laboratory is also supported by an Medical Research Council university grant to the MRC Human Genetics Unit. MDW's work is supported by the Wellcome Trust (210493), Medical Research Council (T029471/1), and the University of Edinburgh. This work was supported by the Edinburgh Protein Production Facility (EPPF), which receives funding from a core grant (203149) to the Wellcome Centre for Cell Biology at the University of Edinburgh. IK was funded by a studentship from Cancer Research UK (C157/A25186) as well as the AG Leventis Foundation (18736). WR is funded by a PhD studentship from the Wellcome Trust (228154/Z/23/Z). LK is a cross-disciplinary post-doctoral fellow supported by funding from the University of Edinburgh and Medical Research Council (MC_UU_00009/2). HF was funded by an ERASMUS+ scholarship. DK is funded by the Darwin Trust of Edinburgh.

## Author contributions

**Francesca Taglini**: Conceptualisation; Formal analysis; Investigation; Writing—original draft; Writing—review and editing. **Ioannis Kafetzopoulos**: Formal analysis; Investigation. **Willow Rolls**: Investigation; Writing—review and editing. **Kamila Irena Musialik**: Investigation. **Heng Yang Lee**: Investigation. **Yujie Zhang**: Investigation. **Mattia Marenda**: Formal analysis. **Lyndsay Kerr**: Formal analysis. **Hannah Finan**: Investigation. **Cristina Rubio-Ramon**: Investigation. **Philippe Gautier**: Formal analysis. **Hannah Wapenaar**: Investigation. **Dhananjay Kumar**: Investigation. **Hazel Davidson-Smith**: Investigation. **Jimi Wills**: Investigation. **Laura C Murphy**: Formal analysis. **Ann Wheeler**: Supervision. **Marcus D Wilson**: Conceptualisation; Supervision; Funding acquisition; Investigation; Writing—original draft; Writing—review and editing. **Duncan Sproul**: Conceptualisation; Formal analysis; Supervision; Funding acquisition; Writing—original draft; Writing—review and editing.

## Disclosure and competing interests statement

The authors declare no competing interests.

