## [Peer Review File · EMBO Reports]

DNMT3B PWWP mutations cause hypermethylation of heterochromatin

Francesca Taglini, Ioannis Kafetzopoulos, Willow Rolls, Kamila Musialik, Heng Lee, Yujie Zhang, Mattia Marena, Lyndsay Kerr, Hannah Finan, Cristina Rubio-Ramon, Philippe Gautier, Hannah Wapenaar, Dhananjay Kumar, Hazel Davidson-Smith, Jimi Wills, Laura Murphy, Ann Wheeler, Marcus Wilson, and Duncan Sproul

DOI: [10.15252/embr.202357357](https://doi.org/10.15252/embr.202357357)

Corresponding author: [Duncan Sproul \(d.sproul@ed.ac.uk\)](mailto:d.sproul@ed.ac.uk) , [Marcus Wilson \(marcus.wilson@ed.ac.uk\)](mailto:marcus.wilson@ed.ac.uk)

Review Timeline:

Transferred from Review Commons:	18th Apr 23
Editorial Decision:	26th Apr 23
Revision Received:	13th Nov 23
Editorial Decision:	7th Dec 23
Revision Received:	21st Dec 23
Accepted:	21st Dec 23

Editor: *Achim Breiling*

Transaction Report:

This manuscript was transferred to EMBO Reports following peer review at Review Commons.

Review #1

1. Evidence, reproducibility and clarity:

Evidence, reproducibility and clarity (Required)

Summary:

This paper by Francesca Taglini, Duncan Sproul, and their coworkers, examines the mechanisms of DNA methylation in a human cancer cell line. They use the human colorectal cancer line HCT116, which has been very widely used to look at epigenetics in cancer, and to dissect the contribution of different proteins and chromatin marks to DNA methylation.

The authors focus on the role of the de novo methyltransferase DNMT3B. It has been shown in ES cells in 2015 that its PWWP domain directs it to H3K36me3, typically found in gene bodies. More recently, the authors showed similar conclusions in colorectal cancer (Masalmeh Nat Comm 2021). Here they examine, more specifically, the role of the PWWP. The conclusions are described below.

Major comments:

1. I feel that this paper has several messages that are somewhat muddled. The main message, as expressed in the title and in the model, is that the PWWP domain of DNMT3B actively drags the protein to H3K36me3-marked regions. Inactivation of this domain by a point mutation, or removal of the Nter altogether, causes DNMT3B to relocate to other genomic regions that are H3K9me3-rich, and that see their DNA methylation increase in the mutant conditions. This first message is clear.

The second message has to do with ICF. A mutant form of DNMT3B bearing a mutation found in ICF, S270P, is actually unstable and, therefore, does not go to H3K9me3 regions. I feel that here the authors go on a tangent that distracts from message #1. This could be moved to the supp data. At any rate, HCT116 are not a good model for ICF. In addition, a previous paper has looked at the S270P mutant, and it did not seem unstable in their hands (Park J Mol Med 2008, PMID: 18762900). So I really feel the authors do not do themselves a favor with this ICF angle.

2. I feel that some major confounders exist that endanger the conclusions of the work. The most worrisome one, in my opinion, is the amount of WT or mutant DNMT3B in the cells. It is clear in figure 4C that the WT rescue construct is expressed much more than the W263A mutant (around 3 or 4 times more). Unless I am mistaken, we are never shown how the level of exogenous rescue protein compares to the level of DNMT3B in WT cells. This bothers me a lot. If the level is too low, we may have partial rescue. If it is too high, we might have artifactual effects of all that DNMT3B. I would also like to see the absolute DNA methylation values (determined by WGBS) compared to the value found in WT. From figure S1A, it looks like WT is around 80% methylation, and 3BKO is around 77% or so. I wonder if the rescue lines may actually have more methylation than WT?

3. I guess the unarticulated assumption is that the gain of DNA methylation seen at H3K9me3

region upon expression of a mutant DNMT3B is due to DNMT3B itself. But we do not know this for sure, unless the authors test a double mutant (PWWP inactive, no catalytic activity). I am not necessarily asking that they do it, but minimally they should mention this caveat.

4. I am confused as to why the authors look at different genomic regions in different figures. In figure 1 we are looking at a portion of the "left" arm of chr 16. But in figure 2B, we now look at a portion of the "right" arm of the same chromosome, which has a large 8-Mb block of H3K9me3, and is surprisingly lowly methylated in the 3BKO. This seems quite odd, and I wonder if there is a possible artifact, for instance mapping bias, deletion, or amplification in HCT116. Showing the coverage along with the methylation values would eliminate some of these concerns.

****Minor comments:****

1. The WGBS coverage is not very high, around 2.5X on average, occasionally 2X. I don't believe this affects the findings, as the authors look at large H3K9me3 regions. But the info in table S2 was hard to find and it is important. I would make it more accessible.
2. It would be nice to have a drawing showing exactly what part of the Nter was removed.
3. some figures could be clearer. I was not always sure when we were looking at a CRISPR mutant clone (W263A) versus a piggyBac rescue.
4. unless I am mistaken, all the ChIP-seq data (H3K9me3, H3K36me3 etc) come from WT cells. It is not 100% certain that they remain the same in the 3BKO, is it? This should be discussed.

2. Significance:

Significance (Required)

Strengths:

The experiments are for the most part well done and well interpreted (save for the limitations mentioned above). The techniques are appropriate and well mastered by the team. The paper is well written, the figures are nice. The authors know the field well, which translates into a good intro and a good discussion. The bioinformatics are convincing.

Limitations:

All the work is done in a single cancer cell line. One might assume the conclusions will hold in other systems, but there is no certainty at this point.

HCT116 are not the best model system to study ICF, which mostly affects lymphocytes

At present, I feel that the biological relevance of the findings is fairly unclear. The authors report what happens when DNMT3B has no functional PWWP domain. I am convinced by their conclusions, but what do they tell us, biologically? Are there, for instance, mutant forms of DNMT3B expressed in disease that have a mutant PWWP? Are there isoforms expressed during development or in certain cell types that do not have a PWWP? In these cell types, does the

distribution of DNA methylation agree with what the authors predict?

In its present state, I feel the appeal of the findings is towards a semi-specialized audience, that is interested in aberrant DNA methylation in cancer and other diseases. This is not a small audience, by the way.

3. How much time do you estimate the authors will need to complete the suggested revisions:

Estimated time to Complete Revisions (Required)

(Decision Recommendation)

Less than 1 month

Yes

Review #2

1. Evidence, reproducibility and clarity:

Evidence, reproducibility and clarity (Required)

In this manuscript, Taglini et al., describe an increased activity of DNMT3B at H3K9me3-marked regions in HCT cells. They first identify that DNA methylation at K9me3-marked regions is strongly reduced in absence of DNMT3B. Next, the authors re-express DNMT3B and DNMT3B mutant variants in the DNMT3B-KO HCT cells and assess DNA methylation by WGBS where they identify a strong preference for re-methylation of K9me3 sites. Based on genome-wide binding maps for DNMT3B, including the mutant variants, they address how the localization of DNMT3B relates to the observed changes in methylation.

****Major points:****

The authors show increased reduction of mCG at H3K9me3 (and K27me3) sites in absence of DNMT3B. This is based on correlating delta %mCG with histone modifications in 2kb bins. I find this approach to not fully support the major claim.

First, the correlation coefficients are very small -0.124 for K9me3 and -0.175 for K27me3, and just marginally better compared to, for example, K36me3 that does not seem to have any influence on mCG according to Sup Fig S1b. While I agree that mCG seems more reduced at K9me3 in absence of DNMT3B (e.g. in Fig 1a), is there a better way to visualize the global effect? The delta mCG Boxplots based on bins are not ideal (this applies to many figures using the same approach in the current manuscript).

Second, the calculation based on delta mCpG does not allow to see how much methylation was initially there. For example, S1b shows a median decrease of ~ 10% in K9me3 and ~7-8% in H3K4me3. What does this mean given that the starting methylation for both marks is completely different?

Following this point, the authors mention that mCG is already low at K9me3 domains in HCT cells (compared to other sites in the genome). I am curious if this may influence the accelerated loss of methylation in absence of DNMT3B? Any comments on this?

One issue is the lack of correlation in DNMT3B binding to H3K9me3 sites in WT cells (Fig 3). How does this explain the requirement for DNMT3B for maintenance of methylation at H3K9me3? While some of the tested mutants show some weak increase at K9me3 sites, these are not comparable to the strong binding preferences observed at K36me3 for the wt or delta N-term version.

Following the above comment, what about other methyltransferases in HCT cells? Could DNMT1 or DNMT3A function be altered in absence of DNMT3B, and the observed methylation changes could be indirectly influenced by DNMT3B? The authors could create a DNMT-TKO HCT cell line and re-introduce DNMT3B in this background and measure methylation to exclude that DNMT1 or DNMT3A could have an influence. In this case, only H3K9me3 should gain DNA methylation.

DNMT3B lacking N-terminal shows reduced K9me3 methylation & some localization by imaging. While the presented experiments show some support for this conclusion, I suggest to re-introduce a W263A mutant lacking the N-terminal part and measure changes in DNA methylation at H3K9. This should help to test the requirement for the N-terminal regions and further indicate which protein part (PWWP or N-term) is more important in regulating the balance between K9me3 and K36me3.

In the first paragraph of the discussion, the authors state: "Our results demonstrate that DNMT3B is recruited to and methylates heterochromatin in a PWWP- independent manner that is facilitated by its N-terminal region." Same statement is found in the abstract. This contradicts the

ChIP-seq results that do not indicate a recruitment of DNMT3B to heterochromatin, and the N-terminal deletions are not fully supporting a role in this targeting since there is no localization to K9me3 to begin with. While changes in methylation are observed, it remains to be determined if this is indeed through direct DNMT3B delocalization or indirectly through influencing the remaining DNMTs.

2. Significance:

Significance (Required)

Advance: detailed analysis of DNMT3B mutants in relation to K9me3. Builds up on previous studies.

Audience: specialised audience

3. How much time do you estimate the authors will need to complete the suggested revisions:

Estimated time to Complete Revisions (Required)

(Decision Recommendation)

Between 3 and 6 months

No

Review #3

1. Evidence, reproducibility and clarity:

Evidence, reproducibility and clarity (Required)

In this work, Taglini et al. examine how the de novo DNA methyltransferase DNMT3B localizes to constitutive heterochromatin marked by the repressive histone modification H3K9me3. The authors utilize a previously generated DNMT3B KO colorectal carcinoma cell line, HCT116 to study recruitment and activity of DNMT3B at constitutive H3K9me3 heterochromatin. The authors noted preferential decrease of DNA methylation (DNAm) at regions of the genome marked with H3K9me3 in DNMT3B KOs. The authors then rescued the deficiency through overexpression of WT and catalytic dead DNMT3A/B and confirmed that DNA methylation increase at H3K9me3+ region in the WT DNMT3B, but not catalytically inactive mutant nor DNMT3A. To examine which protein domains may be mediating DNMT3B's recruitment to H3K9me3 regions, the authors designed a series of mutants, primarily focusing on the PWWP domain which normally recognizes H3K36me3. In the PWWP mutants, DNMT3B binding to the genome is altered, showing depletion at some H3K36me3-marked regions and gain at H3K9me3 heterochromatin, which coincides with DNAm increase at satellites. In contrast, the clinically relevant ICF1 mutation S270P, shows DNMT3B protein destabilization and no such loss of DNAm at heterochromatin. Finally, the authors truncate the N-terminal portion of DNMT3B, and saw that this region of the protein is necessary for heterochromatin localization and subsequent DNAm of H3K9me3+ regions.

The experiments are well done with extensive controls, and the results are interesting and convincing. The structure of the manuscript could be improved for clarity and flow - for example, the PWWP mutations and truncations should be mentioned and compared together. I also found the section on ICF1 mutant to be out-of-place. More emphasis should be placed on the N-terminal mutant as this region seems to be critical to heterochromatin recruitment, and this may address whether the interaction to H3K9me3 is direct or indirect. Finally, while the epigenetic crosstalk is well-examined in this work, I would strongly urge the authors to add RNA-seq data to determine the transcriptional consequence of such chromatin disruptions (e.g. are repetitive sequences up-regulated in DNMT3B KOs?).

Comments

1. A potential caveat to the study is the use of a single cell line - colorectal cancer cell HCT116 - to draw major conclusions on the function of DNMT3B. It is worth noting the Baubec et al. study examining DNMT3B recruitment to H3K36me3 was mainly performed in murine embryonic stem cells (mESCs). It would greatly strengthen the study if the authors could perform similar type of data analysis on an independent DNMT3B KO cell line. For example, does DNMT3B localize to H3K9me3 regions in WT mESCs?
2. Did the PWWP mutant W263A show the expected loss of DNAm at H3K36me3-marked regions? In other words, was there evidence of DNAm redistribution in loss at H3K36me3+ regions and inappropriate gain at H3K9me3+ regions? Please perform intersection analysis of DMRs with other epigenomic marks (e.g. H3K27me3, H3K36me3, CpG shores) in the PWWP mutants. The study would also be strengthened greatly with the in addition of biochemical studies to confirm direct loss of binding, and possibly gain of H3K9me3 binding, in the DNMT3B

PWWP mutants.

3. Examining the tracks in Figure 3A,B, the PWWP mutants showed almost indiscriminate increase across the genome, and not specifically to H3K9me3-marked regions. Would ask the authors to speculate as to why the ChIP-seq of DNMT3B mutants do not recapitulate the heterochromatin co-localization shown by immunofluorescence.

4. It's a shame that the ICF1 mutation S270P was not characterized to the same extent as the PWWP mutants. Would consider adding WGBS for this clinically relevant mutation.

5. Figure 7 - please draw in the ICF1 and the N-terminal mutations in the model figure. Also provide legends.

2. Significance:

Significance (Required)

This is an interesting study on a timely subject. It will be of interest to multiple fields from epigenetics to development and cancer.

My expertise is in cancer, epigenetics, development.

3. How much time do you estimate the authors will need to complete the suggested revisions:

Estimated time to Complete Revisions (Required)

(Decision Recommendation)

Between 1 and 3 months

Yes

Revision Plan

Manuscript number: RC-2023-01843

Corresponding author(s): Duncan Sproul

The “revision plan” should delineate the revisions that authors intend to carry out in response to the points raised by the referees. It also provides the authors with the opportunity to explain their view of the paper and of the referee reports.

The document is important for the editors of affiliate journals when they make a first decision on the transferred manuscript. It will also be useful to readers of the reprint and help them to obtain a balanced view of the paper.

1. General Statements

We thank the reviewers for recognizing the importance of our work and for their supportive and insightful comments.

Our planned revisions focus on addressing all the comments and especially in further elucidating the molecular mechanism underpinning our observations, their consequences for cell phenotypes and reproducing our observations in an additional cell line. Our revision plan is backed up in many cases by preliminary data.

Our submitted manuscript demonstrated that DNMT3B's recruitment to H3K9me3-marked heterochromatin was mediated by the N-terminal region of DNMT3B. Data generated since submission suggest that DNMT3B binds indirectly to H3K9me3 nucleosomes through an interaction mediated by a putative HP1 motif in its N-terminal region.

Specifically, we have found that DNMT3B can pull down HP1 α and H3K9me3 from cell extracts and that this interaction is abrogated when we remove the N-terminal region of DNMT3B (revision plan, *figure 1a*). Using purified proteins *in vitro*, we have shown binding of DNMT3B to HP1 α that is dependent on the presence of DNMT3B's N-terminus suggesting that the interaction with HP1 α is direct and that this mediates DNMT3B's recruitment to H3K9me3 (revision plan, *figure 1b*). AlphaFold multimer modelling identified that DNMT3B's N-terminus binds the interface of a HP1 dimeric chromoshadow domain through a putative HP1 motif. Two point mutations in this motif ablate DNMT3B's interaction with HP1 α *in vitro* (revision plan, *figure 1b* - DNMT3B L166S I168N).

We propose to further characterize DNMT3B's interaction with HP1 α *in vitro* and determine the significance of these observations in cells by microscopy in a revised manuscript. Together with the other proposed experiments and analyses, we believe the extra detail regarding the

molecular mechanisms through which DNMT3B is recruited to H3K9me3 heterochromatin will help address the reviewer's comments.

Revision plan figure 1. DNMT3B binds to HP1 α through an HP1 motif in its N-terminal region. a) DNMT3B pulls down HP1 α and H3K9me3-modified nucleosomes from cells. Western blots following immunoprecipitation of DNMT3B^{WT} or DNMT3B^{ΔN} from DNMT3B KO cells. DNMT3B^{WT} pulls down HP1 α and H3K9me3-modified nucleosomes. Deletion of the N-terminal region ablates this interaction. b) Binding of DNMT3B to HP1 α is direct and mediated by an HP1 motif in DNMT3B's N-terminus. Coomassie stained gel from pull-down assay using His-MBP tagged DNMT3B constructs immobilized on Ni-NTA agarose beads and incubated with HP1 α . DNMT3B L166S I168N carries two mutations in the putative HP1 α motif. Removal of the N-terminal region or mutation of the HP1 α motif ablates interaction with HP1 α . DNMT3B WT = residues 1-554, DNMT3B Δ N=213-554, DNMT3B L166S I168N = 1-350. His-MBP is used as a negative control.

2. Description of the planned revisions

Insert here a point-by-point reply that explains what revisions, additional experimentations and analyses are planned to address the points raised by the referees.

We have reproduced the reviewer's comments in their entirety and highlighted them in *blue italics*.

Reviewer #1 (Evidence, reproducibility and clarity (Required)): Summary:

This paper by Francesca Taglini, Duncan Sproul, and their coworkers, examines the mechanisms of DNA methylation in a human cancer cell line. They use the human colorectal cancer line HCT116, which has been very widely used to look at epigenetics in cancer, and to dissect the contribution of different proteins and chromatin marks to DNA methylation. The authors focus on the role of the de novo methyltransferase DNMT3B. It has been shown in ES cells in 2015 that its PWWP domain directs it to H3K36me3, typically found in gene bodies. More recently, the authors showed similar conclusions in colorectal cancer (Masalmeh Nat Comm 2021). Here they examine, more specifically, the role of the PWWP. The conclusions are described below.

Major comments:

1-I feel that this paper has several messages that are somewhat muddled. The main message, as expressed in the title and in the model, is that the PWWP domain of DNMT3B actively drags the protein to H3K36me3-marked regions. Inactivation of this domain by a point mutation, or removal of the Nter altogether, causes DNMT3B to relocate to other genomic regions that are H3K9me3-rich, and that see their DNA methylation increase in the mutant conditions. This first message is clear.

We thank the reviewer for their positive comments on our observations. However, we note that our results suggest that removal of the N-terminal region has a different effect to point mutations in the PWWP domain. The data we present suggest that the N-terminus facilitates recruitment to H3K9me3 regions.

The second message has to do with ICF. A mutant form of DNMT3B bearing a mutation found in ICF, S270P, is actually unstable and, therefore, does not go to H3K9me3 regions. I feel that here the authors go on a tangent that distracts from message #1. This could be moved to the supp data. At any rate, HCT116 are not a good model for ICF. In addition, a previous paper has looked at the S270P mutant, and it did not seem unstable in their hands (Park J Mol Med 2008, PMID: 18762900). So I really feel the authors do not do themselves a favor with this ICF angle.

Revision Plan

While we agree with the reviewer that HCT116 cells as a cancer cell line are not a good model for ICF1 syndrome, our observation that S270P destabilizes DNMT3B is important to consider in the context of this disease. In addition, the S270P mutant was reported to abrogate the interaction between DNMT3B and H3K36me3 (Baubec et al 2015 Nature PMID: 25607372) making it important to compare it to the other mutations we examine. In our revised version of the manuscript, we propose to move these data to the supplementary materials and add a statement to the discussion noting the caveat that HCT116 cells are likely not to model many aspects of ICF1.

With regard to the differences between our results and that of Park et al, we note that stability of the S270P mutant was not assessed in that study whereas we directly assess stability *in vitro* and in cells. We propose to add discussion of this previous study to the revised manuscript.

2-I feel that some major confounders exist that endanger the conclusions of the work. The most worrisome one, in my opinion, is the amount of WT or mutant DNMT3B in the cells. It is clear in figure 4C that the WT rescue construct is expressed much more than the W263A mutant (around 3 or 4 times more). Unless I am mistaken, we are never shown how the level of exogenous rescue protein compares to the level of DNMT3B in WT cells. This bothers me a lot. If the level is too low, we may have partial rescue. If it is too high, we might have artifactual effects of all that DNMT3B. I would also like to see the absolute DNA methylation values (determined by WGBS) compared to the value found in WT. From figure S1A, it looks like WT is around 80% methylation, and 3BKO is around 77% or so. I wonder if the rescue lines may actually have more methylation than WT?

The rescue cell lines do express DNMT3B to a greater level than observed endogenously. In our manuscript we controlled for this effect by generating the knock-in W263A cells and, as reported in the manuscript, we observe similar effects to the rescue cells (manuscript, *figure 2d*) suggesting that our observations are not driven by the overexpression.

We also expressed ectopic DNMT3B from a weaker promoter (EF1 α) in DNMT3B KO cells but did not include these data in the submitted manuscript. We have previously shown that this promoter expresses DNMT3B at lower levels than the CAG promoter used in the submitted manuscript (Masalmeh et al 2021 Nature Communications PMID: 33514701). Bisulfite PCR of representative non-repetitive loci within heterochromatic H3K9me3 domains show that we observe similar gains of methylation with DNMT3B^{W263A} (revision plan, *figure 2*).

Revision plan figure 2. Expression of *DNMT3B*^{W236A} from a weaker promoter leads to increased DNA methylation at selected H3K9me3 loci. Barplot of mean methylation by BS-PCR at H3K9me3 loci alongside the H3K4me3-marked *BRCA2* promoter in *DNMT3B* mutant cells where *DNMT3B* is expressed from the *EF1* promoter. P-values are from two-sided Wilcoxon rank sum tests.

To reinforce that our conclusions are not solely a result of the level of *DNMT3B* expression, we propose to include these data in the revised manuscript.

The reviewer is also correct that by WGBS, the rescue cell lines have higher levels of overall DNA methylation than HCT116 cells. We will note this in revised manuscript and include HCT116 cells in a revised version of *Figure S1e*.

3-I guess the unarticulated assumption is that the gain of DNA methylation seen at H3K9me3 region upon expression of a mutant DNMT3B is due to DNMT3B itself. But we do not know this for sure, unless the authors test a double mutant (PWWP inactive, no catalytic activity). I am not necessarily asking that they do it, but minimally they should mention this caveat.

The hypothesis that the gains in DNA methylation at H3K9me3 loci result from the direct catalytic activity of *DNMT3B* is supported by our observation that a catalytic dead *DNMT3B* does not remethylate heterochromatin (manuscript, *figures 1d* and *e*). However, we acknowledge that we have not formally shown that the additional DNA methylation seen with *DNMT3B*^{W263A} are a direct result of its catalytic activity. We will conduct an analysis of the effect of catalytically dead *DNMT3B*^{W263A} on DNA methylation at Satellite II and selected H3K9me3 loci and include this in the revised manuscript.

4-I am confused as to why the authors look at different genomic regions in different figures. In figure 1 we are looking at a portion of the "left" arm of chr 16. But in figure 2B, we now look at a portion of the "right" arm of the same chromosome, which has a large 8-Mb block of H3K9me3,

Revision Plan

and is surprisingly lowly methylated in the 3BKO. This seems quite odd, and I wonder if there is a possible artifact, for instance mapping bias, deletion, or amplification in HCT116. Showing the coverage along with the methylation values would eliminate some of these concerns.

By choosing different regions of the genome for different figures, we intended to reassure the reader that our results were not specific to any one region of the genome. In the revised manuscript, we propose to display a consistent genomic region between these figures.

With regard to the low levels of DNA methylation in H3K9me3 domains in DNMT3B KO cells, H3K9me3 domains are partially methylated domains which have reduced methylation in HCT116 cells (see page 5 of the manuscript):

... we found that hidden Markov model defined H3K9me3 domains significantly overlapped with extended domains of overall reduced methylation termed partially methylated domains (PMDs) defined in our HCT116 WGBS (Jaccard=0.575, $p=1.07 \times 10^{-6}$, Fisher's test).

These domains lose further DNA methylation in DNMT3B KO cells leading to the low methylation level noted by the reviewer. The methylation percentages calculated from WGBS are based on the ratio of methylated to total reads. Thus, a lack of coverage generates errors from division by zero rather than the low values observed in this domain in DNMT3B KO cells.

We include a modified version of *figure 2b* from the manuscript below. This includes coverage for the 3 cell lines (revision plan, *figure 3*). Although WGBS coverage is slightly reduced in H3K9me3 domains, reads are still present and overall coverage equal between different cell lines.

Revision plan figure 3. Version of figure 2b showing WGBS coverage. Genome browser plots showing absolute (black) and differential (gain=red/loss=blue) DNA methylation levels, WGBS coverage (CpG coverage, grey), DNMT3B KO ChIP-seq signals at H3K9me3 domains in DNMT3B KO, DNMT3BWT and DNMT3BW263A cells. ChIP-seq are normalised reads per 10^6 .

Revision Plan

While we could potentially include the coverage tracks in revised versions of figures, we note that doing so for multiple cell lines would make these figures extensively cluttered and it would likely be difficult to observe the differences in DNA methylation in these figure panels due to shrinkage of the other tracks.

Minor comments:

1-The WGBS coverage is not very high, around 2.5X on average, occasionally 2X. I don't believe this affects the findings, as the authors look at large H3K9me3 regions. But the info in table S2 was hard to find and it is important. I would make it more accessible.

In the revised manuscript we will specify the mean coverage in the text to ensure this is clearer.

2-It would be nice to have a drawing showing exactly what part of the Nter was removed.

We will add this in the figure in the revised manuscript.

3-some figures could be clearer. I was not always sure when we were looking at a CRISPR mutant clone (W263A) versus a piggyBac rescue.

In the revised manuscript we will clarify in the figure labels to ensure it is clear which data were generated using CRISPR clones.

4-unless I am mistaken, all the ChIP-seq data (H3K9me3, H3K36me3 etc) come from WT cells. It is not 100% certain that they remain the same in the 3BKO, is it? This should be discussed.

We performed ChIP-seq on both HCT116 and 3BKO cell lines and used ChIP-seq data from the 3BKO cell line for the rescue experiments where DNMT3Bs were expressed in 3BKO cells. We will ensure this is clearer in the revised version.

Reviewer #1 (Significance (Required)):

Strengths:

The experiments are for the most part well done and well interpreted (save for the limitations mentioned above). The techniques are appropriate and well mastered by the team. The paper is well written, the figures are nice. The authors know the field well, which translates into a good intro and a good discussion. The bioinformatics are convincing.

Revision Plan

Limitations:

All the work is done in a single cancer cell line. One might assume the conclusions will hold in other systems, but there is no certainty at this point.

We acknowledge this limitation. To demonstrate that our results are applicable beyond HCT116 cells, we will include analysis of experiments on an independent cell line in the revised manuscript.

HCT116 are not the best model system to study ICF, which mostly affects lymphocytes. At present, I feel that the biological relevance of the findings is fairly unclear. The authors report what happens when DNMT3B has no functional PWWP domain. I am convinced by their conclusions, but what do they tell us, biologically? Are there, for instance, mutant forms of DNMT3B expressed in disease that have a mutant PWWP? Are there isoforms expressed during development or in certain cell types that do not have a PWWP? In these cell types, does the distribution of DNA methylation agree with what the authors predict?

As stated in response to point 1, although we acknowledge the limitations of HCT116 cells as a model of ICF, we believe are finding that the S270P mutation results in unstable DNMT3B are still important to consider for ICF syndrome.

We are not aware of reports of mutations affecting the residues of DNMT3B's PWWP domain we have studied. Our preliminary analysis suggests that although mutations in DNMT3B's PWWP domain are frequent, residues in the aromatic cage such as W263 and D266 are absent from the gnomAD catalogue (Karczewski et al 2020 Nature, PMID: 32461654). This suggests that they are incompatible with healthy human development.

A number of different DNMT3B splice isoforms have been reported. These include Δ DNMT3B4 which lacks the PWWP domain and a portion of the N-terminal region (Wang et al 2006 International Journal of Oncology, PMID: 16773201). Δ DNMT3B4 is proposed to be expressed in non-small cell lung cancer (Wang et al 2006 Cancer Research, PMID: 16951144).

We will include analysis of gnomAD and discussion of these points in the revised manuscript.

In its present state, I feel the appeal of the findings is towards a semi-specialized audience, that is interested in aberrant DNA methylation in cancer and other diseases. This is not a small audience, by the way.

We thank the reviewer for their comments and the suggestion that our findings are of interest to a cross-section of researchers.

Revision Plan

Reviewer #2 (Evidence, reproducibility and clarity (Required)):

Note, we have added numbers to the comments made by reviewer 2 to aid cross-referencing.

In this manuscript, Taglini et al., describe an increased activity of DNMT3B at H3K9me3-marked regions in HCT cells. They first identify that DNA methylation at K9me3-marked regions is strongly reduced in absence of DNMT3B. Next, the authors re-express DNMT3B and DNMT3B mutant variants in the DNMT3B-KO HCT cells and assess DNA methylation by WGBS where they identify a strong preference for re-methylation of K9me3 sites. Based on genome-wide binding maps for DNMT3B, including the mutant variants, they address how the localization of DNMT3B relates to the observed changes in methylation.

Major points:

1. The authors show increased reduction of mCG at H3K9me3 (and K27me3) sites in absence of DNMT3B. This is based on correlating delta %mCG with histone modifications in 2kb bins. I find this approach to not fully support the major claim.

First, the correlation coefficients are very small -0.124 for K9me3 and -0.175 for K27me3, and just marginally better compared to, for example, K36me3 that does not seem to have any influence on mCG according to Sup Fig S1b. While I agree that mCG seems more reduced at K9me3 in absence of DNMT3B (e.g. in Fig 1a), is there a better way to visualize the global effect? The delta mCG Boxplots based on bins are not ideal (this applies to many figures using the same approach in the current manuscript).

Our choice to examine the global effects using correlations in windows across the genome was motivated by similar previous analyses in other studies (for example: Baubec et al. 2015 Nature PMID 25607372, Weinberg et al 2021 Nature PMID:33986537, Neri et al 2017 Nature PMID: 28225755). These global analyses result in modest correlation coefficients because the vast majority of genomic windows are negative for a given mark. For this reason, we included specific analyses of H3K36me3, H3K9me3 and H3K27me3 domains in the manuscript (eg manuscript figure 1b, c and d) which reinforce the conclusions drawn from our global analyses.

However, we acknowledge that while our data support a specific activity at H3K9me3 marked heterochromatin, these are not the only changes in DNMT3B KO cells as DNMTs are promiscuous enzymes that are localized to multiple genomic regions. We will add discussion of this point to the revised manuscript.

2. Second, the calculation based on delta mCpG does not allow to see how much methylation was initially there. For example, S1b shows a median decrease of ~ 10% in K9me3 and ~7-8% in H3K4me3. What does this mean given that the starting methylation for both marks is completely different?

Following this point, the authors mention that mCG is already low at K9me3 domains in HCT cells (compared to other sites in the genome). I am curious if this may influence the accelerated loss of methylation in absence of DNMT3B? Any comments on this?

The observation that there is a greater loss at H3K9me3 domains than H3K27me3-only domains which also have low DNA methylation levels in HCT116 argue that the losses are not solely driven by the lower initial level of methylation in H3K9me3 domains. Our analyses later in the manuscript also support a specific activity at H3K9me3. In addition, we propose to reinforce this point through further data on exploring how DNMT3B interacts with HP1 α (see general comments, revision plan *figure 1*).

However, we acknowledge the possibility that part of the loss seen at H3K9me3 domains in DNMT3B KO cells could be in part a result of their low initial level of methylation. In the revised manuscript we propose to include discussion of this possibility.

3. One issue is the lack of correlation in DNMT3B binding to H3K9me3 sites in WT cells (Fig 3). How does this explain the requirement for DNMT3B for maintenance of methylation at H3K9me3? While some of the tested mutants show some weak increase at K9me3 sites, these are not comparable to the strong binding preferences observed at K36me3 for the wt or delta N-term version.

Using ChIP-seq we cannot say that DNMT3B^{WT} does not bind at H3K9me3, only that it binds here to a lower level than at K36me3-marked loci. The normalized DNMT3B^{WT} signal at H3K9me3 domains is higher than the background signal from DNMT3B KO cells (manuscript *figure 3d*) supporting the hypothesis that DNMT3B^{WT} localizes to H3K9me3. This hypothesis is also supported by the observation that the correlation between DNMT3B^{AN} and H3K9me3 is reduced compared to that of DNMT3B^{WT} (manuscript *figure 6c* compared to *figure 3c*).

There are several reasons why the apparent enrichment of DNMT3B at H3K9me3 may appear weaker than at H3K36me3 by ChIP-seq. Previous work has also suggested that formaldehyde crosslinking fails to capture transient interactions in cells (Schmiedeberg et al. 2009 PLoS One PMID: 19247482). H3K9me3-marked heterochromatin is also resistant to sonication (Becker et al. 2017 Molecular Cell PMID: 29272703) and this could further affect our ability to detect DNMT3B in these regions using ChIP-seq. Our new data also suggest that DNMT3B binds to H3K9me3 indirectly through HP1 α (see general comments, revision plan *figure 1*) and this may also lead to weaker ChIP-seq enrichment at H3K9me3 compared to the direct interaction with H3K36me3 through DNMT3B's PWWP domain.

We propose to add discussion of these issues to the revised manuscript.

4. Following the above comment, what about other methyltransferases in HCT cells? Could DNMT1 or DNMT3A function be altered in absence of DNMT3B, and the observed methylation

Revision Plan

changes could be indirectly influenced by DNMT3B? The authors could create a DNMT-TKO HCT cell line and re-introduce DNMT3B in this background and measure methylation to exclude that DNMT1 or DNMT3A could have an influence. In this case, only H3K9me3 should gain DNA methylation.

As discussed in response to reviewer 1 (point 3), we propose to examine the changes in DNA methylation upon expression of catalytically dead DNMT3B^{W263A} to further strengthen the evidence that DNMT3B^{W263A} is directly responsible for the increased DNA methylation at H3K9me3-marked loci.

5. DNMT3B lacking N-terminal shows reduced K9me3 methylation & some localization by imaging. While the presented experiments show some support for this conclusion, I suggest to re-introduce a W263A mutant lacking the N-terminal part and measure changes in DNA methylation at H3K9. This should help to test the requirement for the N-terminal regions and further indicate which protein part (PWWP or N-term) is more important in regulating the balance between K9me3 and K36me3.

We have performed this experiment and the data are shown in manuscript *figure S6c and d*. The results of these experiments show that DNMT3B^{ΔN+W263A} cells showed less methylation at H3K9me3 loci than DNMT3B^{W263A} cells, supporting a role for the N-terminus in recruiting DNMT3B to H3K9me3-marked heterochromatin. In the revised version, we will ensure that these data are more clearly indicated.

In the first paragraph of the discussion, the authors state: "Our results demonstrate that DNMT3B is recruited to and methylates heterochromatin in a PWWP- independent manner that is facilitated by its N-terminal region." Same statement is found in the abstract. This contradicts the ChIP-seq results that do not indicate a recruitment of DNMT3B to heterochromatin, and the N-terminal deletions are not fully supporting a role in this targeting since there is no localization to K9me3 to begin with. While changes in methylation are observed, it remains to be determined if this is indeed through direct DNMT3B delocalization or indirectly through influencing the remaining DNMTs.

As discussed above, there are several potential reasons why DNMT3B ChIP-seq signal at H3K9me3 is weak (reviewer 2, point 3). The additional experiments we propose to include in the revised manuscript could reinforce this statement by clarifying whether DNMT3B is directly responsible for methylating H3K9me3-marked regions (reviewer 1, point 3) and by delineating the role of the putative HP1α motif in DNMT3B's N-terminal region (general comments, revision plan *figure 1*).

Reviewer #2 (Significance (Required)):

Advance: detailed analysis of DNMT3B mutants in relation to K9me3. Builds up on previous studies. Audience: specialised audience

We thank the reviewer for their insights.

Revision Plan

Reviewer #3 (Evidence, reproducibility and clarity (Required)):

In this work, Taglini et al. examine how the de novo DNA methyltransferase DNMT3B localizes to constitutive heterochromatin marked by the repressive histone modification H3K9me3. The authors utilize a previously generated DNMT3B KO colorectal carcinoma cell line, HCT116 to study recruitment and activity of DNMT3B at constitutive H3K9me3 heterochromatin. The authors noted preferential decrease of DNA methylation (DNAm) at regions of the genome marked with H3K9me3 in DNMT3B KOs. The authors then rescued the deficiency through overexpression of WT and catalytic dead DNMT3A/B and confirmed that DNA methylation increase at H3K9me3+ region in the WT DNMT3B, but not catalytically inactive mutant nor DNMT3A. To examine which protein domains may be mediating DNMT3B's recruitment to H3K9me3 regions, the authors designed a series of mutants, primarily focusing on the PWWP domain which normally recognizes H3K36me3. In the PWWP mutants, DNMT3B binding to the genome is altered, showing depletion at some H3K36me3-marked regions and gain at H3K9me3 heterochromatin, which coincides with DNAm increase at satellites. In contrast, the clinically relevant ICF1 mutation S270P, shows DNMT3B protein destabilization and no such loss of DNAm at heterochromatin. Finally, the authors truncate the N-terminal portion of DNMT3B, and saw that this region of the protein is necessary for heterochromatin localization and subsequent DNAm of H3K9me3+ regions.

The experiments are well done with extensive controls, and the results are interesting and convincing. The structure of the manuscript could be improved for clarity and flow - for example, the PWWP mutations and truncations should be mentioned and compared together. I also found the section on ICF1 mutant to be out-of-place.

As described above (reviewer 1, point 1), we propose to move these data to the supplementary materials in the revised manuscript.

More emphasis should be placed on the N-terminal mutant as this region seems to be critical to heterochromatin recruitment, and this may address whether the interaction to H3K9me3 is direct or indirect.

As described above (general comments), the revised manuscript will include experiments clarifying the nature of DNMT3B's interaction with H3K9me3. Our preliminary data support that it is an indirect interaction mediated through HP1 α (revision plan *figure 1*).

Finally, while the epigenetic crosstalk is well-examined in this work, I would strongly urge the authors to add RNA-seq data to determine the transcriptional consequence of such chromatin disruptions (e.g. are repetitive sequences up-regulated in DNMT3B KOs?).

As suggested by the reviewer, we propose to generate and analyse RNA-seq data in the revised manuscript to understand the impact of DNMT3B on transcriptional programs.

Revision Plan

Comments

1. A potential caveat to the study is the use of a single cell line - colorectal cancer cell HCT116 - to draw major conclusions on the function of DNMT3B. It is worth noting the Baubec et al. study examining DNMT3B recruitment to H3K36me3 was mainly performed in murine embryonic stem cells (mESCs). It would greatly strengthen the study if the authors could perform similar type of data analysis on an independent DNMT3B KO cell line. For example, does DNMT3B localize to H3K9me3 regions in WT mESCs?

As described above in response to reviewer 1, we will include analysis in an additional cell line in the revised manuscript to demonstrate that our results are generalizable beyond HCT116 cells.

2. Did the PWWP mutant W263A show the expected loss of DNAm at H3K36me3-marked regions? In other words, was there evidence of DNAm redistribution in loss at H3K36me3+ regions and inappropriate gain at H3K9me3+ regions? Please perform intersection analysis of DMRs with other epigenomic marks (e.g. H3K27me3, H3K36me3, CpG shores) in the PWWP mutants.

Our analysis of DNMT3B KO cells (manuscript *figure s1d*) show that losses of DNA methylation in these cells are not correlated with H3K36me3 in gene bodies suggesting that DNMT3A and DNMT1 are sufficient to compensate in maintaining their methylation in DNMT3B KO cells. To clarify this point for the DNMT3B^{W263A} knock in clones, in the revised manuscript we will directly examine whether these cells show loss of methylation at H3K36me3 marked gene bodies in a similar analysis and add discussion of these results.

The study would also be strengthened greatly with the in addition of biochemical studies to confirm direct loss of binding, and possibly gain of H3K9me3 binding, in the DNMT3B PWWP mutants.

As detailed above (general comments, revision plan *figure 1*), our data suggest that DNMT3B interacts indirectly with H3K9me3 through an HP1 motif in its N-terminal region. We will undertake further biochemical studies on this interaction which will be included in the revised manuscript. Specifically we will focus on using EMSAs with synthetic nucleosomes to clarify the degree to which the HP1 α interaction is responsible for binding of DNMT3B to H3K9me3 modified nucleosomes.

We also propose to undertake *in vitro* biochemical characterization of the effect of DNMT3B PWWP mutations on interaction with H3K36me3 using synthetic nucleosomes. However, we note that in the manuscript we have shown similar effects using two independent point mutations that are predicted to affect H3K36me3 binding (W263A and D266A) and deletion of the entire PWWP domain.

3. Examining the tracks in Figure 3A,B, the PWWP mutants showed almost indiscriminate increase across the genome, and not specifically to H3K9me3-marked regions. Would ask the authors to speculate as to why the ChIP-seq of DNMT3B mutants do not recapitulate the heterochromatin co-localization shown by immunofluorescence.

As discussed in response to reviewer 2 (point 3) we believe that the weak DNMT3B ChIP-seq signal at H3K9me3 loci is likely due to the nature of the interaction that DNMT3B has with chromatin in these regions. We will add discussion of these points to the revised manuscript.

4. It's a shame that the ICF1 mutation S270P was not characterized to the same extent as the PWWP mutants. Would consider adding WGBS for this clinically relevant mutation.

We have shown that this mutant does not produce stable protein *in vitro* or in our cells and we observe little difference in DNA methylation at selected loci. As WGBS is expensive, we believe that carrying out this experiment is not an efficient use of limited research resources.

5. Figure 7 - please draw in the ICF1 and the N-terminal mutations in the model figure. Also provide legends.

We will modify the manuscript to include these details in the revised manuscript.

Reviewer #3 (Significance (Required)):

This is an interesting study on a timely subject. It will be of interest to multiple fields from epigenetics to development and cancer. My expertise is in cancer, epigenetics, development.

We thank the reviewer for highlighting the broad interest of our study.

3. Description of the revisions that have already been incorporated in the transferred manuscript

Please insert a point-by-point reply describing the revisions that were already carried out and included in the transferred manuscript. If no revisions have been carried out yet, please leave this section empty.

We have not yet incorporated revisions into the manuscript.

4. Description of analyses that authors prefer not to carry out

Please include a point-by-point response explaining why some of the requested data or additional analyses might not be necessary or cannot be provided within the scope of a revision. This can be due to time or resource limitations or in case of disagreement about the necessity of such additional data given the scope of the study. Please leave empty if not applicable.

Revision Plan

We suggest that the following experiments/analyses are unlikely to be necessary:

Adding coverage tracks to genome browser shots (reviewer 1 point 4):

While we could potentially include the coverage tracks in revised figures, we note that doing so for multiple cell lines would make these figures extensively cluttered and it would be difficult to observe the differences in DNA methylation in these figure panels with the additional tracks added.

WGBS of the S270P expressing cells (suggested by Reviewer 3, point 4):

As detailed above in response to the reviewer, we have shown that this mutant does not produce stable protein in cells or *in vitro* and we observe minimal differences in DNA methylation at satellite II and H3k9me3 loci. As WGBS is expensive, we believe that carrying out this experiment is not an efficient use of limited research resources.

Dear Dr. Sproul,

Thank you for the transfer of your research manuscript from Review Commons to EMBO reports. I now went through your manuscript, the referee reports from Review Commons (attached again below) and your revision plan. The referees have several comments, concerns, and suggestions to improve the manuscript, indicating that a major revision of the manuscript is necessary to allow publication of the study.

Going through your revision plan, it seems that most of these points will be adequately addressed during revision. Let me point out that a major concern that needs to be addressed experimentally is to strengthen the physiological relevance of the findings (and the disease link), in particular by adding experimental data indicating that the mechanism is a general one, i.e. by including experiments in a more physiological cell line.

I thus invite you to revise your manuscript accordingly with the understanding that all concerns must be addressed in the revised manuscript and in a detailed point-by-point response (as indicated in your revision plan). Acceptance of your manuscript will depend on a positive outcome of another round of review using the same set of referees. It is EMBO reports policy to allow a single round of major revision only and acceptance of the manuscript will therefore depend on the completeness of your responses included in the next, final version of the manuscript.

- 1) a .docx formatted version of the final manuscript text (including legends for main figures, EV figures and tables), but without the figures included. Figure legends should be compiled at the end of the manuscript text.
- 2) individual production quality figure files as .eps, .tif, .jpg (one file per figure), of main figures (up to 8) and EV figures. Please upload these as separate, individual files upon re-submission.

For more details, please refer to our guide to authors:
<http://www.embopress.org/page/journal/14693178/authorguide#manuscriptpreparation>

Please consult our guide for figure preparation:
http://wol-prod-cdn.literatumonline.com/pb-assets/embo-site/EMBOPress_Figure_Guidelines_061115-1561436025777.pdf

See also the guidelines for figure legend preparation:
<https://www.embopress.org/page/journal/14693178/authorguide#figureformat>

- 4) a complete author checklist, which you can download from our author guidelines (<https://www.embopress.org/page/journal/14693178/authorguide>). Please insert page numbers in the checklist to indicate where the requested information can be found in the manuscript. The completed author checklist will also be part of the RPF.

5) that primary datasets produced in this study (e.g. RNA-seq, ChIP-seq, structural and array data) are deposited in an appropriate public database. If no primary datasets have been deposited, please also state this in a dedicated section (e.g. 'No primary datasets have been generated and deposited'), see below.

The accession numbers and database should be listed in a formal "Data Availability" section (placed after Materials & Methods) that follows the model below. This is now mandatory (like the COI statement). Please note that the Data Availability Section is restricted to new primary data that are part of this study. This section is mandatory. As indicated above, if no primary datasets have been deposited, please state this in this section

Data availability

8) Regarding data quantification and statistics, please make sure that the number "n" for how many independent experiments were performed, their nature (biological versus technical replicates), the bars and error bars (e.g. SEM, SD) and the test used to calculate p-values is indicated in the respective figure legends (also for potential EV figures and all those in the final Appendix). Please also check that all the p-values are explained in the legend, and that these fit to those shown in the figure. Please provide statistical testing where applicable. Please avoid the phrase 'independent experiment', but clearly state if these were biological or technical replicates. Please also indicate (e.g. with n.s.) if testing was performed, but the differences are not significant. In case n=2, please show the data as separate datapoints without error bars and statistics. See also: <http://www.embopress.org/page/journal/14693178/authorguide#statisticalanalysis>

9) Please also note our reference format:

10) We updated our journal's competing interests policy in January 2022 and request authors to consider both actual and perceived competing interests. Please review the policy <https://www.embopress.org/competing-interests> and update your competing interests if necessary. Please name this section 'Disclosure and Competing Interests Statement' and put it after the Acknowledgements section.

11) We now use CRedit to specify the contributions of each author in the journal submission system. CRedit replaces the author contribution section. Please use the free text box to provide more detailed descriptions and do not provide an author contributions section in the main manuscript text file. See also guide to authors: <https://www.embopress.org/page/journal/14693178/authorguide#authorshipguidelines>

12) Please add scale bars of similar style and thickness to all the microscopic images, using clearly visible black or white bars (depending on the background). Please place these in the lower right corner of the images themselves. Please do not write on

or near the bars in the image but define the size in the respective figure legend.

13) Please add up to 5 keywords to the title page (below the abstract).

14) Please provide an abstract with not more than 175 words and order the manuscript sections like this, using these names: Title page - Abstract - Keywords - Introduction - Results - Discussion - Materials and Methods - Data availability section - Acknowledgements - Disclosure and Competing Interests Statement - References - Figure legends - Expanded View Figure legends

I look forward to seeing a revised version of your manuscript when it is ready. Please let me know if you have questions or comments regarding the revision.

Best,

Achim Breiling
Senior editor
EMBO reports

Referee #1:

This paper by Francesca Taglini, Duncan Sproul, and their coworkers, examines the mechanisms of DNA methylation in a human cancer cell line. They use the human colorectal cancer line HCT116, which has been very widely used to look at epigenetics in cancer, and to dissect the contribution of different proteins and chromatin marks to DNA methylation.

The authors focus on the role of the de novo methyltransferase DNMT3B. It has been shown in ES cells in 2015 that its PWWP domain directs it to H3K36me3, typically found in gene bodies. More recently, the authors showed similar conclusions in colorectal cancer (Masalmeh Nat Comm 2021). Here they examine, more specifically, the role of the PWWP. The conclusions are described below.

****Major comments:****

1. I feel that this paper has several messages that are somewhat muddled. The main message, as expressed in the title and in the model, is that the PWWP domain of DNMT3B actively drags the protein to H3K36me3-marked regions. Inactivation of this domain by a point mutation, or removal of the Nter altogether, causes DNMT3B to relocate to other genomic regions that are H3K9me3-rich, and that see their DNA methylation increase in the mutant conditions. This first message is clear.

The second message has to do with ICF. A mutant form of DNMT3B bearing a mutation found in ICF, S270P, is actually unstable and, therefore, does not go to H3K9me3 regions. I feel that here the authors go on a tangent that distracts from message #1. This could be moved to the supp data. At any rate, HCT116 are not a good model for ICF. In addition, a previous paper has looked at the S270P mutant, and it did not seem unstable in their hands (Park J Mol Med 2008, PMID: 18762900). So I really feel the authors do not do themselves a favor with this ICF angle.

2. I feel that some major confounders exist that endanger the conclusions of the work. The most worrisome one, in my opinion, is the amount of WT or mutant DNMT3B in the cells. It is clear in figure 4C that the WT rescue construct is expressed much more than the W263A mutant (around 3 or 4 times more). Unless I am mistaken, we are never shown how the level of exogenous rescue protein compares to the level of DNMT3B in WT cells. This bothers me a lot. If the level is too low, we may have partial rescue. If it is too high, we might have artifactual effects of all that DNMT3B. I would also like to see the absolute DNA methylation values (determined by WGBS) compared to the value found in WT. From figure S1A, it looks like WT is around 80% methylation, and 3BKO is around 77% or so. I wonder if the rescue lines may actually have more methylation than WT?

3. I guess the unarticulated assumption is that the gain of DNA methylation seen at H3K9me3 region upon expression of a mutant DNMT3B is due to DNMT3B itself. But we do not know this for sure, unless the authors test a double mutant (PWWP inactive, no catalytic activity). I am not necessarily asking that they do it, but minimally they should mention this caveat.

4. I am confused as to why the authors look at different genomic regions in different figures. In figure 1 we are looking at a portion of the "left" arm of chr 16. But in figure 2B, we now look at a portion of the "right" arm of the same chromosome, which has a large 8-Mb block of H3K9me3 and is surprisingly lowly methylated in the 3BKO. This seems quite odd, and I wonder if there is a possible artifact, for instance mapping bias, deletion, or amplification in HCT116. Showing the coverage along with the methylation values would eliminate some of these concerns.

****Minor comments:****

1. The WGBS coverage is not very high, around 2.5X on average, occasionally 2X. I don't believe this affects the findings, as the authors look at large H3K9me3 regions. But the info in table S2 was hard to find and it is important. I would make it more

accessible.

2. It would be nice to have a drawing showing exactly what part of the Nter was removed.

3. some figures could be clearer. I was not always sure when we were looking at a CRISPR mutant clone (W263A) versus a piggyBac rescue.

4. unless I am mistaken, all the ChIP-seq data (H3K9me3, H3K36me3 etc) come from WT cells. It is not 100% certain that they remain the same in the 3BKO, is it? This should be discussed.

2. Significance:

Significance

The experiments are for the most part well done and well interpreted (save for the limitations mentioned above). The techniques are appropriate and well mastered by the team. The paper is well written, the figures are nice. The authors know the field well, which translates into a good intro and a good discussion. The bioinformatics are convincing.

All the work is done in a single cancer cell line. One might assume the conclusions will hold in other systems, but there is no certainty at this point.

HCT116 are not the best model system to study ICF, which mostly affects lymphocytes

At present, I feel that the biological relevance of the findings is fairly unclear. The authors report what happens when DNMT3B has no functional PWWP domain. I am convinced by their conclusions, but what do they tell us, biologically? Are there, for instance, mutant forms of DNMT3B expressed in disease that have a mutant PWWP? Are there isoforms expressed during development or in certain cell types that do not have a PWWP? In these cell types, does the distribution of DNA methylation agree with what the authors predict?

In its present state, I feel the appeal of the findings is towards a semi-specialized audience, that is interested in aberrant DNA methylation in cancer and other diseases. This is not a small audience, by the way.

Referee #2:

In this manuscript, Taglini et al., describe an increased activity of DNMT3B at H3K9me3-marked regions in HCT cells. They first identify that DNA methylation at K9me3-marked regions is strongly reduced in absence of DNMT3B. Next, the authors re-express DNMT3B and DNMT3B mutant variants in the DNMT3B-KO HCT cells and assess DNA methylation by WGBS where they identify a strong preference for re-methylation of K9me3 sites. Based on genome-wide binding maps for DNMT3B, including the mutant variants, they address how the localization of DNMT3B relates to the observed changes in methylation.

Major points:

The authors show increased reduction of mCG at H3K9me3 (and K27me3) sites in absence of DNMT3B. This is based on correlating delta %mCG with histone modifications in 2kb bins. I find this approach to not fully support the major claim.

First, the correlation coefficients are very small -0.124 for K9me3 and -0.175 for K27me3, and just marginally better compared to, for example, K36me3 that does not seem to have any influence on mCG according to Sup Fig S1b. While I agree that mCG seems more reduced at K9me3 in absence of DNMT3B (e.g. in Fig 1a), is there a better way to visualize the global effect? The delta mCG Boxplots based on bins are not ideal (this applies to many figures using the same approach in the current manuscript).

Second, the calculation based on delta mCpG does not allow to see how much methylation was initially there. For example, S1b shows a median decrease of ~ 10% in K9me3 and ~7-8% in H3K4me3. What does this mean given that the starting methylation for both marks is completely different?

Following this point, the authors mention that mCG is already low at K9me3 domains in HCT cells (compared to other sites in the genome). I am curious if this may influence the accelerated loss of methylation in absence of DNMT3B? Any comments on this?

One issue is the lack of correlation in DNMT3B binding to H3K9me3 sites in WT cells (Fig 3). How does this explain the requirement for DNMT3B for maintenance of methylation at H3K9me3? While some of the tested mutants show some weak increase at K9me3 sites, these are not comparable to the strong binding preferences observed at K36me3 for the wt or delta N-term version.

Following the above comment, what about other methyltransferases in HCT cells? Could DNMT1 or DNMT3A function be altered in absence of DNMT3B, and the observed methylation changes could be indirectly influenced by DNMT3B? The authors could create a DNMT-TKO HCT cell line and re-introduce DNMT3B in this background and measure methylation to exclude that

DNMT1 or DNT3A could have an influence. In this case, only H3K9me3 should gain DNA methylation.

DNMT3B lacking N-terminal shows reduced K9me3 methylation & some localization by imaging. While the presented experiments show some support for this conclusion, I suggest to re-introduce a W263A mutant lacking the N-terminal part and measure changes in DNA methylation at H3K9. This should help to test the requirement for the N-terminal regions and further indicate which protein part (PWWP or N-term) is more important in regulating the balance between K9me3 and K36me3.

In the first paragraph of the discussion, the authors state: "Our results demonstrate that DNMT3B is recruited to and methylates heterochromatin in a PWWP- independent manner that is facilitated by its N-terminal region." Same statement is found in the abstract. This contradicts the ChIP-seq results that do not indicate a recruitment of DNMT3B to heterochromatin, and the N-terminal deletions are not fully supporting a role in this targeting since there is no localization to K9me3 to begin with. While changes in methylation are observed, it remains to be determined if this is indeed through direct DNMT3B delocalization or indirectly through influencing the remaining DNMTs.

Significance

Advance: detailed analysis of DNMT3B mutants in relation to K9me3. Builds up on previous studies.

Audience: specialised audience

Referee #3:

In this work, Taglini et al. examine how the de novo DNA methyltransferase DNMT3B localizes to constitutive heterochromatin marked by the repressive histone modification H3K9me3. The authors utilize a previously generated DNMT3B KO colorectal carcinoma cell line, HCT116 to study recruitment and activity of DNMT3B at constitutive H3K9me3 heterochromatin. The authors noted preferential decrease of DNA methylation (DNAm) at regions of the genome marked with H3K9me3 in DNMT3B KOs. The authors then rescued the deficiency through overexpression of WT and catalytic dead DNMT3A/B and confirmed that DNA methylation increase at H3K9me3+ region in the WT DNMT3B, but not catalytically inactive mutant nor DNMT3A. To examine which protein domains may be mediating DNMT3B's recruitment to H3K9me3 regions, the authors designed a series of mutants, primarily focusing on the PWWP domain which normally recognizes H3K36me3. In the PWWP mutants, DNMT3B binding to the genome is altered, showing depletion at some H3K36me3-marked regions and gain at H3K9me3 heterochromatin, which coincides with DNAm increase at satellites. In contrast, the clinically relevant ICF1 mutation S270P, shows DNMT3B protein destabilization and no such loss of DNAm at heterochromatin. Finally, the authors truncate the N-terminal portion of DNMT3B, and saw that this region of the protein is necessary for heterochromatin localization and subsequent DNAm of H3K9me3+ regions.

The experiments are well done with extensive controls, and the results are interesting and convincing. The structure of the manuscript could be improved for clarity and flow - for example, the PWWP mutations and truncations should be mentioned and compared together. I also found the section on ICF1 mutant to be out-of-place. More emphasis should be placed on the N-terminal mutant as this region seems to be critical to heterochromatin recruitment, and this may address whether the interaction to H3K9me3 is direct or indirect. Finally, while the epigenetic crosstalk is well-examined in this work, I would strongly urge the authors to add RNA-seq data to determine the transcriptional consequence of such chromatin disruptions (e.g. are repetitive sequences up-regulated in DNMT3B KOs?).

Comments

1. A potential caveat to the study is the use of a single cell line - colorectal cancer cell HCT116 - to draw major conclusions on the function of DNMT3B. It is worth noting the Baubec et al. study examining DNMT3B recruitment to H3K36me3 was mainly performed in murine embryonic stem cells (mESCs). It would greatly strengthen the study if the authors could perform similar type of data analysis on an independent DNMT3B KO cell line. For example, does DNMT3B localize to H3K9me3 regions in WT mESCs?
2. Did the PWWP mutant W263A show the expected loss of DNAm at H3K36me3-marked regions? In other words, was there evidence of DNAm redistribution in loss at H3K36me3+ regions and inappropriate gain at H3K9me3+ regions? Please perform intersection analysis of DMRs with other epigenomic marks (e.g. H3K27me3, H3K36me3, CpG shores) in the PWWP mutants. The study would also be strengthened greatly with the in addition of biochemical studies to confirm direct loss of binding, and possibly gain of H3K9me3 binding, in the DNMT3B PWWP mutants.
3. Examining the tracks in Figure 3A,B, the PWWP mutants showed almost indiscriminate increase across the genome, and not specifically to H3K9me3-marked regions. Would ask the authors to speculate as to why the ChIP-seq of DNMT3B mutants do not recapitulate the heterochromatin co-localization shown by immunofluorescence.
4. It's a shame that the ICF1 mutation S270P was not characterized to the same extent as the PWWP mutants. Would consider adding WGBS for this clinically relevant mutation.
5. Figure 7 - please draw in the ICF1 and the N-terminal mutations in the model figure. Also provide legends.

Significance

This is an interesting study on a timely subject. It will be of interest to multiple fields from epigenetics to development and cancer.

Response to Reviewers

We thank the reviewers for their supportive and insightful comments that have helped us substantially improve the manuscript. In particular, in the revised manuscript we provide additional experimental work delineating the molecular mechanism through which DNMT3B is recruited to H3K9me3 nucleosomes, controls demonstrating that DNMT3B is directly responsible for methylation at H3K9me3-marked heterochromatin and replication in an additional cell line.

We provide a point-by-point response to each comment below. Reviewers' comments are indicated by *blue italic* text. Any positions referenced correspond to those in the revised document. We also enclose a revised manuscript, with changes highlighted in blue.

Reviewer #1 (Evidence, reproducibility and clarity (Required)):

Summary:

This paper by Francesca Taglini, Duncan Sproul, and their coworkers, examines the mechanisms of DNA methylation in a human cancer cell line. They use the human colorectal cancer line HCT116, which has been very widely used to look at epigenetics in cancer, and to dissect the contribution of different proteins and chromatin marks to DNA methylation.

The authors focus on the role of the de novo methyltransferase DNMT3B. It has been shown in ES cells in 2015 that its PWWP domain directs it to H3K36me3, typically found in gene bodies. More recently, the authors showed similar conclusions in colorectal cancer (Masalmeh Nat Comm 2021). Here they examine, more specifically, the role of the PWWP. The conclusions are described below.

We thank the reviewer for their comments which we feel have improved the manuscript and have modified the numbering associated with their points to facilitate cross-referencing.

Major comments:

1. I feel that this paper has several messages that are somewhat muddled. The main message, as expressed in the title and in the model, is that the PWWP domain of DNMT3B actively drags the protein to H3K36me3-marked regions. Inactivation of this domain by a point mutation, or removal of the Nter altogether, causes DNMT3B to relocate to other genomic regions that are H3K9me3-rich, and that see their DNA methylation increase in the mutant conditions. This first message is clear.

The second message has to do with ICF. A mutant form of DNMT3B bearing a mutation found in ICF, S270P, is actually unstable and, therefore, does not go to H3K9me3 regions. I feel that here the authors go on a tangent that distracts from message #1. This could be moved to the supp data. At any rate, HCT116 are not a good model for ICF. In addition, a previous paper has looked at the S270P mutant, and it did not seem unstable in their hands (Park J Mol Med 2008, PMID: 18762900). So I really feel the authors do not do themselves a favor with this ICF angle.

Previous studies have suggested that the ICF1 DNMT3B S270P mutation abrogates interaction of DNMT3B with H3K36me3 (Baubec et al 2015, <https://doi.org/10.1038/nature14176> and Rondelet et al 2016 <https://doi.org/10.1016/j.jsb.2016.03.013>) and prevents it localising to heterochromatin (Chen et al 2004, <https://doi.org/10.1128/mcb.24.20.9048-9058.2004> and Ge et al 2004, <https://doi.org/10.1074/jbc.m312296200>). Given that these results are different from those we observed for the mutations we tested within the methyl-lysine binding pocket, we felt it was important to test the S270P mutation in our assays. While ICF syndrome was not the focus of our study, we believe our result that S270P destabilises DNMT3B is important to consider in this context. However, we acknowledge the reviewer's point that the HCT116 colorectal cancer cell line is not a good model of ICF.

During the revision period we undertook experiments to express our mutations of interest in GM12878 lymphoblastoid cells which are more representative of B-cells which display clinically critical defects in ICF1. However, despite repeated attempts we were unable to achieve robust

expression of catalytically active DNMT3B in these cells during the timescale allocated for revisions (see *rebuttal figure 1*). We believe that this results from selective pressure against expressing DNMT3B in these cells as we were able to express catalytic dead DNMT3B mutants (*rebuttal figure 1*). We believe future work will be required to fully understand the implications of our work for ICF syndrome.

Rebuttal figure 1. Western blots showing expression of tagged T7-DNMT3B in GM12878 lymphoblastoid cells. GAPDH is included as a loading control.

In the revised manuscript, we have modified the discussion to reflect the caveat regarding our use of HCT116 cells:

However, we note that HCT116 are a colorectal cancer cell line and thus it is possible that the mutation behaves differently in the lymphocytes which are the primary cell type affected by ICF1 (Weemaes et al., 2013).

We have also moved the data regarding the S270P mutation to *supplementary figure 5* as suggested by the reviewer.

2. I feel that some major confounders exist that endanger the conclusions of the work. The most worrisome one, in my opinion, is the amount of WT or mutant DNMT3B in the cells. It is clear in figure 4C that the WT rescue construct is expressed much more than the W263A mutant (around 3 or 4 times more). Unless I am mistaken, we are never shown how the level of exogenous rescue protein compares to the level of DNMT3B in WT cells. This bothers me a lot. If the level is too low, we may have partial rescue. If it is too high, we might have artifactual effects of all that DNMT3B. I would also like to see the absolute DNA methylation values (determined by WGBS) compared to the value found in WT. From figure S1A, it looks like WT is around 80% methylation, and 3BKO is around 77% or so. I wonder if the rescue lines may actually have more methylation than WT?

As the reviewer suggests, the rescue cell lines do express DNMT3B to a greater level than observed endogenously. We have controlled for this effect by generating the genomic knock-in W263A cells and, as reported in the manuscript, we observe similar effects to the rescue cells (*figure 2d*). This suggests that our observation that the W263A mutation causes hypermethylation of H3K9me3-marked heterochromatin is not simply caused by overexpression.

To further address this, we have included an experiment where ectopic DNMT3B is expressed from the EF1 α promoter. We have previously shown that the EF1 α promoter results in lower levels of DNMT3B expression than the CAG promoter used in our other experiments (Masalmeh et al 2021, <https://doi.org/10.1038/s41467-020-20716-w>). Bisulfite PCR of representative non-repetitive loci within heterochromatic H3K9me3 domains show that we observe similar gains of methylation when DNMT3B^{W263A} is expressed from EF1 α (*figure S2m*).

The reviewer is also correct that by WGBS, the rescue cell lines have higher levels of overall DNA methylation than HCT116 cells. As suggested, we have included the mean level of methylation in HCT116 cells in a revised version of *figure S2e*.

3. I guess the unarticulated assumption is that the gain of DNA methylation seen at H3K9me3 region upon expression of a mutant DNMT3B is due to DNMT3B itself. But we do not know this for sure, unless the authors test a double mutant (PWWP inactive, no catalytic activity). I am not necessarily asking that they do it, but minimally they should mention this caveat.

We acknowledge that our original manuscript does not formally show that the additional hypermethylation observed in DNMT3B^{W263A} cells was due to DNMT3B's catalytic activity. As suggested, we have now added an experiment demonstrating that expression of catalytically dead DNMT3B^{W263A} does not result in gains of methylation at Satellite II sequences and representative H3K9me3 loci. These data are included in *figure S2h-k*.

4. I am confused as to why the authors look at different genomic regions in different figures. In figure 1 we are looking at a portion of the "left" arm of chr 16. But in figure 2B, we now look at a portion of the "right" arm of the same chromosome, which has a large 8-Mb block of H3K9me3, and is surprisingly lowly methylated in the 3BKO. This seems quite odd, and I wonder if there is a possible artifact, for instance mapping bias, deletion, or amplification in HCT116. Showing the coverage along with the methylation values would eliminate some of these concerns.

By choosing different regions of the genome for different figures, we intended to reassure the reader that our results were not specific to any one region of the genome. As the reviewer suggests, we have now updated *figures 1a, 1d, 2b, 5a and s1g* to show the same genomic region.

With regard to the low levels of DNA methylation observed in H3K9me3 domains in DNMT3B KO cells, H3K9me3 domains are partially methylated domains which have reduced methylation in HCT116 cells (*see page 5 of the manuscript*):

... we found that hidden Markov model defined H3K9me3 domains significantly overlapped with PMDs defined in our HCT116 WGBS (Jaccard=0.575, $p=1.07 \times 10^{-6}$, Fisher's test).

These domains lose further DNA methylation in DNMT3B KO cells leading to the low methylation level noted by the reviewer. The methylation percentages calculated from WGBS are based on the ratio of methylated to total reads. Thus, a lack of coverage generates errors from division by zero rather than the low values observed in this domain in DNMT3B KO cells.

To reassure the reviewer we include a modified version of the original *figure 2b* below (*rebuttal figure 2*). This includes coverage for the 3 cell lines. Although we observe WGBS coverage is slightly reduced in H3K9me3 domains, reads are still present and overall coverage is approximately equal between different cell lines.

Rebuttal figure 2. Representative genomic region showing WGBS coverage. Genome browser plots showing absolute (black) and differential (gain=red/loss=blue) DNA methylation levels, WGBS coverage (CpG coverage, grey), DNMT3B KO ChIP-seq signals at H3K9me3 domains in DNMT3B KO, DNMT3BWT and DNMT3B^{W263A} cells. ChIP-seq are normalised reads per 10⁶.

To avoid over cluttering figure panels that already include many tracks and to prevent the need to shrink the existing tracks obscuring the data, we have not included this coverage information in the genome browser figures in the manuscript.

Minor comments:

5. The WGBS coverage is not very high, around 2.5X on average, occasionally 2X. I don't believe this affects the findings, as the authors look at large H3K9me3 regions. But the info in table S2 was hard to find and it is important. I would make it more accessible.

We have added this to the revised manuscript:

To understand the role of DNMT3B in methylating different parts of the genome we examined DNMT3B knockout HCT116 cells (DNMT3B KO) (Rhee et al., 2002) using whole genome bisulfite sequencing (WGBS, mean autosomal CpG coverage 2.6, see supplementary table 2).

6. It would be nice to have a drawing showing exactly what part of the Nter was removed.

The revised manuscript includes this in *figure S7a*.

7. some figures could be clearer. I was not always sure when we were looking at a CRISPR mutant clone (W263A) versus a piggyBac rescue.

We have modified *figures 2* and *s2-3* to ensure this is clearer.

8. unless I am mistaken, all the ChIP-seq data (H3K9me3, H3K36me3 etc) come from WT cells. It is not 100% certain that they remain the same in the 3BKO, is it? This should be discussed.

We performed ChIP-seq on both HCT116 and DNMT3B KO cell lines. ChIP-seq data from the 3BKO cell line was used for the rescue experiments where DNMT3B and its mutated versions were expressed in DNMT3B KO cells.

Reviewer #1 (Significance (Required)):

Strengths:

9. The experiments are for the most part well done and well interpreted (save for the limitations mentioned above). The techniques are appropriate and well mastered by the team. The paper is well written, the figures are nice. The authors know the field well, which translates into a good intro and a good discussion. The bioinformatics are convincing.

We thank the reviewer for their kind comments.

Limitations:

10. All the work is done in a single cancer cell line. One might assume the conclusions will hold in other systems, but there is no certainty at this point.

In the revised manuscript we have incorporated data showing that DNMT3B^{W263A} also causes hypermethylation of satellite II in RKO cells (*figure S3c, d*).

11. HCT116 are not the best model system to study ICF, which mostly affects lymphocytes.

As discussed in response to point 2 we have revised the manuscript in accordance with this point.

12. At present, I feel that the biological relevance of the findings is fairly unclear. The authors report what happens when DNMT3B has no functional PWWP domain. I am convinced by their conclusions, but what do they tell us, biologically? Are there, for instance, mutant forms of DNMT3B expressed in disease that have a mutant PWWP? Are there isoforms expressed during development or in certain cell types that do not have a PWWP? In these cell types, does the distribution of DNA methylation agree with what the authors predict?

The delineation of the molecular mechanisms through which epigenetic marks are distributed in the genome remains a central fundamental question in understanding gene regulation and genome stability in biological systems. While the mutations we have analysed have mostly not currently been observed in human diseases we believe biological relevance of our findings is very high. In the revised manuscript we have included an extended discussion of the potential biological implications of our findings:

Loss of DNA methylation from heterochromatic loci is a hallmark of cancer that results in the formation of PMDs (Berman et al., 2011, Zhou et al., 2018) and is thought to lead to activation of retrotransposons and genome instability (Du et al., 2019, Zhou et al., 2018). Our observation that DNMT3B plays a role in DNA methylation homeostasis in PMDs suggest that modulation of its activity could play a role in this hypomethylation. Our results here and those of our previous study of CpG islands (Masalmeh et al., 2021) suggest that, similarly to normal cells, the majority of DNMT3B in cancer cells localises and is active at H3K36me3-marked regions. However, here we show that this balance of DNMT3B activity in cells can be shifted by mutation or removal of the PWWP domain to cause increased methylation at H3K9me3-marked regions. Many different DNMT3B splice isoforms have been reported to be aberrantly expressed in cancer (Gopalakrishnan et al., 2009, Ostler et al., 2007). These include Δ DNMT3B4 which lacks the PWWP domain and a portion of the N-terminal region (Wang et al., 2006b) and is reported to be expressed in non-small cell lung cancer (Wang et al., 2006a). It is also possible that this balance between recruitment of DNMT3B to H3K9me3 and H3K36me3 is regulated by cellular signalling pathways which could be altered in carcinogenesis.

In its present state, I feel the appeal of the findings is towards a semi-specialized audience, that is interested in aberrant DNA methylation in cancer and other diseases. This is not a small audience, by the way.

We thank the reviewer for their comments.

Reviewer #2 (Evidence, reproducibility and clarity (Required)):

In this manuscript, Taglini et al., describe an increased activity of DNMT3B at H3K9me3-marked regions in HCT cells. They first identify that DNA methylation at K9me3-marked regions is strongly reduced in absence of DNMT3B. Next, the authors re-express DNMT3B and DNMT3B mutant variants in the DNMT3B-KO HCT cells and assess DNA methylation by WGBS where they identify a strong preference for re-methylation of K9me3 sites. Based on genome-wide binding maps for DNMT3B, including the mutant variants, they address how the localization of DNMT3B relates to the observed changes in methylation.

We thank the reviewer for their comments which we believe have improved the manuscript. To facilitate cross referencing, we have numbered the comments.

Major points:

1. The authors show increased reduction of mCG at H3K9me3 (and K27me3) sites in absence of DNMT3B. This is based on correlating delta %mCG with histone modifications in 2kb bins. I find this approach to not fully support the major claim.

First, the correlation coefficients are very small -0.124 for K9me3 and -0.175 for K27me3, and just marginally better compared to, for example, K36me3 that does not seem to have any influence on mCG according to Sup Fig S1b. While I agree that mCG seems more reduced at K9me3 in absence of DNMT3B (e.g. in Fig 1a), is there a better way to visualize the global effect? The delta mCG Boxplots based on bins are not ideal (this applies to many figures using the same approach in the current manuscript).

Our choice to examine the global effects using correlations in windows across the genome was motivated by similar analyses in previous studies (for example: Baubec et al. 2015 <https://doi.org/10.1038/nature14176>, Weinberg et al 2021 <https://doi.org/10.1038/s41588-021-00856-5>, Neri et al 2017, <https://doi.org/10.1038/nature21373>). These global analyses result in modest correlation coefficients because the vast majority of genomic windows are negative for a given mark. However, the results are highly statistically significant. To verify the results of these global analyses, we have included specific analyses of H3K36me3, H3K9me3 and H3K27me3 domains in the manuscript (eg figures 1b, c, e, s1c).

2. Second, the calculation based on delta mCpG does not allow to see how much methylation was initially there. For example, S1b shows a median decrease of ~ 10% in K9me3 and ~7-8% in H3K4me3. What does this mean given that the starting methylation for both marks is completely different?

Following this point, the authors mention that mCG is already low at K9me3 domains in HCT cells (compared to other sites in the genome). I am curious if this may influence the accelerated loss of methylation in absence of DNMT3B? Any comments on this?

The observation that there is a greater loss at H3K9me3 domains than H3K27me3-only domains (figure 1c) which also have low DNA methylation levels in HCT116 argue that the losses are not solely driven by the lower initial level of methylation in H3K9me3 domains. Our analyses later in the manuscript also support a specific activity of DNMT3B at H3K9me3.

To reinforce this point, we have analysed the correlation between loss of methylation in DNMT3B KO cells and the initial levels of methylation in HCT116 cells across the whole genome. This reveals that there is a very low correlation between these two (rebuttal figure 3).

Rebuttal figure 3. A low level of correlation is observed between loss of methylation in DNMT3B KO cells and mean methylation levels in HCT116 cells. Boxplot showing the difference in DNA methylation in DNMT3B KO to HCT116 cells at 2.5kb genomic windows divided into deciles according to their level of methylation in HCT116 cells. The Pearson's correlation coefficient (R) is shown alongside its associated p-value. Lines = median; box = 25th-75th percentile; whiskers = 1.5x interquartile range.

3. One issue is the lack of correlation in DNMT3B binding to H3K9me3 sites in WT cells (Fig 3). How does this explain the requirement for DNMT3B for maintenance of methylation at H3K9me3? While some of the tested mutants show some weak increase at K9me3 sites, these are not comparable to the strong binding preferences observed at K36me3 for the wt or delta N- term version.

There are several experimental reasons why the apparent enrichment of DNMT3B at H3K9me3 may appear weaker than at H3K36me3 by ChIP-seq. We have added discussion of these points to the revised manuscript:

Here we observe that the enrichment of DNMT3B at H3K9me3-marked heterochromatin by ChIP-seq is less distinct than at H3K36me3 but that heterochromatic localisation is more apparent by microscopy. Previous work has suggested that formaldehyde crosslinking fails to capture transient interactions in cells (Schmiedeberg et al., 2009). Such an effect could affect our ability to detect DNMT3B at H3K9me3 by ChIP-seq. H3K9me3-marked heterochromatin is also resistant to sonication (Becker et al., 2017) and this could decrease DNMT3B ChIP-seq signal at H3K9me3. Furthermore, our data also suggests that DNMT3B's interaction with H3K9me3-modified nucleosomes is indirect and mediated by HP1 α . This may also lead to weaker ChIP-seq enrichment at H3K9me3 compared to the direct interaction with H3K36me3 through DNMT3B's PWWP domain.

Additional data provided in the revised manuscript demonstrate that the N-terminal region of DNMT3B directly interacts with HP1 α and that this facilitates interaction with H3K9me3-marked nucleosomes (included in a new section *DNMT3B's N-terminal region interacts with HP1 α* and figure 6). In combination with our analysis by microscopy in live cells, these orthogonal data reinforce our

conclusion that DNMT3B has specific activity at H3K9me3-marked heterochromatin and delineate the molecular mechanism behind this association.

4. Following the above comment, what about other methyltransferases in HCT cells? Could DNMT1 or DNMT3A function be altered in absence of DNMT3B, and the observed methylation changes could be indirectly influenced by DNMT3B? The authors could create a DNMT-TKO HCT cell line and re-introduce DNMT3B in this background and measure methylation to exclude that DNMT1 or DNMT3A could have an influence. In this case, only H3K9me3 should gain DNA methylation.

This is a good point and in the revised manuscript, we include analysis of catalytically dead DNMT3B^{W263A} (see also response to reviewer 1, point 4). This experiment reinforces the conclusion that the methylation at H3K9me3 we observe is a direct consequence of the catalytic activity of DNMT3B^{W263A}.

5. DNMT3B lacking N-terminal shows reduced K9me3 methylation & some localization by imaging. While the presented experiments show some support for this conclusion, I suggest to re-introduce a W263A mutant lacking the N-terminal part and measure changes in DNA methylation at H3K9. This should help to test the requirement for the N-terminal regions and further indicate which protein part (PWWP or N-term) is more important in regulating the balance between K9me3 and K36me3.

We have performed this experiment and the data are shown in manuscript *figure S7d* and *e*. The data show that DNMT3B^{ΔN W263A} cells showed less methylation at H3K9me3 loci than DNMT3B^{W263A} cells, supporting a role for the N-terminal region in recruiting DNMT3B to H3K9me3-marked heterochromatin.

6. In the first paragraph of the discussion, the authors state: "Our results demonstrate that DNMT3B is recruited to and methylates heterochromatin in a PWWP- independent manner that is facilitated by its N-terminal region." Same statement is found in the abstract. This contradicts the ChIP-seq results that do not indicate a recruitment of DNMT3B to heterochromatin, and the N-terminal deletions are not fully supporting a role in this targeting since there is no localization to K9me3 to begin with. While changes in methylation are observed, it remains to be determined if this is indeed through direct DNMT3B delocalization or indirectly through influencing the remaining DNMTs.

To bolster our conclusions, that DNMT3B localises to H3K9me3 marked regions, we reconstituted interaction of DNMT3B with H3K9me3 nucleosomes and found that this interaction is stimulated by bridging HP1. We further mapped the interaction of DNMT3B with HP1 α to the N-terminal region (new *figure 6*). Overall these mechanistic experiments combined with the localisation by microscopy is consistent with DNMT3B being driven to H3K9me3-marked heterochromatin. The specific activity of DNMT3B at H3K9me3 marked heterochromatin is also supported by our experiments with catalytic dead mutants as described above (point 4).

Reviewer #2 (Significance (Required)):

*Advance: detailed analysis of DNMT3B mutants in relation to K9me3. Builds up on previous studies.
Audience: specialised audience*

Reviewer #3 (Evidence, reproducibility and clarity (Required)):

In this work, Taglini et al. examine how the de novo DNA methyltransferase DNMT3B localizes to constitutive heterochromatin marked by the repressive histone modification H3K9me3. The authors utilize a previously generated DNMT3B KO colorectal carcinoma cell line, HCT116 to study recruitment and activity of DNMT3B at constitutive H3K9me3 heterochromatin. The authors noted preferential decrease of DNA methylation (DNAm) at regions of the genome marked with H3K9me3 in DNMT3B KOs. The authors then rescued the deficiency through overexpression of WT and catalytic dead DNMT3A/B and confirmed that DNA methylation increase at H3K9me3+ region in the WT DNMT3B, but not catalytically inactive mutant nor DNMT3A. To examine which protein domains may be mediating DNMT3B's recruitment to H3K9me3 regions, the authors designed a series of mutants,

primarily focusing on the PWWP domain which normally recognizes H3K36me3. In the PWWP mutants, DNMT3B binding to the genome is altered, showing depletion at some H3K36me3-marked regions and gain at H3K9me3 heterochromatin, which coincides with DNAm increase at satellites. In contrast, the clinically relevant ICF1 mutation S270P, shows DNMT3B protein destabilization and no such loss of DNAm at heterochromatin. Finally, the authors truncate the N-terminal portion of DNMT3B, and saw that this region of the protein is necessary for heterochromatin localization and subsequent DNAm of H3K9me3+ regions.

The experiments are well done with extensive controls, and the results are interesting and convincing. The structure of the manuscript could be improved for clarity and flow - for example, the PWWP mutations and truncations should be mentioned and compared together. I also found the section on ICF1 mutant to be out-of-place. More emphasis should be placed on the N-terminal mutant as this region seems to be critical to heterochromatin recruitment, and this may address whether the interaction to H3K9me3 is direct or indirect. Finally, while the epigenetic crosstalk is well-examined in this work, I would strongly urge the authors to add RNA-seq data to determine the transcriptional consequence of such chromatin disruptions (e.g. are repetitive sequences up-regulated in DNMT3B KOs?).

We thank the reviewer for their comments which we feel have improved the manuscript.

We have included an RNA-seq analysis in the revised manuscript showing that H3K9me3-marked regions are significantly enriched in differentially expressed genes between HCT116 and DNMT3B KO cells (*figure s1i*). However, the effect is modest suggesting that other repressive mechanisms prevent wholesale changes in gene expression in these regions upon loss of methylation.

We agree with the reviewer regarding the data on the S270P and as described in response to reviewer 1 (*point 2*), we have moved these data to the supplementary material and included discussion of the caveats of HCT116 cells as a model of ICF1.

Comments

1. A potential caveat to the study is the use of a single cell line - colorectal cancer cell HCT116 - to draw major conclusions on the function of DNMT3B. It is worth noting the Baubec et al. study examining DNMT3B recruitment to H3K36me3 was mainly performed in murine embryonic stem cells (mESCs). It would greatly strengthen the study if the authors could perform similar type of data analysis on an independent DNMT3B KO cell line. For example, does DNMT3B localize to H3K9me3 regions in WT mESCs?

As suggested by the reviewer, we now provide data showing that the expression of DNMT3B^{W263A} in RKO cells also results in hypermethylation of constitutive satellite II heterochromatin (*figure s3c, d*).

2. Did the PWWP mutant W263A show the expected loss of DNAm at H3K36me3-marked regions? In other words, was there evidence of DNAm redistribution in loss at H3K36me3+ regions and inappropriate gain at H3K9me3+ regions? Please perform intersection analysis of DMRs with other epigenomic marks (e.g. H3K27me3, H3K36me3, CpG shores) in the PWWP mutants.

As suggested by the reviewer, we have analysed DNA methylation levels at gene bodies stratified by H3K36me3 level in the knock-in DNMT3B^{W263A} cells. This analysis shows that losses of DNA methylation have only a weak correlation with H3K36me3 levels in these cells (*figure s3b*). Like our results in DNMT3B KO cells (*figure s1c*), this suggests that other DNMTs are able to maintain methylation at H3K36me3 marked loci. Due to the low coverage of our WGBS, we were not able to undertake a robust analysis of DMRs in these cells.

The study would also be strengthened greatly with the in addition of biochemical studies to confirm direct loss of binding, and possibly gain of H3K9me3 binding, in the DNMT3B PWWP mutants.

We have now added biochemical and cellular data elucidating the molecular mechanism through which DNMT3B is recruited to H3K9me3-marked chromatin (new results section *DNMT3B's N-*

terminal region interacts with HP1 α and figures 6, s8). We show that DNMT3B does not have a direct preferential interaction with H3K9me3 peptides or defined modified nucleosomes. Instead, we find that the N-terminal region of DNMT3B is necessary and sufficient for direct interaction with HP1 α . Furthermore, we demonstrate that HP1 α greatly strengthens DNMT3B interaction with H3K9me3-modified synthetic nucleosomes *in vitro*. Overall, we propose that DNMT3B recruitment to heterochromatin is mediated as part of a ternary complex between DNMT3B, HP1 α and H3K9me3-modified nucleosomes.

During the period allocated for revising the manuscript, we were not able to set up a robust biochemical assay to detect the interaction between DNMT3B's PWWP domain and H3K36me3. However, the hypothesis that mutations affecting H3K36me3 binding result in hypermethylation of H3K9me3-marked heterochromatin is supported by our analysis of multiple mutations, removal of the PWWP domain and the ChIP-seq data. We have noted the caveat that we did not directly demonstrate loss of H3K36me3 binding in the discussion:

We note that here we have not directly demonstrated that the mutations studied affect the interaction between DNMT3B's PWWP and H3K36me3.

3. Examining the tracks in Figure 3A,B, the PWWP mutants showed almost indiscriminate increase across the genome, and not specifically to H3K9me3-marked regions. Would ask the authors to speculate as to why the ChIP-seq of DNMT3B mutants do not recapitulate the heterochromatin co-localization shown by immunofluorescence.

As suggested by the reviewer, we have included discussion of this point in the revised manuscript (see also reviewer 2, point 3):

Here we observe that the enrichment of DNMT3B at H3K9me3-marked heterochromatin by ChIP-seq is less distinct than at H3K36me3 but that heterochromatic localisation is more apparent by microscopy. Previous work has suggested that formaldehyde crosslinking fails to capture transient interactions in cells (Schmiedeberg et al., 2009). Such an effect could affect our ability to detect DNMT3B at H3K9me3 by ChIP-seq. H3K9me3-marked heterochromatin is also resistant to sonication (Becker et al., 2017) and this could decrease DNMT3B ChIP-seq signal at H3K9me3. Furthermore, our data also suggests that DNMT3B's interaction with H3K9me3-modified nucleosomes is indirect and mediated by HP1 α . This may also lead to weaker ChIP-seq enrichment at H3K9me3 compared to the direct interaction with H3K36me3 through DNMT3B's PWWP domain.

4. It's a shame that the ICF1 mutation S270P was not characterized to the same extent as the PWWP mutants. Would consider adding WGBS for this clinically relevant mutation.

We have shown that this mutant does not produce stable protein *in vitro* or in our cells and we observe little difference in DNA methylation at selected loci. As WGBS is expensive, we believe that carrying out this experiment is not an efficient use of limited research resources.

5. Figure 7 - please draw in the ICF1 and the N-terminal mutations in the model figure. Also provide legends.

We have included the effect of the N-terminal mutation in *figure 7*. Given the reviewer suggestions to de-emphasise the data on the S270P mutation, we have not included this mutation in the figure.

Reviewer #3 (Significance (Required)):

This is an interesting study on a timely subject. It will be of interest to multiple fields from epigenetics to development and cancer. My expertise is in cancer, epigenetics, development.

We thank the reviewer for their comments.

Dear Dr. Sproul,

Thank you for the submission of your revised manuscript to our editorial offices. I have now received the reports from the three referees that I asked to re-evaluate your study, you will find below. As you will see, the referees now fully support the publication of the study in EMBO reports. Referee #2 has some remaining comments and suggestions to improve the manuscript, I ask you to address in a final revised manuscript. Please also provide a final p-b-p-response regarding these points.

During our standard image analysis, we detected potential aberrations in the figure set, and we would like to clarify these issues. The first three heatmaps in Fig. 2C and Fig. 5C look very similar or have identical features, but from the legend and the labeling above the maps it remains unclear if here partly identical data is shown. Please check and comment on the perceived reuse of the image. If purposeful re-use of an image has occurred, please state this clearly in the figure legend. If you make changes to the figure set, we require a further response describing what you have changed and why.

- We updated our journal's competing interests policy in January 2022 and request authors to consider both actual and perceived competing interests. Please review the policy <https://www.embopress.org/competing-interests> and update your competing interests if necessary. Please name this section 'Disclosure and Competing Interests Statement' and put it after the Acknowledgements section.
- We now use CRediT to specify the contributions of each author in the journal submission system. CRediT replaces the author contribution section. Please use the free text box to provide more detailed descriptions and do not provide your final manuscript text file with an author contributions section. See also our guide to authors: <https://www.embopress.org/page/journal/14693178/authorguide#authorshipguidelines>
- Please move all the funding information to the acknowledgements section.
- Please also make sure that all the funding information is also entered into the online submission system and that it is complete and similar to the one in the acknowledgement section of the manuscript text file. Presently, University of Edinburgh, the Edinburgh Protein Production Facility (EPPF), CRUK (C157/A25186), A. G. Leventis Foundation (18736), the Wellcome Trust (228154/Z/23/Z), ERASMUS+ scholarship, the Darwin Trust of Edinburgh are only mentioned in the manuscript text file.
- Please move the keywords (there is room for 2 more keywords) below the abstract and order the manuscript sections like this, using these names:
Title page - Abstract - Keywords - Introduction - Results - Discussion - Materials and Methods - Data availability section - Acknowledgements - Disclosure and Competing Interests Statement - References - Figure legends - Expanded View Figure legends
- Please remove the referee tokens from the data availability section and make sure that the datasets are public latest when the paper is published online.
- Please make sure that the number "n" for how many independent experiments were performed, their nature (biological versus technical replicates), the bars and error bars (e.g. SEM, SD) and the test used to calculate p-values is indicated in the respective figure legends (for main, EV and Appendix figures) of the final revised manuscript. Please also check that all the p-values are explained in the legend, and that these fit to those shown in the figure. Please provide statistical testing where applicable. Please avoid the phrase 'independent experiment', but clearly state if these were biological or technical replicates. Please also indicate (e.g. with n.s.) if testing was performed, but the differences are not significant. In case n=2, please show the data as separate datapoints without error bars and statistics. See also: <http://www.embopress.org/page/journal/14693178/authorguide#statisticalanalysis>

If n<5, please show single datapoints for diagrams. In particular:

- Information related to n is missing in the legends of figures 1c, e; 2c, d; 3c, d, e; 4d; 5b-e
- Error bars are not defined in the legend of figure 4d.
- I would suggest mentioning the p-values for all the statistical tests only in the figure legends (main, EV and Appendix figures) and mark these in the diagrams using asterisks (*, **, *** etc. ...).
- Please format the figure legends according to our journal style. See the respective section in our guide to authors (please find the link below). Please separate each panel description by a line break and make sure that the panels are listed in alphabetic order. Moreover, please add to each legend a 'Data Information' section explaining the statistics used or providing information regarding replicates and scales.

- Please provide a properly formatted Appendix file. The Appendix should have page numbers and needs to include a table of content on the first page (with page numbers) and legends for all content. Please follow the nomenclature Appendix Figure Sx, Appendix Table Sx etc. for callouts throughout the text, and also label the figures and tables in the Appendix according to this nomenclature.

- Please use our reference style (et al needs to be used after 10th authors name, year should be in brackets):
<http://www.embopress.org/page/journal/14693178/authorguide#referencesformat>

In addition, I would need from you:

Please use this link to submit your revision: <https://embor.msubmit.net/cgi-bin/main.plex>

Best,

Referee #1:

I have reviewed a previous version of this manuscript for Review Commons. The authors have addressed my main concerns satisfactorily.

Referee #2:

The authors have added a great number of results to provide additional support for the proposed model. I have a few minor comments below that should help to improve the final version. Otherwise, I don't have anything else to add, than to congratulate the authors on these results.

Regarding the comment on the global analysis using correlations based on 2kb bins. There is nothing wrong with using this approach, but the way how the data is currently represented (boxplots in Fig S1b) is not very helpful. It's not clear if each box contains equal number of bins ranked by enrichment or grouped according to histone modification enrichment range. It would be much better to directly show scatterplots showing the histone mod enrichment versus delta mCG for each individual data point (2kb bin). Clearly, the statistical significance will always be high at this amount of datapoints, therefore I don't think this adds any meaning to the interpretation of the data.

The HP1a - N-term interaction experiments are very interesting and could point to a mechanism that explains the observed increase of mCG at H3K9me3. The results presented are already well executed and convincing. I think it would be even stronger if the authors could remove HP1a in cells and measure DNA methylation, to test if they observe a similar reduction at H3K9me3, as seen in D3B KOs.

Referee #3:

The authors addressed my comments.

Thank you for the submission of your revised manuscript to our editorial offices. I have now received the reports from the three referees that I asked to re-evaluate your study, you will find below. As you will see, the referees now fully support the publication of the study in EMBO reports. Referee #2 has some remaining comments and suggestions to improve the manuscript, I ask you to address in a final revised manuscript. Please also provide a final p-b-p-response regarding these points.

Thank you for the additional comments, we have included a point by response to both the reviewer and editorial comments below. For clarity, comments are in blue text.

During our standard image analysis, we detected potential aberrations in the figure set, and we would like to clarify these issues. The first three heatmaps in Fig. 2C and Fig. 5C look very similar or have identical features, but from the legend and the labeling above the maps it remains unclear if here partly identical data is shown. Please check and comment on the perceived reuse of the image. If purposeful re-use of an image has occurred, please state this clearly in the figure legend. If you make changes to the figure set, we require a further response describing what you have changed and why.

The majority of the data shown in the heatmaps included as Fig 2c and Fig 5c are indeed different. The heatmap headed 3BKO H3K9me3 is shared between the two figure panels as both figures use the same DNMT3B KO H3K9me3 ChIP-seq.

The other panels are derived from whole genome bisulfite sequencing and the data are entirely independent. Both experiments feature DNMT3B KO cells and DNMT3B^{WT} cells but these were generated on separate days for comparison with the data derived from DNMT3B^{W263A} and DNMT3B^{ΔN} cells examined in Fig 2c and Fig 5c respectively. The data for DNMT3B KO and DNMT3B^{WT} cells for Fig 2c can be found on NCBI GEO as samples GSM7818547 and GSM7818548. The corresponding data for Fig 5C are samples GSM7818559 and GSM7818560/GSM7818562. The data in Fig 5c for DNMT3B^{WT} and DNMT3B^{ΔN} cells are the mean of two biological replicates as stated in the figure legend.

We have amended the figure legend to make it clear that Fig 5c shows the same DNMT3B KO H3K9me3 ChIP-seq data as Fig 2c.

- We updated our journal's competing interests policy in January 2022 and request authors to consider both actual and perceived competing interests. Please review the policy <https://www.embopress.org/competing-interests> and update your competing interests if necessary. Please name this section 'Disclosure and Competing Interests Statement' and put it after the Acknowledgements section.

We have reviewed the guidelines and updated the wording and placement of this statement in accordance with this request.

- We now use CRediT to specify the contributions of each author in the journal submission system. CRediT replaces the author contribution section. Please use the free text box to

provide more detailed descriptions and do not provide your final manuscript text file with an author contributions section. See also our guide to authors:
<https://www.embopress.org/page/journal/14693178/authorguide#authorshipguidelines>

We have used the CRediT system to specify each author's contribution in the manuscript submission system and removed the contributions from the manuscript as requested.

- Please move all the funding information to the acknowledgements section.

We have moved the funding statement to the acknowledgements section.

- Please also make sure that all the funding information is also entered into the online submission system and that it is complete and similar to the one in the acknowledgement section of the manuscript text file. Presently, University of Edinburgh, the Edinburgh Protein Production Facility (EPPF), CRUK (C157/A25186), A. G. Leventis Foundation (18736), the Wellcome Trust (228154/Z/23/Z), ERASMUS+ scholarship, the Darwin Trust of Edinburgh are only mentioned in the manuscript text file.

We have included these funders in the online submission statement with grant numbers where these are available. Note the Wellcome trust grant funding the Edinburgh Protein Production Facility is already included on the submission system (203149).

- Please move the keywords (there is room for 2 more keywords) below the abstract and order the manuscript sections like this, using these names:
Title page - Abstract - Keywords - Introduction - Results - Discussion - Materials and Methods - Data availability section - Acknowledgements - Disclosure and Competing Interests Statement - References - Figure legends - Expanded View Figure legends

We have reordered and renamed the sections in accordance with these recommendations.

- Please remove the referee tokens from the data availability section and make sure that the datasets are public latest when the paper is published online.

We have removed the reviewer tokens and will inform the repositories to make the data available once we have notification of the online publication date.

We have also uploaded our imaging data to EBI's Bioimage Archive repository and provide links for this data in the revised manuscript as suggested by the editorial team upon our last submission. Details are provided in the data availability section.

- Please make sure that the number "n" for how many independent experiments were performed, their nature (biological versus technical replicates), the bars and error bars (e.g. SEM, SD) and the test used to calculate p-values is indicated in the respective figure legends (for main, EV and Appendix figures) of the final revised manuscript. Please also check that all the p-values are explained in the legend, and that these fit to those shown in the figure. Please provide statistical testing where applicable. Please avoid the phrase 'independent experiment', but clearly state if these were biological or technical replicates. Please also

indicate (e.g. with n.s.) if testing was performed, but the differences are not significant. In case n=2, please show the data as separate datapoints without error bars and statistics. See also:

<http://www.embopress.org/page/journal/14693178/authorguide#statisticalanalysis>

If n<5, please show single datapoints for diagrams. In particular:

- Information related to n is missing in the legends of figures 1c, e; 2c, d; 3c, d, e; 4d; 5b-e
- Error bars are not defined in the legend of figure 4d.

We have included this information in the figure legends.

- I would suggest mentioning the p-values for all the statistical tests only in the figure legends (main, EV and Appendix figures) and mark these in the diagrams using asterisks (*, **, *** etc. ...).

For clarity we have kept the p-values in the figure panels. There are multiple tests in many figures and it would become very unclear which p-value corresponded to which comparison if they were moved to the figure legends.

- Please format the figure legends according to our journal style. See the respective section in our guide to authors (please find the link below). Please separate each panel description by a line brake and make sure that the panels are listed in alphabetic order. Moreover, please add to each legend a 'Data Information' section explaining the statistics used or providing information regarding replicates and scales.

We have formatted the figure legends in accordance with these guidelines.

- Please provide a properly formatted Appendix file. The Appendix should have page numbers and needs to include a table of content on the first page (with page numbers) and legends for all content. Please follow the nomenclature Appendix Figure Sx, Appendix Table Sx etc. for callouts throughout the text, and also label the figures and tables in the Appendix according to this nomenclature.

We have provided an appendix file formatted as requested and amended all callouts in the text.

- Please use our reference style (et al needs to be used after 10th authors name, year should be in brackets):

We have downloaded and used the EMBO reports citation style.

In addition, I would need from you:

- a short, two-sentence summary of the manuscript (not more than 35 words).

The DNA methyltransferase DNMT3B is recruited to gene bodies through its PWWP domain and H3K36me3. This study reveals that DNMT3B PWWP mutations can cause hypermethylation of H3K9me3-marked heterochromatin that is facilitated by DNMT3B's N-terminal region.

- two to four short (!) bullet points highlighting the key findings of your study (two lines each).

- Removal of DNMT3B results in losses of DNA methylation from H3K9me3 methylated heterochromatin.
- Mutation or deletion of DNMT3B's PWWP results in the hypermethylation of H3K9me3-marked heterochromatin.
- Recruitment of DNMT3B to H3K9me3 is facilitated by its N-terminal region which interacts with HP1 α .

- a schematic summary figure as separate file that provides a sketch of the major findings (not a data image) in jpeg or tiff format (with the exact width of 550 pixels and a height of not more than 400 pixels) that can be used as a visual synopsis on our website.

We have included a figure with the submission (*Taglini_et_al_summary_figure_201223.tiff*).

Thank you, during the re-submission, I was unable to find out how to associate all authors with their ORCID IDs on the journal website, these are included in the table below for inclusion:

Francesca Taglini	none
Ioannis Kafetzopoulos	0000-0001-5948-3455
Willow Rolls	0000-0002-2697-4212
Kamila Irena Musialik	0000-0003-2543-5011
Heng Yang Lee	0000-0001-8006-5194
Yujie Zhang	none
Mattia Marenda	0000-0001-5951-3212
Lyndsay Kerr	0000-0002-6667-7175
Hannah Finan	None
Cristina Rubio-Ramon	None
Philippe Gautier	0000-0003-3019-6262
Hannah Wapenaar	0009-0003-8790-4511
Dhananjay Kumar	0000-0003-0557-0603
Hazel Davidson-Smith	none
Jimi Wills	0000-0003-1669-007X
Laura C. Murphy	none
Ann Wheeler	0000-0001-8617-827X

Marcus D. Wilson	0000-0001-9551-5514
Duncan Sproul	0000-0001-6168-4563

Reviewer's comments

Referee #1:

I have reviewed a previous version of this manuscript for Review Commons. The authors have addressed my main concerns satisfactorily.

We thank the reviewer for their comments.

Referee #2:

The authors have added a great number of results to provide additional support for the proposed model. I have a few minor comments below that should help to improve the final version. Otherwise, I don't have anything else to add, than to congratulate the authors on these results.

Regarding the comment on the global analysis using correlations based on 2kb bins. There is nothing wrong with using this approach, but the way how the data is currently represented (boxplots in Fig S1b) is not very helpful. It's not clear if each box contains equal number of bins ranked by enrichment or grouped according to histone modification enrichment range. It would be much better to directly show scatterplots showing the histone mod enrichment versus delta mCG for each individual data point (2kb bin). Clearly, the statistical significance will always be high at this amount of datapoints, therefore I don't think this adds any meaning to the interpretation of the data.

This analysis contains a large number of datapoints (eg 485,159 2.5Kb windows for H3K9me3) and, in each individual analysis, the vast majority of windows lack the histone mark in question. This mean that scatter plots largely show the behaviour of windows that lack a particular mark. In addition, marks such as H3K9me3 are characterised by broad diffuse domains. This means that the difference in enrichment between positive and negative windows is far smaller than for marks with sharp focal peaks like H3K4me3 for which a bimodal distribution of enrichment values is apparent (Rebuttal Figure 1A). This means that it is difficult to discern the relationship between broad marks and DNA methylation in a scatter plot (Rebuttal Figure 1B). In our study we have used the binned histograms as an initial analysis to discern the trend before confirming our interpretation of them with more specific analyses of gene bodies, H3K9me3 domains, etc.

The boxplots in Appendix Fig. S1b have bins which are ranked by enrichment and contain an equal number of windows. We have clarified this in the figure legends.

Rebuttal Figure 1. Distributions of H3K4me3 and H3K9me3 ChIP-seq data compared to DNA methylation. a) density histograms of \log_2 ChIP/input values in 2.5Kb windows across the genome for H3K4me3 (left) and H3K9me3 (right). For H3K4me3 a shallow secondary peak is present with a mode of approximately $\log_2 2$. This corresponds to the bins with H3K4me3 peaks. A bimodal distribution is not visible for H3K9me3. b) Density scatter plots of change in DNA methylation between DNMT3B KO and HCT116 cells versus of \log_2 ChIP/input values in 2.5Kb windows across the genome for H3K4me3 (left) and H3K9me3 (right). Windows with H3K4me3 peaks are discernible with a mode of approximately $\log_2 2$. In both cases $n=485,214$ and $485,159$ windows for H3K4me3 and H3K9me3 respectively and \log_2 ChIP/input values are the mean of 2 biological replicates.

The HP1a - N-term interaction experiments are very interesting and could point to a mechanism that explains the observed increase of mCG at H3K9me3. The results presented are already well executed and convincing. I think it would be even stronger if the authors could remove HP1a in cells and measure DNA methylation, to test if they observe a similar reduction at H3K9me3, as seen in D3B KOs.

We agree with the reviewer that this would be an interesting experiment. However, there are three HP1 paralogues in human cells that have redundant functions (Bosch-Presegué et al 2017 Cell Reports, PMID: 29166597) and could potentially be involved in the recruitment of DNMT3B to H3K9me3-marked heterochromatin. The need to dissect this complexity means that such an experiment is beyond the scope of this manuscript.

We have added a sentence to the discussion of the manuscript to highlight this point:

Further work will be required to dissect the precise role that the N-terminal region and

interaction with HP1 α play in DNMT3B recruitment and whether DNMT3B also interacts with the other 2 HP1 paralogues found in humans (Jones et al, 2000).

Referee #3:

The authors addressed my comments.

We thank the reviewer for their comments.

Dr. Duncan Sproul
MRC Human Genetics Unit, University of Edinburgh
Western General Hospital, Crewe Road
Edinburgh, Edinburgh EH4 2XU
United Kingdom

Dear Dr. Sproul,

I am very pleased to accept your manuscript for publication in the next available issue of EMBO reports. Thank you for your contribution to our journal.

Yours sincerely,
